# Bridging two insect flight modes in evolution, physiology and robophysics

Jeff Gau[1,2,8], James Lynch[3,8], Brett Aiello[4,5,6,8], Ethan Wold[5,7], Nick Gravish[3✉] & Simon Sponberg[4,5✉]

Since taking flight, insects have undergone repeated evolutionary transitions between two seemingly distinct flight modes[1–3]. Some insects neurally activate their muscles synchronously with each wingstroke. However, many insects have achieved wingbeat frequencies beyond the speed limit of typical neuromuscular systems by evolving flight muscles that are asynchronous with neural activation and activate in response to mechanical stretch[2–8]. These modes reflect the two fundamental ways of generating rhythmic movement: time-periodic forcing versus emergent oscillations from self-excitation[8–10]. How repeated evolutionary transitions have occurred and what governs the switching between these distinct modes remain unknown. Here we find that, despite widespread asynchronous actuation in insects across the phylogeny[3,6], asynchrony probably evolved only once at the order level, with many reversions to the ancestral, synchronous mode. A synchronous moth species, evolved from an asynchronous ancestor, still preserves the stretch-activated muscle physiology. Numerical and robophysical analyses of a unified biophysical framework reveal that rather than a dichotomy, these two modes are two regimes of the same dynamics. Insects can transition between flight modes across a bridge in physiological parameter space. Finally, we integrate these two actuation modes into an insect-scale robot[11–13] that enables transitions between modes and unlocks a new self-excited wingstroke strategy for engineered flight. Together, this framework accounts for repeated transitions in insect flight evolution and shows how flight modes can flip with changes in physiological parameters.

Unlike the many insects that power each wingstroke with one-to-one 'synchronous' neural activation of flight muscles at up to approximately 100 Hz (Fig. 1a), some insect species require high power output at even higher frequencies. In these asynchronous species, the flight power muscles possess a delayed stretch activation response[2], which causes wing oscillations to self-excite without the need for regular timing from the nervous system (Fig. 1a). This delayed stretch activation is a physiological property of some muscles, in which an imposed stretch causes a time-lagged rise in tension, even under constant neural activation (Fig. 1b). Neural activation potentiates asynchronous muscle through the sustained release of calcium, but oscillations arise owing to the antagonistic action of two muscles, both with delayed stretch activation properties. Although insects with asynchronous muscle evolved from synchronous ancestors, these two modes of flight have been widely thought of as distinct strategies, but with multiple transitions between them[3,7]. However, with new phylogenies of flying insects and dynamic systems modelling of insect wing mechanics, we have the opportunity to reexamine this dichotomy and why repeated transitions can occur.

## A single origin of asynchronous muscle

We first examined the evolution of synchrony and asynchrony using maximum-likelihood phylogenetic state reconstruction[14] (Methods). We find that there has most probably been only one evolution of flight muscle asynchrony at the order level. There is an 86% probability that a single transition from synchronous to asynchronous flight in the ancestor of the clade of Thysanoptera, Hemiptera, Psocodea and holometabolous insects (node 200) occurred 407 million years ago (Fig. 1c, Extended Data Figs. 1–3 and Supplementary Tables 4 and 5). Although asynchrony was thought to have evolved seven to ten times throughout insect flight[2,3,6], previous analyses have not utilized phylogenetic ancestral state reconstruction. Only recently has an insect-wide phylogeny enabled resolution of the major orders[15]. We established the state of extant species from existing literature on muscle ultrastructure and histology (Supplementary Table 4). We first assumed an equal rates model of evolution and utilized a Markov chain Monte Carlo approach to estimate the state of ancient insects. In this reconstruction, there have been many independent reversions back to synchronous flight muscle from the single origin of asynchrony at the order level.

[1]Interdisciplinary Bioengineering Graduate Program, Georgia Institute of Technology, Atlanta, GA, USA. [2]George W. Woodruff School of Mechanical Engineering, Georgia Institute of Technology, Atlanta, GA, USA. [3]Mechanical and Aerospace Engineering Department, University of California San Diego, San Diego, CA, USA. [4]School of Physics, Georgia Institute of Technology, Atlanta, GA, USA. [5]School of Biological Sciences, Georgia Institute of Technology, Atlanta, GA, USA. [6]Department of Biology, Seton Hill University, Greensburg, PA, USA. [7]Quantitative Biosciences Graduate Program, Georgia Institute of Technology, Atlanta, GA, USA. [8]These authors contributed equally: Jeff Gau, James Lynch, Brett Aiello. ✉e-mail: ngravish@ucsd.edu; sponberg@gatech.edu

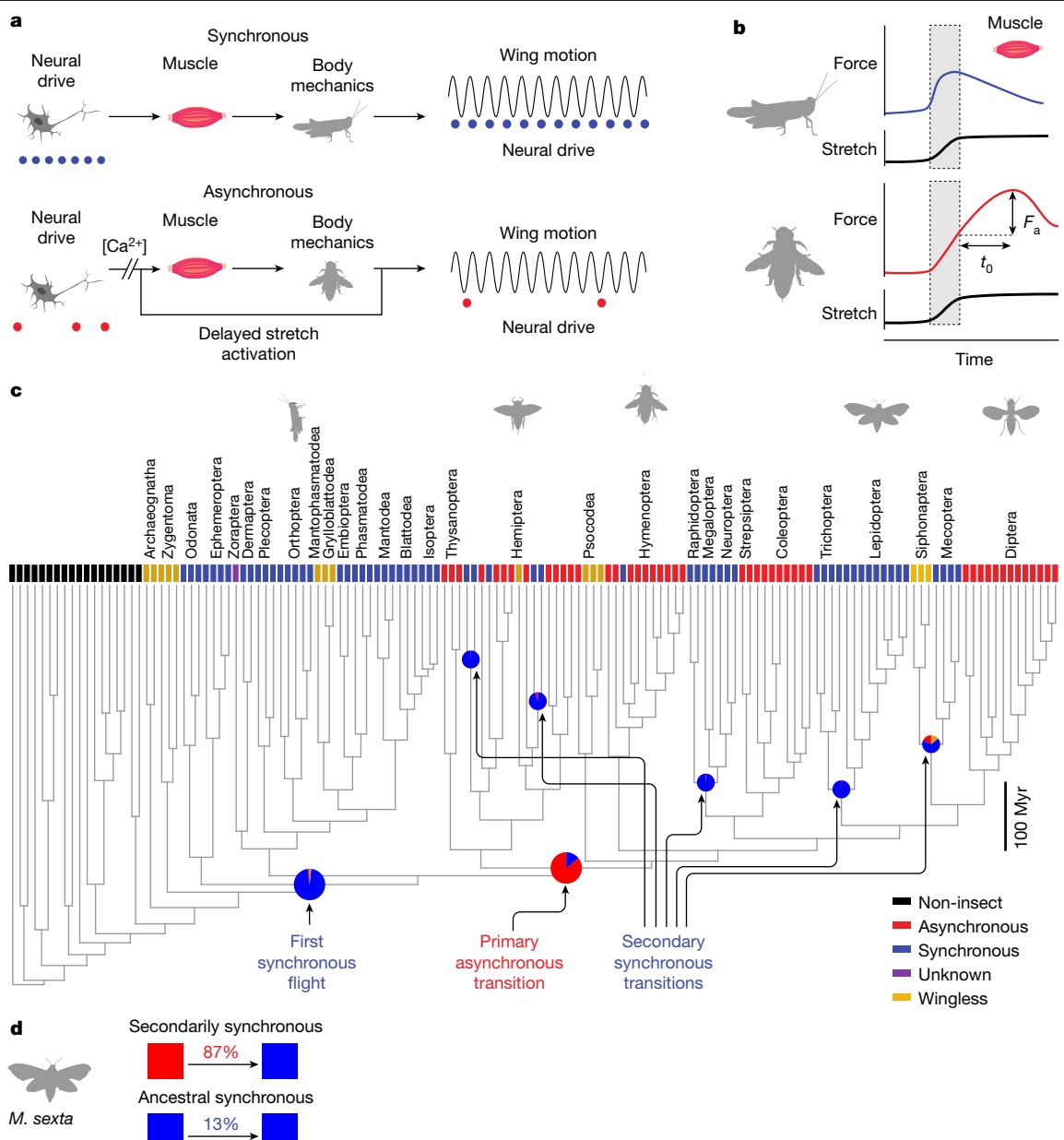

**Fig. 1 | Phylogenetic comparative analysis of insect wingbeat actuation reveals a probable single origin of asynchronous flight muscle.**
**a**, Synchronous muscle has a 1:1 relationship between neural activation (blue dots) and muscle contraction. Asynchronous muscle contraction is independent of the precise timing of neural activation (red dots), arising from delayed stretch activation[2]. **b**, The physiological signature of an asynchronous muscle is that when impulsively stretched it produces a delayed force of magnitude $F_a$ that peaks after a characteristic time $t_0$, determined by the rising and falling rate constants $r_3$ and $r_4$ (Methods). **c**, Ancestral state reconstruction[14] based on muscle ultrastructure (not physiology) reveals that a single evolutionary

origin of asynchronous fibre types is more probable using an insect-wide phylogeny resolved to the ordinal level[15]. Tip states were identified from the literature (Methods). Pie charts represent the posterior probabilities of the ancestral state reconstruction at these particular nodes given an equal rates model of evolution (full posterior probabilities in Extended Data Fig. 3 and Supplementary Table 5). **d**, By iteratively constraining ancestral nodes (Methods), we find an 87% posterior probability that some node ancestral to Lepidoptera and Trichoptera (including *M. sexta*) was asynchronous (making this clade secondarily synchronous) as opposed to all nodes ancestral to Lepidoptera being synchronous (ancestral synchronous). Myr, million years.

Species with this reversion are secondarily synchronous flyers (Fig. 1c and Extended Data Fig. 3). We found that Mecoptera, Lepidoptera, Neuroptera, Megaloptera and Raphidoptera are all most probably secondarily synchronous orders.

This pattern of transitions is consistent across alternative models of evolution. The best fit model (all transition rates different with ambiguous coding of wingless species) actually produced a 100% posterior probability for a single asynchronous origin at node 200. However, even if muscle structural data is available across most orders,

we still only have samples from a small number of all insect species. Therefore, we show the more conservative equal rates model (Fig. 1c). Incorporating heterogenous rates across the phylogeny[16,17] did not produce better model fits (Supplementary Discussion A). Ancestral state reconstruction can change with more sampling and different phylogenetic reconstructions, but the current best evidence supports a single origin of asynchrony at the order level. Most importantly, our analysis raises the possibility that the physiological properties associated with asynchrony, such as delayed stretch activation, could be

conserved in secondarily synchronous fliers. If so, this would provide evidence that both modes can co-occur across the phylogeny even if the muscle ultrastructure appears as a specific type.

## Latent asynchrony in synchronous fliers

Previous tests of the synchronous flight muscle of locusts (Fig. 1b) found no evidence for delayed stretch activation[2]. This contributed to the idea that delayed stretch activation was a specialization restricted to asynchronous muscle and that there is a dichotomy in muscle properties associated with the two flight modes. In the presence of tonic calcium levels maintained by a relatively slow neural drive, asynchronous muscles exhibit delayed stretch activation and also a delayed drop in force following shortening[2,3] (delayed shortening deactivation). These complementary effects enable power production by establishing a time delay between force and displacement. However, because orthopterans (including the ancestors of modern locusts) diverged from other insects before the first asynchronous fliers, the lack of delayed stretch activation in locusts may not generalize to secondarily synchronous insects (Fig. 1c). We next explored whether asynchronous muscle properties were conserved in the hawkmoth species *Manduca sexta*—a secondarily synchronous lepidopteran (Fig. 1d).

Unlike in the locust example, we identified a delayed increase in force following stretch in *M. sexta* flight muscle—the hallmark feature of delayed stretch activation (Fig. 2a,b). After reaching $203 \pm 44$ kN m$^{-2}$ during constant activation at 0% strain, we stretched the primary flight downstroke muscles (the dorsolongitundinal muscles (DLMs)) to 4.5% strain and observed a subsequent increase in stress of $32.1 \pm 9.9$ kN m$^{-2}$ that was delayed by $29.0 \pm 6.6$ ms after the conclusion of the stretch. However, following shortening we did not observe delayed shortening deactivation. Many stretch–hold–release–hold experiments on asynchronous muscle detect both delayed stretch activation and delayed shortening deactivation[2,18,19], whereas others observe delayed stretch activation without delayed shortening deactivation[20,21]. Delayed shortening deactivation may be driven by distinct molecular mechanisms and may not be a necessary feature for asynchrony. Thus, although *M. sexta* is a synchronous flyer, its flight muscle exhibits the necessary physiological properties to enable asynchronous flight.

The presence of delayed stretch activation in a synchronous insect creates a dilemma, because delayed stretch activation does not cause wingstrokes that are asynchronous in this moth species. This limitation could arise from ineffective timing or insufficient magnitude of the delayed stretch activation. The timing of the delayed stretch activation response is typically characterized by fitting a sum of three exponential terms[18] with rate constants $r_2$, $r_3$ and $r_4$ (red curve in Fig. 2c and Methods). The rate constants represent three phases of delayed stretch activation: a fast drop in tension ($r_2$) corresponding to the fall of the viscoelastic response (stress relaxation), a delayed tension rise associated with stretch activation ($r_3$), and a slow drop in tension as stretch activation decays ($r_4$). The rising rate of stretch activation tension ($r_3$) is linearly related to wingbeat frequency in asynchronous insects[18] and is linked to the rates of crossbridge attachment and detachment[22]. These relationships suggest that $r_3$ is the single critical parameter that establishes the time delay necessary for self-excitation.

We found that the relationship between the delayed stretch activation rate constant $r_3$ and the wingbeat frequency of 25 Hz in *M. sexta* is consistent with the broad scaling relationship observed by Molloy across asynchronous insects[18,23] (Fig. 2d and Methods). The stretch activation of hawkmoth flight muscle has the appropriate timescale to be asynchronous (Fig. 2c).

However, the magnitude of delayed stretch activation was only $36.2 \pm 13.6\%$ of the tetanic force (Fig. 2e). Direct comparison to literature values is difficult because of varying experimental conditions[24], but the ratio of delayed stretch activation magnitude to tetanus is typically between 100 and 300% in asynchronous beetles, waterbugs and

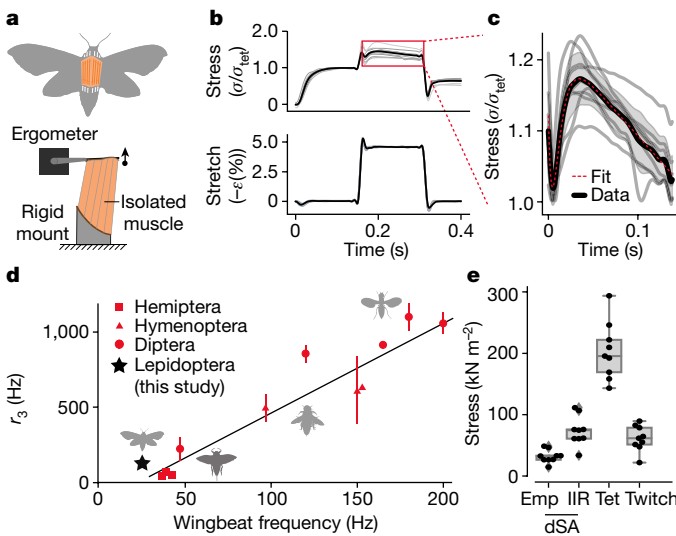

**Fig. 2 | Secondarily synchronous hawkmoth flight muscle exhibits delayed stretch activation, a hallmark of asynchronous flight. a**, Intact, downstroke flight muscle (DLMs) from *M. sexta* ($n = 9$ independent moths from the same source colonies, each sampled a single time) was mounted on an ergometer and electrically stimulated at 150 Hz to establish tetanus. Muscle viability was maintained with a saline drip at a constant 35 °C. **b**, We applied stretch–hold–release–hold strains, matching in vivo strain amplitudes[55] of 4.5% while measuring stress normalized to tetanus. Positive strain ($\varepsilon$) and force are defined in the shortening direction (opposite stretch). The black line denotes mean muscle stress normalized to tetanic stress, grey lines show individual trials. **c**, Magnification of the region outlined in **b** shows the delayed stretch activation response, characteristic of asynchronous muscle physiology. A sum-of-exponentials mathematical formulation of delayed stretch activation (equation (5); red line) accurately fits the mean normalized stress (black line; shaded region is ±s.d.). The initial transient is the viscoelastic response of the muscle and the subsequent rise and fall is the stretch activation. **d**, Despite being synchronous, the delayed stretch activation rising rate constant ($r_3$) of *M. sexta* lies near the prior empirical finding of a linear relationship between $r_3$ and wingbeat frequency[18] ($123.4 \pm 52.6$ s$^{-1}$ at 25 Hz; the black star shows the mean, error bars (obscured) show s.d.). Non-lepidopteran data and the black regression line are replotted from Molloy et al.[18], with error bars representing the full range of data. We scaled $r_3$ values to ambient temperature using published relationships (equations (2) and (3) from Molloy et al.[18]). **e**, Peak stress for *M. sexta* delayed stretch activation ($F_a$), tetanic force (Tet) and twitch. Delayed stretch activation (dSA) stress is shown with (IIR) and without (Emp) infinite impulse response correction (Methods). Box plots denote mean and quartiles, and whiskers are 1.5 × the interquartile range.

flies[18,25]. Even correcting for the non-instantaneous stretch used in the physiological experiment, the idealized delayed stretch activation response (an infinite impulse response – IIR) is still far below tetanic force in *M. sexta* (Fig. 2e). Thus *M. sexta* seems to be a synchronous flyer not because it lacks the physiological capabilities for asynchronous activation, but rather because it occupies a region of delayed stretch activation parameter space in which asynchronous forces are not sufficient to dominate the neurally driven activation and relaxation of flight muscle (synchronous forcing). Additionally, *M. sexta* muscle probably reuptakes calcium more quickly than most asynchronous muscles, further reducing the contribution of the delayed stretch activation to in vivo flight conditions.

However, if the delayed stretch activation magnitude were larger, then it is possible that *M. sexta* could generate asynchronous wingbeats. It is not known precisely what mechanism controls the magnitude and rate of delayed stretch activation[26], but it is dependent on calcium levels and likely involves recruitment of additional myosin heads to actin binding sites (cross-bridges) through stretch-sensitive

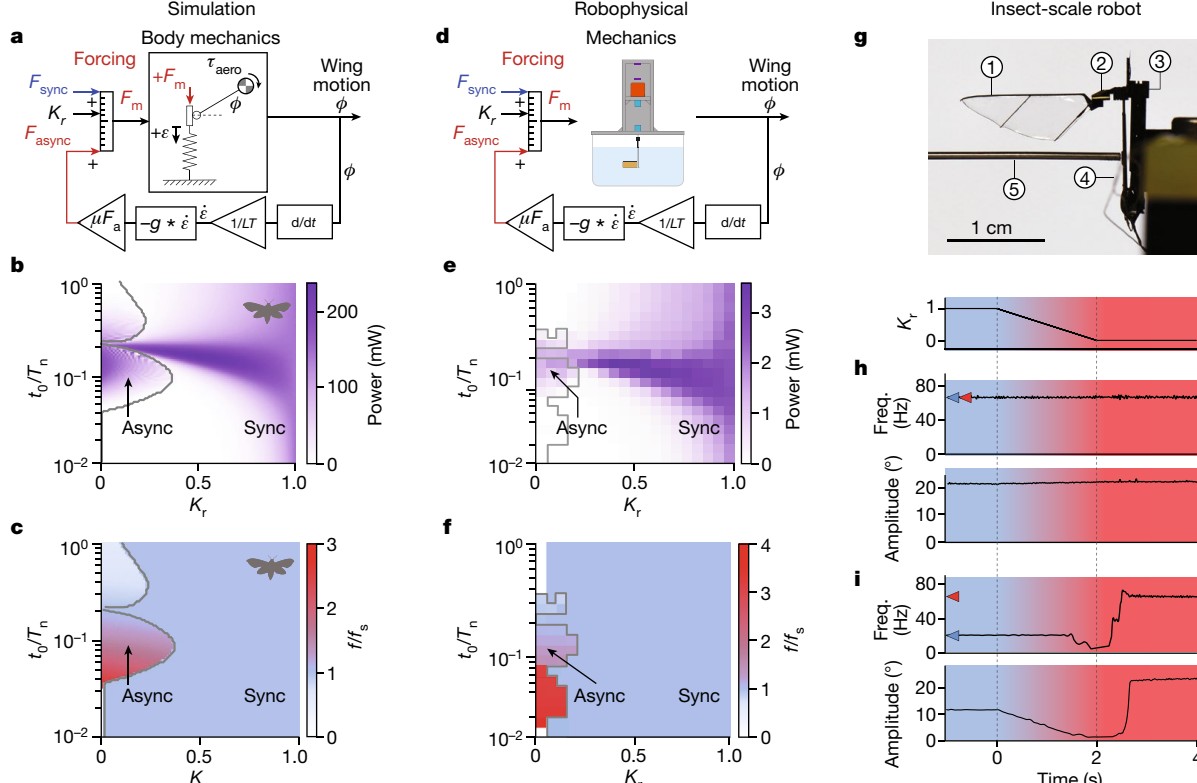

**Fig. 3 | Transitions between synchronous and asynchronous modes in simulation and robotics. a**, A unified biophysical model combines hawkmoth body mechanics (equation (3)) with time-periodic, neurogenic (synchronous) and delayed stretch activation (asynchronous) forcing. Stretch activation is implemented as a feedback filter (or convolution) of wing angle ($\phi$) converted to muscle strain rate ($\varepsilon$) and scaled to wingstroke conditions ($\mu F_a$) (equations (10) and (11)). The parameter $K_r$ interpolates between the two sources of muscle force (equation (1)). **b**, $K_r$ and stretch activation time-to-peak normalized to the mechanical natural frequency ($t_0/T_n$) define a parameter space. High-power flapping occurs at both extremes, but intermediate modes only generate appreciable power along a bridge where the rate of stretch activation approximately matches the synchronous drive (25 Hz). *M. sexta* is plotted on the basis of estimates of $t_0/T_n$ and $\overline{K}_r$ from quasi-static data. **c**, Emergent wingbeat frequencies ($f$) normalized by the synchronous drive frequency ($f_s$). Dark blue indicates regions where the emergent wingbeat frequency is

entrained to the synchronous driving frequency ($f = f_s$). The red regions indicate where the asynchronous dynamics dominate ($f \neq f_s$). The grey line indicates the boundary between synchronous- and asynchronous-dominant dynamics. **d**, A robophysical system (roboflapper) implementing both types of actuation, plus real-world fluid physics and friction. **e,f**, Results from the setup in **d** are qualitatively similar to the simulations in **b,c**, but with a region of no wingstrokes due to system friction with low $K_r$ and high $t_0/T_n$. **g**, A centimetre-scale robotic wing modelled after the Harvard robobee[12], consisting of (1) a wing; (2) a transmission; (3) a carbon fibre frame; (4) a piezoelectric bending actuator; and (5) a wing displacement sensor. **h**, A single hybrid robobee transitioning from synchronous ($K_r = 1$, blue) to asynchronous ($K_r = 0$, red) in real time. Transitions are smooth when synchronous and asynchronous frequencies are approximately equal (blue and red markers, respectively). **i**, When the frequencies differ, interference causes frequency and amplitude fluctuations in the transition regime.

myofilament proteins[21,27]. In asynchronous insects, neural activation typically only recruits 30% of cross-bridges, which explains why the stretch activation can far exceed tetanic activation[28]. One possible way this could be regulated is by the different isoforms of the regulatory molecule troponin, which promotes release of myosin-binding sites[29]. Different troponin isoforms are calcium-activated or mechanically activated. The ratio of these is correlated with asynchronous force output[24,30]. Surprisingly, the stretch-activated troponin isoforms found in asynchronous insects are also found in *M. sexta*[30,31]. This provides one possible mechanism for residual delayed stretch activation in moths. Our physiological results indicate that delayed stretch activation can be present in quite reduced magnitudes, and it is already known that the rate constants can vary widely[18]. The flight muscles of different orders with asynchronous (and synchronous) flight modes may have further specialized, especially in extreme cases of performance. This may contribute to the molecular differences observed in some groups[32–34]. Our results show that conserved molecular components are potentially part of the same continuous dynamical parameter space that spans across synchronous and asynchronous flight modes.

## Unifying dynamics for two flight modes

The presence of delayed stretch activation in a synchronous insect, coupled with evidence of many evolutionary transitions from asynchrony to synchrony, suggests that synchronous and stretch-activated contractions can be two regimes of a single actuation strategy. Building on the extensive characterization of synchronous and asynchronous flight muscle[7,18], quasi-steady flapping aerodynamics[23,35,36] and body mechanics[37–39], we next developed a biologically grounded model of insect flight in which we can control the relative contributions of synchronous and asynchronous forcing (Fig. 3a). We coupled models of both synchronous and asynchronous forcing to an established model of *M. sexta* mechanics that includes the elasticity of the deformable exoskeleton, wing inertia and aerodynamic loads[38,39] (Fig. 3a).

We developed a model of delayed stretch activation that captures the strain-dependent force output of asynchronous muscles (Methods). The active muscle tension response to a stretch–hold–release–hold (step) strain input can also be thought of as the impulse response of the muscle to a strain rate input. The dynamics of delayed stretch

activation are modelled as a transformation of a strain rate input into muscle tension force output, similar to a dynamical filter, which then feeds back into the muscles' force production. Since delayed stretch activation is the impulse response to a strain rate input, and since prior experiments have demonstrated that, for low amplitudes, the delayed stretch activation response is linear[40], we may apply the delayed stretch filter to continuously varying patterns of strain to produce continuous muscle tension, such as during a wingstroke.

We accomplish this by defining a convolution expression $F_{async}(\dot{\varepsilon}, t) = \mu F_a(-g * \dot{\varepsilon})(t)$ and fit it to the stretch–hold–release–hold response, where $F_a$ is the magnitude of the asynchronous forcing (Fig. 1b), $\dot{\varepsilon}$ is the muscle strain rate, and $g$ is the velocity impulse response (Fig. 3a, Extended Data Fig. 4 and Methods). We include $\mu$ as a scaling factor that is determined through a fitting procedure such that a purely asynchronous force is able to elicit wingstrokes with a realistic sweep angle of 117° in a model hawkmoth within a range of timescales encompassing the hawkmoth stretch activation. The negative sign in front of the convolution indicates that a negative strain (muscle stretch) induces a positive (shortening) force response, since muscle physiology conventions define positive in the shortening direction. The muscle strain rate scales with wing angular velocity by a factor of $LT$, where $L$ is the resting length of the muscle, and $T$ is the transmission ratio of angular wing displacement to linear muscle displacement, $LT\dot{\varepsilon}(t) = \dot{\phi}(t)$. Values of $L$ and $T$ are taken from the literature[41,42]. To validate our delayed stretch activation model, we showed that it could capture the asynchronous response of *M. sexta* from our experiments and also that it could reconstruct the muscle power curves of *Lethocerus indicus* and *Vespula vulgaris*, which are asynchronous species (Supplementary Discussion B).

We also define synchronous forcing, $F_{sync}(t) = F_s \sin(2\pi f_s t)$, where $f_s$ is the synchronous wingbeat frequency (25 Hz). $F_s$ is the synchronous forcing amplitude defined as the force necessary to elicit physiologically realistic wingstrokes in our model under purely synchronous activation (2,720 mN in *M. sexta*; Methods). We then combined both types of forcing via an interpolation factor, $K_r \in [0, 1]$, to obtain the total muscle force, $F_m$, where

$$F_m(\dot{\varepsilon}, t) = K_r F_{sync}(t) + (1 - K_r)F_{async}(\dot{\varepsilon}, t) \qquad (1)$$

The value of $K_r$ reflects the relative importance of synchronous versus asynchronous forcing in the system. Biologically, a high $K_r$ means that the force and crossbridge recruitment due to neural activation is large compared to the crossbridge recruitment due to stretch activation. The sensitivity of flight muscle to calcium compared to the stretch-sensitivity of the myofilaments gives a plausible mechanism for $K_r$ to vary across species and over evolutionary timescales. Because in-flight measurements of the relative contributions of $F_{async}(t)$ and $F_{sync}(t)$ are unavailable, we estimate $K_r$ as $\widetilde{K}_r$ using the proportion of synchronous to total force (synchronous and asynchronous) measured from isolated muscle under quasi-static conditions (see Methods),

$$\widetilde{K}_r = \frac{F_{tet}}{F_a + F_{tet}} \qquad (2)$$

In *M. sexta*, for example, $\widetilde{K}_r$ is relatively high (0.86) reflecting that the magnitude of the delayed stretch activation response is low compared to the forces generated via neural activation alone. Asynchronous species produce a delayed stretch activation force several times higher than isometric tetanus[25] and would have a very low $\widetilde{K}_r$. Although the exact biological correlate of $K_r$ and asynchronous forcing is not known, one possible physiological interpretation of $K_r$ is the proportion of calcium-activated troponin isoforms to the total number (calcium-activated plus mechanically activated) that are activated under flight conditions. By adjusting $K_r$ from fully synchronous ($K_r = 1$) to fully asynchronous ($K_r = 0$), we can explore the emergent interactions of synchronous and stretch-activated forcing in the same system at intermediate values of $K_r$.

The interactions between strain-dependent forcing and the passive mechanics of the insect flight system have a key role in establishing self-excited oscillations. To incorporate these interactions, we first modelled aerodynamic damping using a quasi-steady approximation, with aerodynamic torque equal to wing angular velocity squared, multiplied by a coefficient ($\Gamma$) that accounts for wing shape, air density and experimentally measured drag coefficients[23] (equation (14)). We then used prior estimates of *M. sexta* wing inertia[23] ($I$), thorax elasticity[38] ($k$) and transmission ratio[42] ($T$) (Extended Data Fig. 1). Using the fact that measurements of the *M. sexta* thorax are well approximated by a linear elastic spring in parallel with muscle[38], this yields our mechanics model, which we refer to as a 'spring–wing' system:

$$\frac{F_m}{T} = I\ddot{\phi} + \Gamma|\dot{\phi}|\dot{\phi} + \frac{k}{T^2}\phi \qquad (3)$$

This equation captures the indirect actuation of synchronous or asynchronous insect flight muscle, which act via the deformation of the thorax in parallel with the muscle to sweep the wings back and forth.

To reduce the complexity of the delayed stretch activation model we combine $r_3$ and $r_4$ into one timescale, $t_0$ (Methods), which is the rise time to peak force (Fig. 1b). To compare across systems we then normalize this time to $T_n$, the natural period of the wing–thorax system. $T_n$ is determined by the body mechanics alone,

$$T_n = 2\pi\sqrt{\frac{I}{k}} \qquad (4)$$

The interactions between synchronous forcing amplitude and frequency, delayed stretch activation rates, and mechanical time constants define a parameter space that encompasses both synchronous and asynchronous oscillations. As expected, while *M. sexta* does have delayed stretch activation, it is firmly in the synchronous regime ($K_r = 0.88$, $t_0/T_n = 0.54$). Its wingstrokes are largely unaffected by delayed stretch activation (Fig. 3b). Delayed stretch activation, while present in *M. sexta*, has been reduced to a point where it is less consequential than synchronous activation at steady state, although it may still have a role under perturbed conditions with faster strains and frequency modulation[43].

Using our hawkmoth mechanics, we simulated the rest of the parameter space (Fig. 3b,c). The asynchronous regime is capable of generating large amplitude limit-cycle oscillations even with hawkmoth mechanics, but only when synchronous forces are much smaller than asynchronous—that is, $K_r \ll 1$. As the time to reach peak force of the asynchronous muscle ($t_0$) is increased, we observe a bifurcation where asynchronous wingbeats appear as $t_0/T_n$ crosses a critical value (see Extended Data Fig. 5). When $t_0$ is small the muscle tension increase is faster than the natural oscillation frequency of the body, and thus the delayed stretch activation acts as a brake. However, when $t_0$ is large enough (that is, the muscle response is sufficiently slow), the delayed stretch activation force is produced during the contraction phase and self-excited oscillations occur. We found that these regimes of qualitatively distinct oscillations, one periodically forced (synchronous) and one self-excited (asynchronous), are both able to generate wing kinematics with comparable amplitudes and frequencies. However, as we transition between these two regimes by varying $K_r$, we observed complex dynamics where synchronous and asynchronous modes interact.

## Transition dynamics between flight modes

A major function of the flight musculature is to power flight. Therefore, a gradual transition between synchrony and asynchrony is only

evolutionarily feasible if a set of high-power, steady, periodic oscillations connects the two regimes. Otherwise, intermediate evolutionary steps between synchrony and asynchrony would result in individuals that could not fly. On the basis of our simulations, smooth transitions between the synchronous and asynchronous modes are possible, but only with appropriate matching of the muscular and mechanical timescales. At intermediate values of $K_r$, synchronous and asynchronous dynamics are both present, and high-power oscillations only occur along a 'bridge' where the synchronous and asynchronous dynamics do not interfere (Fig. 3c). Thus, transitions in insect flight actuation modes are possible across this parameter space, but cannot occur when muscle parameters and body mechanics diverge.

For the hawkmoth, the origin of the bridge occurs at $t_0/T_n \approx 0.2$ along the $K_r = 0$ axis, which is the location in parameter space where the asynchronous emergent frequency and the synchronous frequency exactly match (Extended Data Fig. 5). As the synchronous forcing becomes stronger relative to the asynchronous dynamics, we see that the region near $t_0/T_n \approx 0.2$ becomes entrained to the synchronous frequency. Entrainment is the process where a self-excited oscillating system is forced to oscillate exactly at the frequency of an external driving frequency (resulting in a phenomenon called an Arnold tongue[44]). As we move away from the bridge along the $t_0/T_n$ axis, there is a bifurcation, and the asynchronous and synchronous frequencies diverge, ending the entrainment, and leading to emergent asynchronous oscillations (Extended Data Fig. 6 and Supplementary Discussion C). Crossing between these two regimes leads to interference between these oscillatory modes, thus leading to lower power, less smooth flapping trajectories that are unsuitable for flight (Extended Data Figs. 7 and 8). The grey lines in Fig. 3b,c,e,f illustrate the boundary between the synchronous and asynchronous dynamics. Thus, although complex aerodynamic phenomena[45] and sensorimotor feedback systems[8] can exhibit unpredictable flapping wing behaviour, our results indicate that even simplified fluid and body mechanics under combined synchronous and asynchronous actuation are sufficient to lead to erratic wingbeat dynamics.

Matching muscular and mechanical timescales is evidently a critical requirement for both synchronous and asynchronous power production. However, variation in the strength of the delayed stretch activation response (changing $K_r$), its timescale ($t_0$) or the resonant mechanics of the thorax and wings ($T_n$) could enable smooth, gradual transitions across the bridge, especially over evolutionary timescales. Biologically, these parameters will be closely tied to the molecular components of the delayed stretch activation, such as crossbridge binding, calcium responsiveness and the troponin isoforms mentioned above. Evolutionary transitions need not necessarily be smooth, but our model and analysis reveals the existence of a pathway for gradual transitions between a fully synchronous and asynchronous regime even while maintaining high-power wingstrokes. This bridge may have facilitated the many subsequent shifts between asynchrony and synchrony in insects (Fig. 1c). However, clades such as Lepidoptera, which appear uniformly synchronous, may have subsequently specialized away from the bridge, reflected by the location of *M. sexta* in the model simulation.

## Transitions in a robophysical model

To test the hypothesis that insects can realize both synchronous and asynchronous oscillations simply by changing a ratio of timescales and an interpolation factor even with realistic environmental interactions, we built a dynamically scaled robophysical spring–wing system, or roboflapper (Fig. 3d and Extended Data Fig. 4). Unlike previous robophysical investigations of flapping wing flight[46,47], we did not directly prescribe wing angle versus time in our roboflapper. Instead, we provided torque commands to a motor that were either feedforward periodic (synchronous, sinusoidal forcing) or generated by velocity feedback using a real-time implementation of our delayed stretch activation dynamics model (see Methods). The wing dynamics and frequency were emergent properties. To mimic aerodynamic damping and the body elasticity of indirect actuation, the motor drove a dynamically scaled wing[39] in parallel with a custom-moulded silicone torsion spring. Experiments were run on a $20 \times 20$ grid of parameters with outputs averaged over 15 s of steady-state data.

High-power synchronous and asynchronous regimes emerge in the robophysical model as in the hawkmoth simulations (Fig. 3e,f and Supplementary Video 1). As in simulation, these regions are connected by a narrow bridge that enables high-power transitions between the two regimes where the synchronous frequency matches the asynchronous frequency. Unlike the simulation, the fully asynchronous roboflapper at $K_r = 0$ does not oscillate when $t_0/T_n$ exceeds approximately 0.3, probably owing to friction and viscous damping (Supplementary Discussion D) that are present in the experiment and not the simulation (Fig. 3f).

## Asynchrony in an insect-scale robot

The robophysical model tested our dynamics framework over a wide range of parameters in a real system. We next test whether these dynamics could produce both synchronous and asynchronous oscillations at the scale of an insect. Demonstrating synchronous to asynchronous transitions at the centimetre scale is important because unsteady aerodynamics do not necessarily scale as quasi-steady phenomena and mechanical systems at small scales can have unexpected emergent behaviour[48]. Moreover, state-of-the-art insect-scale robots currently utilize a time-periodic voltage ('synchronous') input to excite a piezoelectric actuator at the resonance frequency of the mechanical system (for example, the Harvard 'robobee'[11,12]). The robobee can achieve untethered flight, but only if there is a sufficient power source[13]. It uses wingbeat frequencies[12,49] (50–170 Hz) similar to those of many asynchronous insects[18], yet relies on time-periodic actuation. This prompted us to explore whether a similar platform can generate self-excited oscillations with the addition of delayed stretch activation (Fig. 3g).

To generate delayed stretch activation in the robobee, we used a fibre-optic displacement sensor to estimate wing velocity. The instantaneous wing velocity was supplied to the same real-time delayed stretch activation dynamics model as in the robophysical model but with parameters adapted for the robobee. The output of this model was converted to a voltage that was amplified and supplied to the piezoelectric actuator. Thus, we were able to establish a real-time feedback loop between wing velocity and actuator voltage that could generate the first asynchronous wingbeats in the robobee. We found that we could generate stable oscillations in both the fully asynchronous and fully synchronous modes.

Having shown that we can generate asynchronous flapping in an insect-scale robot, we wanted to see whether the same system could transition smoothly between synchronous and asynchronous modes as $K_r$ varies, as we predict insects may have done over evolutionary timescales (Fig. 3h,i, Extended Data Fig. 9 and Supplementary Videos 2 and 3). We combined the outputs of feedforward synchronous actuation and delayed stretch activation (equation (1)) in real-time experiments. The synchronous forcing frequency $f_s$ was set to either: (1) match the emergent oscillation frequency of the fully asynchronous system, 67 Hz (with $r_3 = 225$, $r_4 = 135$; Fig. 3h and Supplementary Video 2); or (2) not match the asynchronous system—that is, $f_s = 20$ Hz (Fig. 3i, Extended Data Fig. 9 and Supplementary Video 3). We started by setting $K_r = 1$ (fully synchronous) and allowing the system to reach a stable amplitude. Then, we changed $K_r$ linearly from 1 to 0 over 2 s in the Simulink real-time control system. When $f_s = f_a$, there is no appreciable change in amplitude, and high-power oscillations are maintained across the full range of $K_r$ (Fig. 3h). However, when $f_s \neq f_a$, interference causes the amplitude to decrease as the asynchronous and synchronous dynamics interfere (Fig. 3i). Eventually, oscillations

develop in the fully asynchronous system, but only after interference from synchronous dynamics is no longer present. Further illustrative experiments on the insect-scale robot (Extended Data Fig. 9) and dynamically scaled system (Extended Data Fig. 10) demonstrate that these transitions are robust and reversible, but only smooth when on the bridge.

By capturing one of the key evolutionary innovations that enabled high frequency insect flight, this framework may unlock the potential for robotic systems to benefit from both asynchronous and synchronous actuation modes. For insects, asynchronous muscle enabled the decoupling between muscle contractions and neural input that enables wingbeat frequencies to exceed the limits of neural firing frequency[2,3,5,50]. An asynchronous flapping wing robot may benefit from this decoupling of power and control. Moreover, the ability to transition between synchronous and asynchronous modes suggests opportunities for even more versatile and adaptive control.

## Shared dynamics across flight modes

Through the introduction of a dynamics model for delayed stretch activation, we have revealed new insights into asynchronous wingbeat generation in insect flight. When combined with synchronous actuation, aerodynamics, and body mechanics, this unified spring–wing framework recapitulates the transition between synchronous and asynchronous regimes. Furthermore, both types of actuation can coexist even when a dominant wingbeat strategy emerges, reflecting the presence of delayed stretch activation in the synchronous flight muscles of the moth *M. sexta*. Overall, broader physiological testing of delayed stretch activation, especially in other synchronous species and those close to the bridge, may further resolve the nuances in these two modes of insect flight. Mapping specific parameters of stretch-activated myosin recruitment, delayed stretch activation time constants, troponin isoform ratios and calcium activation would connect the potential molecular basis of asynchronous and synchronous flight to the model parameter space across more species. This framework also provides a starting point for the examination of how more complex models of body dynamics, muscle force production and aerodynamics contribute to the emergent wingstroke oscillations of flapping wing insects.

The coupling of indirectly actuated wings to an elastic thorax (spring–wing mechanics) with both periodic neural activation and delayed stretch activation enables multiple solutions to the challenges of high-power, periodic wingstrokes. Given that the capacity for asynchronous flight was gained and then preserved even in secondarily synchronous descendants (Figs. 1d and 2c), transitions between the two flight modes are not necessarily caused by a switch in morphology or physiology. However, asynchronous and synchronous flight muscle do have different ultrastructure and can show molecular adaptations to each mode of flight[3,5,6,32,51,52]. Still, their physiological properties (embodied in our model by $K_r$ and $t_0/T_n$) can manifest on a continuum. This may explain the multiple evolutionary transitions between asynchrony and synchrony within insects. It is likely that highly specialized fliers—such as many dipterans and hymenopterans—have further specializations to enhance asynchronous flight[26,32,33,53,54], and these may prevent these orders from having reversions to synchronous flight. However, these specializations also do not preclude a common underlying physiological mechanism for delayed stretch activation which can vary in magnitude and timing. Supporting this, we see multiple asynchronous–synchronous transitions in the earlier diverging orders such as Hemiptera (Fig. 1c). This also suggests that hemipterans and other orders with multiple transitions may have muscle physiological parameters closer to the bridge in parameter space, thereby enabling more frequent transitions.

The evolutionary history of insects has shown a great deal of diversification in flight strategies. Central to these patterns are the transitions between synchronous and asynchronous modes. Together, our evolutionary reconstructions, muscle physiology results, dynamics simulations and robotic models show that the capacity for both synchronous and asynchronous flight can exist in the same system. Moreover, we demonstrate that when synchronous and asynchronous actuation modes act harmoniously there can be a smooth evolutionary pathway (a bridge) between asynchronous to synchronous regimes that enables high-power wingbeats across these two extremes.

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

# Methods

## Ancestral state reconstruction

**Muscle type labelling.** We encoded the orders Odonata, Ephemeroptera, Dermaptera, Plecoptera, Orthoptera, Embioptera, Phasmatodea, Mantodea, Blattodea, Isoptera, Raphidioptera, Megaloptera, Neuroptera, Trichoptera, Lepidoptera and Mecoptera as synchronous and the orders Thysanoptera, Strepsiptera, Coleoptera and Diptera as asynchronous[3,5,6,51,52]. Muscle type in Zoraptera remains unknown, and the orders Mantophasmatodea, Grylloblattodea, Siphonaptera are all wingless. The three remaining orders, Hemiptera, Psocodea and Hymenoptera are known to have both synchronous and asynchronous species[3,5,6,52,56]. The muscle type for each tip species of the insect phylogeny and its associated reference are included as a raw data file (Supplementary Table 4, 'Species muscle type with sources'). We also conducted a more detailed literature analysis on all orders.

**Psocodea.** Psocodea is a particularly notable clade that has the potential to strongly influence the ancestral state and single origin of asynchronous muscle. This order is historically considered to have mixed muscle types[3] based on the muscle structure data for various species[6]. Of the tip species present in the phylogeny used in this study, we were able to discern muscle types based on muscle structural data or absence of wings (see Fig. 1). Muscle structure data for the longest-branched genus (*Ectopsocus*) are inferred from a closely related species investigated by Cullen[6] and the wingless state of other tip species was determined through multiple sources[57,58]. Additional confirmation of the states of this group was completed by cross referencing the phylogeny used in this study (Fig. 1) with a more densely sampled phylogeny of Psocodea[59] and searching for additional muscle type data for Psocodea species not represented in the phylogeny used in this study. We found that one species (*Trogium pulsatorium*) belonging to the most ancient suborder within the group (Trogiomorpha) is reported to have synchronous flight muscle[6]. The next most ancient subclade (suborder: Psocomorpha) within Psocodea is known to have multiple species with asynchronous flight muscle based on its structure[6]. Data from a more recently diverging clade, the Amphientometae infraorder (within the Troctomorpha suborder), are absent and the remaining species of the larger clade (Troctomorpha), where Amphientometae is nested, are wingless[57,58].

All evidence together suggests that the ancestral state of Psocodea is asynchronous. However, there remains uncertainty in this group due to poor muscle data and their large degree of winglessness. Using scaling relationships based on body mass and measurements of Psocodea wing sizes, it seems likely that most winged psocodean species fly with wingbeat frequencies well over 100 Hz (150–500 Hz), even when we allow body mass to differ by an order of magnitude[60–62] from 0.1 mg to 1 mg. Wingbeat frequencies above 100 Hz are strongly associated with the evolution of asynchronous flight muscle. Second, in support of our scaling argument, other authors report that all species of winged Psocodea are asynchronous based on their necessarily high wingbeat frequencies[63], directly conflicting with the muscle type data from Cullen[6]. Finally, the winged clade of unknown muscle type, Amphientometae, has a most recent common ancestor with the Psocomorpha clade, which does possess asynchronous muscle[6]. If Amphientometae does have asynchronous muscle as expected, this would most probably result in an asynchronous ancestral condition of the entire Psocodea order. Thus, multiple lines of evidence support asynchronous muscle type as the ancestral condition of this clade with possibly a single, independent reversion to synchrony within the Trogiomorpha clade, which is not present in the phylogeny used in this study.

**Hemiptera.** Although Hemiptera is another clade with interspecific variation in muscle type we are again confident in our reconstruction of the ancestral state. First, many species of Hemiptera were investigated in ref. 6 and the clade is relatively well sampled in the phylogeny used in this study. In addition, we mapped additional muscle type data from ref. 6 onto a more densely sampled Hemiptera phylogeny[64] for at least one species from most (19 out of 29) families. Here, we find that all investigated species from the Heteroptera suborder, which includes 20 (11 of which have been investigated) of the 29 families present in the phylogeny in Johnson et al.[64], use asynchronous flight muscle[6]. Second, two of the four longest branch families with the most ancient diverging Hemiptera suborder (Sternorrhyncha) are also asynchronous. The Sternorrhyncha suborder is only represented by two synchronous species in the phylogeny used in this study and thus is likely overweighted when inferring the ancestral condition of this group. Despite that, we still recover an ancestral condition of asynchronous muscle. Other authors reviewing known muscle types and flight neuromechanics also concluded that most Hemiptera species rely on asynchronous flight muscle[63].

Two additional phylogeny tips within Hemiptera from the genera (*Xenophysella* and *Nilaparvata*) did not have published muscle structure data from any species within the same family, warranting further investigation. First, we code the *Xenophysella* tip as wingless because the majority of investigated species (24 out of 25) from the Coleorrhyncha suborder that includes *Xenophysella* are reported to be flightless[65]. Second, we code the *Nilaparvata* tip as asynchronous for the following reasons: *Nilaparvata* myofibril diameter has been reported as 1.8 µm in insects three days post-emergence[66], which is above the 1.5-µm threshold for differentiating synchronous from asynchronous muscle[6,56]. From transmission electron microscopy of *Nilaparvata*, the sarcoplasmic reticulum appears to be sparse[66] which is a proposed hallmark of asynchronous muscle[6]. Despite this evidence, there is still some uncertainty about the *Nilaparvata* muscle type for the following reasons: a different species (*Sogatella furcifera*) from the same family (Delphacidae) is reported to have muscle with myofibril diameters ranging between 1.5 and 2.0 µm 3 days post-emergence[67], which also falls on the border of the diameter (1.5 µm) used to differentiate muscle type in this group[6]. Further, *Nilaparvata* belongs to the Fulgoroidea superfamily[64]. The Fulgoroidea superfamily shares a most recent common ancestor with the Membracoidea superfamily based on the Hemiptera phylogeny in Johnson et al. (2018)[64], and Membracoidea contains species of both synchronous and asynchronous muscle type[6], making the identification of *Nilaparvata* equivocal based on its phylogenetic position alone. However, the most direct histological evidence supports our classification of *Nilaparvata* as asynchronous.

The high variation of muscle type within Hemiptera makes this clade particularly interesting for future studies on the evolution of synchronous and asynchronous muscle physiology and structure. Based on our assessment of muscle type across Hemiptera, there appears to have been multiple reversions back and forth between the two types, where both types have likely evolved at least once from an ancestor of the other type. These bidirectional transitions within Hemiptera support the thesis that muscle physiology lies on a continuum rather than as two discrete types and may transition across the bridge in parameter space (Fig. 3b, e). Despite the diversity within Hemiptera, the reconstruction of the ancestral node is confidently asynchronous.

**Hymenoptera.** All Hymenoptera muscle types were assigned based on published muscle structures and supported by other investigations of muscle physiology. As noted above, variation in muscle myofibril diameter is directly related to muscle type, where myofibril diameters less than 1.5 µm are considered synchronous muscle[6]. Myofibril diameter was measured in 46 species distributed across the Hymenoptera phylogeny[68]. All but one of the 46 species are reported[56] to have myofibril diameters greater than 1.5 µm. A second line of evidence that relies on the muscle being defined as 'close-packed' versus 'fibrillar' supports these results[56]. Thus, while Hymenoptera is considered to be a group of mixed muscle type[3], we find the presence of synchronous muscle to be relatively rare. In support of these conclusions, ref. 63 also reports that most species of Hymenoptera rely on asynchronous flight muscle.

**All other clades.** All other insect orders are reported to be invariant in flight muscle type or are known to be completely wingless. Therefore, we relied on their invariant classification from published reviews[2,3,52] where authors used total evidence from both muscle structure and physiology data across species of each order to determine the muscle type classification of each order. However, we do code the single Zoraptera tip as unknown because we could not find any data on any species from this order. The coding of this tip as synchronous, which would be the most conservative classification, does not impact our results.

**Implementation.** We used a previously published molecular phylogeny grounded in fossil records spanning all insect orders[15], which modifies the fossil calibration of the extensive insect phylogeny developed by Misof et al.[69]. For ancestral state reconstruction, we assumed an equal rates model of evolution and used maximum-likelihood estimation to estimate the posterior probability of ancestral states using the Phytools R Package[14]. These analyses were performed in RStudio (v. 1.1.383) using R (v. 4.0.2). In Supplementary Discussion A we test other models of evolution using different character states, allowing all rates to be different rather than equal, and consider hidden rates models, using the *Phytools* and *corHMM* R Packages[16,70]. The latter allows for heterogenous rates of evolution but quickly increases the number of parameters in the model[16,17,70]. All models with a single rate class are consistent with the order-level reconstruction of synchronoy and asynchrony, including the single origin of asynchrony at node 200 (Thysanoptera, Hemiptera, Psocodea and holometabolous insects). Some models with multiple rate classes can produce more ambiguous reconstructions with more possible patterns but are overfit as assessed by Akaike information criterion values (see Supplementary Discussion A for further discussion).

We also determine the posterior probability that the Lepidoptera plus Trichoptera clade evolved from synchronous ancestry without reverting from a single asynchronous ancestor (for example, there were multiple, independent, more recent evolutions of asynchronous muscle for different synchronous orders). To do so, we first recorded the probability that the node representing the origin of asynchronous muscle (node 200) is synchronous (Supplementary Table 5, 'Ancestral state posterior probabilities per node'). Next, we constrained that same ancient node (200) to be 100% synchronous and recorded the probability of synchrony in the next most recent node (218) in the branching path towards the origin of Lepidoptera. We iteratively continued this process, constraining each node between the origin of asynchronous muscle and the origin of the Lepidoptera clade (Node 255). We then multiplied through the probability that each node is synchronous to calculate the total probability that the Lepidoptera clade evolved from a synchronous ancestor (Fig. 1d).

## Muscle physiology

**Animals.** *M. sexta* were obtained as pupae from a colony maintained at the University of Washington. Moths were kept on a 12 h:12 h light dark cycle. We used 6 female and 3 male adults, all 2–6 days post eclosion, with a mean body mass of $2.32 \pm 0.46$ g.

**Experimental preparation.** After a 30 min cold anaesthesia, we removed the head, wings, abdomen, legs and first thoracic segment from each moth to isolate the second and third thoracic segments. We then used digital callipers to measure the DLM length as the distance between the anterior and posterior phragma. These structures are the physiological attachment points of the DLMs. We measured a mean muscle length of $11.7 \pm 0.5$ mm.

We used a similar experimental paradigm as Tu and Daniel[55] for dynamic, whole muscle experiments on *M. sexta* DLMs. The key difference in our protocol is that we used a dual-mode ergometer (305 C Muscle Lever, Aurora Scientific) capable of prescribing a length trajectory while measuring the force necessary to follow that trajectory.

For the anterior muscle attachment, we used cyanoacrylate glue to rigidly mount the anterior phragma to a custom three-dimensional printed ABS shaft, which was secured to our experimental table. For the posterior attachment, we attached a pair of tungsten prongs to the ergometer lever. We inserted these prongs at the invagination between the second and third thoracic segments. This ensures that the prongs adhere to the posterior face of the posterior phragma. Cyanoacrylate glue ensured a strong connection. In all preparations, we ensured that the anterior and posterior attachments were rigidly bonded following the experiment.

At this point, the intact second and third thoracic segments were rigidly mounted on our ergometer. We relieved any force buildup during this procedure by manually adjusting the ergometer length until the force reading was at zero. We then shortened the muscle by 2% because the in vivo muscle length during flight is 2% shorter than its length during rest[41].

We next decoupled the anterior and posterior sections of the muscle by removing a transverse ring of exoskeleton. To minimize inertial loads on the ergometer, we removed all other muscles and any remaining exoskeleton on the anterior side of the thorax. We made sure to excise the ganglia to prevent spontaneous muscle activation.

To activate the muscles, we inserted two tungsten electrodes into the anterior end of the muscle by piercing the exoskeleton of the anterior phragma. We repeated this procedure for the posterior side by piercing the posterior end of the scutum. To ensure muscle viability, we maintained a steady drip of saline and held a constant temperature of 35 °C measured at the muscle.

Because $Ca^{2+}$ is required for delayed stretch activation[2], we stimulated the muscles at 150 Hz to induce tetanus. We found that 150 Hz stimulation was the minimum stimulation frequency to establish a fused tetanus. Our experiments to measure delayed stretch activation consisted of a stretch–hold–release–hold cycle. We first maintained zero strain for 150 ms to enable a plateau in force, indicating constant activation (Fig. 2b), we stretched the muscle to in vivo strains (4.5%) at peak in vivo strain rates[41] while measuring muscle force output. We calculated peak in vivo strain rate as $\dot{\varepsilon}_0 = 2\pi f \varepsilon_0$, where $f$ is the wingbeat frequency (25 Hz) and $\varepsilon_0$ is peak in vivo strain. We then held the strain constant for 150 ms. We returned the muscle to rest length at the same strain rate as before.

We measured the peak twitch, delayed stretch activation and tetanic force produced by the muscle. In all cases, we normalized force production by the cross-sectional area of the muscle. To determine the rate constants of delayed stretch activation we fit the muscle force data with the equation

$$F_{\text{step}}(t) = K_2 e^{-r2t} + K_3(1 - e^{-r3t}) + K_4 e^{-r4t} + c \tag{5}$$

where $K_i$ and $r_i$ are the coefficients and rate constants associated with a particular phase of the delayed stretch activation response: a fast decay ($r_2 \gg$ wingbeat frequency), a slower rise ($r_3 \approx$ wingbeat frequency), and a very slow decay ($r_4 \ll$ wingbeat frequency). The constant $c$ represents the passive stiffness of the muscle. Phase 1 is the immediate material response of any muscle to a transient strain and is not relevant to characterizing delayed stretch activation. We quantified delayed stretch activation force magnitude as the difference between the lowest force immediately following stretch to the peak force during the plateau phase.

Following experiments, we removed and weighed the DLMs. We measured an average muscle mass of $0.123 \pm 0.012$ g. This corresponds to a body mass-normalized muscle mass of $5.28 \pm 1.41\%$, which is in rough agreement with prior measurements[55] of $5.96 \pm 0.62\%$. From muscle length and muscle mass, we calculated a cross-sectional area of $10.7 \pm 2$ mm² under the assumption that muscle density is 1 g cm⁻³.

All conditions were replicated on all individual preparations. Twitches were always done first to confirm that stimulation of muscle preparations produced force and that the muscle was viable.

No further randomization was necessary because all data were collected from a single continuous ramp-hold experiment. Experiments were not blinded because there were not multiple conditions.

**Accounting for non-ideal strain rate.** Our input stimulus to the ergometer for stretch–hold–release–hold experiments was a ramp with a speed matching the in vivo strain rate of muscle contraction in a hawkmoth. Our modelling assumes that $r_3$ is the rate of tension rise in response to an infinite impulse, which is not possible to implement in any real physical system. To examine the discrepancy between muscle's response to an infinite impulse and a non-ideal finite impulse, we follow the following procedure. First, we construct a rectangular pulse with a width and height that match the width and height of the actual strain rate pulse we imposed in experiment. We then compute the empirical transfer function between the sum-of-exponentials fit to our force data and the rectangular pulse. This transfer function represents the response of hawkmoth muscle to an infinite impulse in strain rate, and is equal to our fit multiplied by a scalar. We compute this scalar to be 2.29, by dividing the IIR by our fit. We then scale our force data by this constant factor and plot it, labelled as 'IIR' in Fig. 2e. Data presented in Fig. 2b–d are raw and unscaled by the method described here.

### Delayed stretch activation model
To study how delayed stretch activation produces wing oscillations we needed to generate a feedback model for delayed stretch activation. No current detailed muscle model can predict both neural and stretch-activated force components under general dynamic conditions, in part because of limitations in our understanding of the multiscale interactions in muscle[71]. Thus, to model asynchronous force, we do not try to build a detailed molecular model that can predict force from arbitrary activation and strains. Instead, we seek to capture the basic functional input–output relations for delayed stretch activation between an imposed strain and the resulting force. We first constructed a reduced order model of the delayed stretch activation that was able to capture the stretch–hold–release–hold behaviour we observed in experiment. This single-parameter model is described by the 'time to peak' ($t_0$) of the delayed stretch activation force response, which allows us to study how the relative timescales of delayed stretch activation ($t_0$), body mechanics (the natural resonance period, $T_n$), and the synchronous timescale $\left(\frac{1}{f_s}\right)$ govern the emergent wingbeats. To implement delayed stretch activation in simulation, we then generated a computational representation of delayed stretch activation and coupled it to a computational representation of body mechanics.

**One-parameter model of asynchrony.** Measurements of delayed stretch activation in the literature[72] consist of imposing a step change in muscle length and fitting the force response, $F_{step}(t)$, with a sum of three exponentials given by equation (5). This seven-parameter model was used to fit the delayed stretch activation response in the hawkmoth muscle (Fig. 2c). However, the initial viscoelastic drop following lengthening is unlikely to be important for generating self-excited oscillations, and the symmetry between $r_2$ and $r_4$ leaves those parameters sensitive to initial conditions of a curve fit procedure. Because the delay between stretch and force production is likely the critical feature of delayed stretch activation, we only considered the delayed tension rise (determined by $r_3$) and the delayed tension drop (determined by $r_4$). In doing so, we eliminated $K_2$ and $r_2$ from the fit equation (5). We further eliminated the passive muscle stiffness from equation (3) because it does not contribute significantly to the elastic response of the thorax in *M. sexta*[38]. It is possible that active muscle stiffness contributes to body mechanics. Incorporating estimates of active muscle elasticity and body dissipation that may be present in the thorax does not affect the overall conclusions or features of the simulation (see Supplementary

Discussion D). We fit the hawkmoth muscle data to the reduced convolution kernel

$$g(t) = \frac{1}{g_0}(-e^{-r_3 t} + e^{-r_4 t}) \tag{6}$$

where $g_0$ is a scalar that normalizes by the area under the kernel (which depends on the kernel rate constants and has units of seconds). We sought to further reduce this convolution kernel to be parameterized by a single variable, the time to reach peak tension from a step input ($t_0$; Fig. 1b). We first assumed a constant ratio between $r_3$ and $r_4$ such that $r_4 = \kappa r_3$ (for *M. sexta*, $\kappa = 0.62$). We can then solve for $t_0$, from the reduced kernel to obtain

$$t_0 = \frac{\ln\left(\frac{r_3}{r_4}\right)}{r_3 - r_4} \tag{7}$$

$$= \frac{\ln(\kappa)}{r_3(\kappa - 1)} \tag{8}$$

For hawkmoth muscle this yields a relationship between $t_0$ and $r_3$ of

$$t_0 = \frac{1.258}{r_3} \tag{9}$$

The two-parameter fit model yields an $r_3$ of $36.39 \pm 0.09\ \text{s}^{-1}$ and $r_4$ of $22.80 \pm 0.04\ \text{s}^{-1}$, a corresponding $t_0 = 0.034 \pm 0.001\ \text{s}$, and this two-parameter fit matches the experimental data well (Extended Data Fig. 4a). While our single-parameter model is simplified from the classic seven-parameter delayed stretch activation model, the qualitative features of the Fig. 3 heat maps and associated conclusions are insensitive to the precise value of $\kappa$.

**Discrete FIR filter implementation.** To implement delayed stretch activation in our simulation and robot experiments, we require a model of delayed stretch activation that can produce stretch dependent forces in real-time which can provide a delayed stretch activation force to the simulation or robotics experiments. We describe the delayed stretch activation response of asynchronous muscle as a convolution of the muscle strain velocity with a kernel $g$ such that

$$F_{async}(t) = \mu F_a(-g * \dot{\varepsilon})(t) \tag{10}$$

where $F_a$ is the asynchronous forcing magnitude and dictates the strength of the delayed stretch activation feedback taken from the quasi-static experiments, and $\mu$ is a fitting parameter that scales the stretch activation response to flight conditions.

We implement this convolution in simulation and in the robotic models in MATLAB simulink (Mathworks) using a finite impulse response (FIR) filter, which is an instantaneous (real time at 10 kHz) evaluation of a convolution operation. We construct the filter such that the input is the muscle strain rate, and the output is the delayed stretch activation force. We convert from the angular rotational units of our wing to actuator strain through the equation $\dot{\phi} = LT\dot{\varepsilon}$, where we divide wing velocity ($\dot{\phi}$) by a factor $LT$ where $L$ is the resting muscle or actuator length (in the robot models, $L = T = 1$ because all scaling can be captured in $\mu$). This yields the following delayed stretch-activated force

$$F_{async}(t) = \mu \frac{F_a}{LT}(-g * \dot{\phi})(t) \tag{11}$$

The value of $\mu$ is tuned to each system (simulation, roboflapper and robobee wing; see below for details), but is not changed

between experiments where the relative magnitudes of synchronous and asynchronous forcing are varied (that is, when $K_r$ is varied as in Fig. 3).

For simulations and experiment the delayed stretch activation force is generated from an FIR filter which requires a numerical evaluation of the convolution $(-g * \dot{\phi})(t)$. First, we generate the response curve $g(t)$ based on the delayed stretch activation parameters ($r_3$, $r_4$) in MATLAB (Mathworks), sampled at the system rate $\Delta t$ over the simulation/experiment duration (simulation, roboflapper and robobee wing; see below for details). The normalization parameter $g_0$ in equation (6) is the numerical area under the curve. We then find the value of $t$ for which $g(t) < 0.01 \times \max(g(t))$ and truncate the response since the finite impulse response filter requires a finite kernel. Last, we multiply the output of the numerical convolution by the coefficient $\frac{\mu F_a}{LT}$. The delayed stretch activation step response is thus represented as a vector of numbers that are supplied to the 'filter coefficients' input of the FIR block in Simulink. When synchronous and asynchronous forces are applied together we scale $F_{async}$ by $(1 - K_r)$ according to the combined forcing equation (equation (1)).

## Hawkmoth simulation

**Hawkmoth body mechanics model.** We used a dynamics model of the wing rotation in the stroke plane ($\phi$) for the hawkmoth which has previously been derived by Gau et al.[38]. In brief, we assumed a body mechanics model with aerodynamic drag whose magnitude depends on angular velocity squared ($|\dot{\phi}|\dot{\phi}$) with a constant drag coefficient ($\Gamma$), a parallel-elastic spring due to thorax elasticity ($k$), and rotational inertia from wing and added mass ($I$). These assumptions yield the equations of motion presented in equation (3). To generate equivalent torques from the linear muscle force and the linear thorax elasticity we require the transmission ratio between linear muscle displacement and the rotational wing movement. We assumed a linear transmission ratio such that $T = \phi/X$, where $\phi$ is the wing rotation and $X$ is the linear displacement of the muscle and thorax. We calculated the transmission ratio for *M. sexta* as $T = \phi_0/X_0$, where $\phi_0$ is the peak-to-peak wingstroke amplitude and $X_0$ is the peak-to-peak muscle displacement amplitude (values can be found in Supplementary Table 1 and ref. 42). The equivalent torque about the wing hinge produced by the muscle force $F_m$ is given by $F_m/T$. The equivalent elastic torque from the thorax linear stiffness is calculated as $k/T^2$. These two transformations can be derived using conservation of energy: work done at the rotational joint must equal work done on the linear elements (spring and muscle).

The wing inertia, $I$, includes the added mass effects from aerodynamics. The parameter values can be found in Supplementary Table 1. Following the derivations of Ellington[73], wing inertia ($I$) is the sum of inertia due to wing mass ($I_w$) and added mass ($I_a$),

$$I_w = R_2^2(m) m_w L_w^2 \text{ and } I_a = R_2^2(v) v L_w^2 \tag{12}$$

where $R_2(m)$ is the radius of the second moment of wing mass, $R_2(v)$ is the radius of the second moment of wing added mass, and $L_w$ is the wing length. Note that in the aerodynamics literature, the second moments are often denoted $r_2$ and the wing length denoted $R$, but we change the convention here to avoid confusion with the rate constants $r$, used in the delayed stretch activation experiments. Dimensional added mass ($v$) is defined as $v = \frac{2\rho\pi\hat{v}L_w^2}{(AR)^2}$, where $\hat{v}$ is the non-dimensional added mass of the wing pair and AR is the aspect ratio of the wings. Parameter values are in Supplementary Table 1.

The lumped aerodynamic parameter $\Gamma$ was calculated by following the work of Whitney and Wood[74]. The quasi-steady aerodynamic drag force ($F_{aero}$) on insect wings over a single wingstroke can be modelled as

$$F_{aero} = \frac{1}{2}\rho\widetilde{C}_D A_w R_2^2(s) L_w^2 |\dot{\phi}|\dot{\phi} \tag{13}$$

where $R_2(s)$ is the radius of the second moment of wing shape. Setting the drag torque $\tau_{aero} = F_{aero}l_{cp}$ and $\tau_{aero} = \Gamma|\dot{\phi}|\dot{\phi}$, where $l_{cp}$ is the centre of pressure[75], yields the velocity-squared aerodynamic damping coefficient ($\Gamma$) as

$$\Gamma = \frac{1}{2}\rho\widetilde{C}_D A_w R_2^2(s) L_w^2 l_{cp} \tag{14}$$

**Simulation details.** We used MATLAB and Simulink (Mathworks) to run simulations of combined synchronous and asynchronous forcing on a mechanical model of the hawkmoth. Extended Data Fig. 4b,c presents a representation of the Simulink model. The system dynamics block implements the equation of motion (equation (3)) using hawkmoth parameters. It takes the combined muscle forcing as an input and generates the wing angle and angular velocity as outputs (Extended Data Fig. 4b). The wing rotational velocity is then an input into the delayed stretch activation simulation described in the previous section and calculated by equation (11).

As insect flight is driven by pairs of antagonistic muscles, we represent the upstroke and downstroke muscles separately in our Simulink simulation. The antagonistic configuration means that the sign on both strain velocity and output force is different for each muscle, as shown in Extended Data Fig. 4c. Additionally, a sine wave generator is used to produce synchronous forcing based on the amplitude $F_s$ and frequency $f_s$. The output force is a weighted sum of synchronous and asynchronous forces defined by $K_r$.

Both the synchronous and asynchronous forces in the muscle block are saturated so that they only output tension forces. Additionally, the sine wave generators are operated with different initial phase $\theta_0 = 0$ for the upstroke muscle and $\theta_0 = \pi$ for the downstroke muscle. The overall effect is that all of the negative torque is produced by the upstroke muscle, and all of the positive torque is produced by the downstroke muscle. The phase shift in the sine wave generator blocks also enforces that the fully synchronous output is identical to a single sinusoidal torque source.

**Iterative force tuning procedure.** The parameter $K_r$ describes the relative amounts of synchronous and delayed stretch-activated forcing. To study how an insect that is actuated purely through delayed stretch activation ($K_r = 0$) can transition to being purely actuated through synchronous forcing ($K_r = 1$) we need to establish values of $\mu F_a$ and $F_s$ that produce feasible wingbeat motions in both of these regimes. In the hawkmoth simulation we determined that a sinusoidal forcing amplitude of $F_s = 2,720$ mN generates wingbeat kinematics that match in vivo observation of 117 degrees peak-to-peak. This value was previously used to synchronously drive an identical simulation to physiological wingbeat amplitudes[42].

However, the wingbeat kinematics in the purely asynchronous regime ($K_r = 0$) are emergent and thus we need to determine an appropriate $F_{async}$ that can drive our insect model to appropriate wingbeat kinematics. We used a simple iterative force tuning procedure to determine the value of $\mu$ such that asynchronous actuation ($K_r = 0$) can produce wingbeats with peak-to-peak amplitude of $\phi_0 = 117°$. We slowly increment the value of $\mu$ until the output steady-state wingbeat amplitude is within 1% of the desired amplitude of $\phi_0$. In this way we ensure that the both synchronous ($K_r = 1$) and asynchronous ($K_r = 0$) actuation can produce the same wingbeat amplitude.

**Calculation of $K_r$ for *M. sexta*.** Direct computation of $K_r$ for *M. sexta* is challenging since realistic measures of muscle force (for example, from work loop experiments) will contain a mixture of synchronous and asynchronous effects, which are unlikely to be distinguishable under flight conditions. To estimate $K_r$ from physiological measurements, we approximate the relative contributions of synchronous and asynchronous force using quasi-static measurements of neurogenic

force and stretch activation. We use the maximum stretch-activated force above baseline from our stretch–hold–release–hold experiments, $F_a$, as a static representation of the asynchronous muscle force when the amplitude and rate of the stretch is equivalent to those during flight. The tetanic muscle force $F_{tet}$ is a static representation of the maximum neurogenic (synchronous) muscle force at operating length. Both $F_a$ and $F_{tet}$ are generated under maximum activation, and $F_{tet}$ is measured isometrically, so it will not include any stretch-activated force. We can then approximate $K_r$ as $\widetilde{K_r}$, which is the proportion of neurogenic force ($F_{tet}$) to total force ($F_a + F_{tet}$). This gives the ratio in equation (2), which is equal to 1 when there is no stretch activation and approaches zero when the stretch activation far exceeds neurogenic force.

This approximation is based on equating the proportion of synchronous and asynchronous contributions during flight conditions to those measured at maximum activation in quasi-static conditions. While the absolute value of synchronous and asynchronous forces are likely to be very different in flight and quasi-static conditions, their relative importance is likely to be more comparable. For example, if $\widetilde{K_r}$ is related biologically to the proportion of troponin isoforms that are calcium-activated, then this proportion would likely affect quasi-static and flight conditions similarly. Nonetheless, $\widetilde{K_r}$ is still an approximation and we provide it as a way of estimating this parameter from currently available experiments. This approach also enables future experiments to characterize the relative contributions of synchronous and asynchronous force magnitudes using standardized experimental methods. The simulations and models here do not depend on *M. sexta* having a particular value of $K_r$. Future work would benefit from trying to parse the physiological contributions of synchronous and asynchronous forcing in flight conditions and resolving their specific molecular correlates.

**Simulation parameter sweep and analysis.** To evaluate how the presence of both synchronous and delayed stretch activation in an insect muscle influences the wing kinematics we performed simulations varying both $K_r$ and the time to reach peak tension of the delayed stretch activation, $t_0$. We incorporated mechanical timescales of the system by dividing $t_0$ by the natural period to yield the parameter $t_0/T_n$. For a given insect, $T_n$ is assumed to be constant. To sweep across delayed stretch activation timescales, we adjusted $r_3$ via equation (9) to sweep over a range of $t_0/T_n$ values from 0.01 to 1. We varied $K_r$ from 0 to 1. For each set of $K_r$ and $t_0/T_n$ values, we first calculated $r_3$ from $t_0$ (equation (9)). We then generated the delayed stretch activation kernel as described in the delayed stretch activation model section. With $\mu F_a$ from the section above, we could now combine synchronous and asynchronous forcing (equation (1)). We initialized the wing position at 0.1 rad to initiate oscillations when there was no synchronous forcing. All simulations were performed with a fixed sample time of $\Delta t = 1 \times 10^{-4}$ s over a duration of 5 s.

For each set of parameter values, we recorded the emergent force $F_m$, wing position, and wing velocity. We determined the emergent oscillation frequency by taking the Fourier transform of the last 2.5 s of position and identifying the frequency with the largest magnitude. To calculate power, we extracted five periods of oscillation after the system reached steady state. We then numerically integrated force over position and divided by the time elapsed. Lastly, we computed the variation in the peak-to-peak wing amplitude by using the findpeaks command in Matlab to locate all of the wingbeat peaks. The amplitude variation is calculated as the standard deviation of the peak-to-peak wingbeat angles.

**Robophysical experiment**
**Robot details.** Experiments were performed on a dynamically scaled robophysical model described previously in Lynch et. al.[39]. The device consists of a silicone torsion spring with known, linear characteristics[39]; a brushless DC motor (ODrive Robotics, D6374) under closed-loop torque control; and a rigid, fixed-pitch acrylic wing submerged in a tank of water (Extended Data Fig. 4). The wing span and chord ($10 \times 3.6 \times 0.5$ cm) were selected such that the wing, flapping in water with an amplitude between 10° and 60° and frequency between 1 and 4 Hz, has a Reynolds number between $10^3$ and $10^4$, which is approximately the same range as *M. sexta*[23]. Friction is minimized via a set of radial air bearings and a thrust ball-bearing. We measured the spring stiffness and system inertia and calculated the wing drag torque coefficient (Supplementary Table 2)[39]. We also calculated the natural period $T_n$ of the robophysical system using equation (4) as $T_n = 0.416$ s.

The robophysical experiment was designed to mimic the hawkmoth simulation, replacing the virtual hawkmoth dynamics with those of a real system. We tracked wing angle using an optical encoder (US Digital, 4096 CPR) fixed to the wing shaft and a DAQ (National Instruments, PCIe 6323) sampled at 1 kHz. The encoder angle was used as input to a Simulink Desktop Real-Time (Mathworks) model running an identical combined forcing model as described previously in the sections on the delayed stretch activation model, and the hawkmoth simulations. The velocity was calculated by taking a derivative of the encoder position and fed into the model. The model prescribed a motor torque which was sent via USB serial connection to an open source motor controller (ODrive v3.6) that converted it to a current command to the brushless DC motor. The control loop for sensing wing position and sending torque commands to the motor ran at a sample time $\Delta t = 1 \times 10^{-3}$ s.

**Experiment details.** To study how the robophysical system transitions between delayed asynchronous and synchronous forcing modes (Fig. 3d–f) we varied $K_r$ and $t_0/T_n$ in experiments. The robophysical experiments used approximately the same range of actuation parameters as the simulation: $K_r$ spanning 0 to 1 and $t_0/T_n$ from 0.02 to 1. The synchronous gain was set manually so that oscillations did not trigger the overload-current safety features of the motor driver, and the asynchronous gain was set using the same iterative force tuning procedure described above. We ran experiments for 30 s and we measured output power and frequency over the last 15 s of the experiment.

In a separate set of experiments, we studied the frequency entrainment properties of the robophysical system under combinations of both synchronous and asynchronous forcing. We first determined a value of $\mu F_a$ in experiment that yielded high-amplitude asynchronous oscillations (106 ± 3° peak-to-peak) at 3.2 Hz. Next, we performed experiments with a constant $\mu F_a$, but with varied synchronous frequency $f_s = [0.815, 6.515]$ Hz at three levels of forcing magnitude, $F_s^* = [0.1, 0.2, 0.3]F_s$, with respect to the purely synchronous forcing magnitude of $F_s$. We then measured the output wingbeat angle and computed: (1) the emergent frequency using the peak frequency of the Fourier transform; and (2) the peak-to-peak variation in wingbeat amplitude. The results of this experiment are shown in Extended Data Fig. 8.

**Robobee experiment**
We fabricated a single-winged version of a 'dual-actuator' Harvard Robobee, following the smart composite microstructure (SCM) process pioneered by the Harvard Microrobotics Lab[11,12]. Wing parameter values are provided in Supplementary Table 3. The carbon fibre airframe, which holds the piezoelectric bending actuator and SCM transmission, was fixed to an acrylic mount on a manual translation stage to enable displacement sensor calibration.

To implement the delayed stretch activation model, it is necessary to estimate wing velocity in real time. We achieved this via a fibre-optic displacement sensor (D21, Philtec) pointed at a small piece of reflective tape glued to the bending actuator. The sensor is able to measure actuator displacement at which are fed into a Simulink model that converts sensor voltage to displacement through a calibration curve, and then takes a numerical derivative to calculate wing rotational velocity. Wing rotational velocity is then supplied to an identical Simulink model as in the hawkmoth simulations and roboflapper

experiments described above. The simulation of delayed stretch activation force was converted into an amplified voltage signal (0 V < $V_{sig}$ < 200 V) and sent to the piezoelectric actuator resulting in wing oscillations. The control loop for sensing wing position and sending torque commands to the motor ran at a sample time $\Delta t = 1 \times 10^{-4}$ s. The asynchronous gain was chosen such that flapping angles were large but the actuator did not saturate, and the synchronous gain was set using the iterative force tuning procedure above. The Robobee flapping amplitudes did not exceed 50° peak-to-peak.

Observations of the Robobee wing angle were taken via a high-framerate video camera (Phantom VEO-410) at 2,500 frames per second. Video frames were processed in MATLAB to get the wing angle. Flapping amplitude was estimated by finding oscillation peaks, and flapping period or frequency was estimated by computing the time between peaks and smoothing the resulting curve (Extended Data Fig. 9).

### Reporting summary

Further information on research design is available in the Nature Portfolio Reporting Summary linked to this article.

### Data availability

Trait data and posterior probabilities of the phylogenetic reconstructions are located in Supplementary Tables 4 and 5. Raw physiological data for the muscle physiology experiments and data from the robophysical experiments are available at the Georgia Tech SmartTech data repository, under accession code 1853/66777. Means and ranges of the non-lepidopteran comparative data in Fig. 2d were digitized from the source publication[18] using webplotdigitizer[76] (https://automeris.io/WebPlotDigitizer/).

### Code availability

All code associated with simulations, robophysical models, and the robotics platform are available at the following public GitHub repository: https://github.com/agilesystemslab/synch_asynch_sim.

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

**Acknowledgements** This work was supported by US National Science Foundation RAISE grant no. IOS-2100858 to S.S. and N.G. and 1554790 (MPS-PoLS) and a Dunn Family Professorship to S.S. as well as the US National Science Foundation Physics of Living Systems SAVI student research network (GT node grant no. 1205878).

**Author contributions** J.G. performed muscle physiology experiments and simulations. J.L. performed robot experiments. B.A. and J.G. collected all muscle ultrastructure data. B.A. and S.S. performed ancestral state reconstruction. E.W. performed analysis of data and simulations. S.S. and N.G. oversaw all experiments, simulation and data analysis. All authors participated in writing the manuscript.

**Competing interests** The authors declare no competing interests.

**Additional information**
**Correspondence and requests for materials** should be addressed to Nick Gravish or Simon Sponberg.
Nature thanks Michael Engel, David Lentink and the other, anonymous, reviewer(s) for their contribution to the peer review of this work. Peer review reports are available.

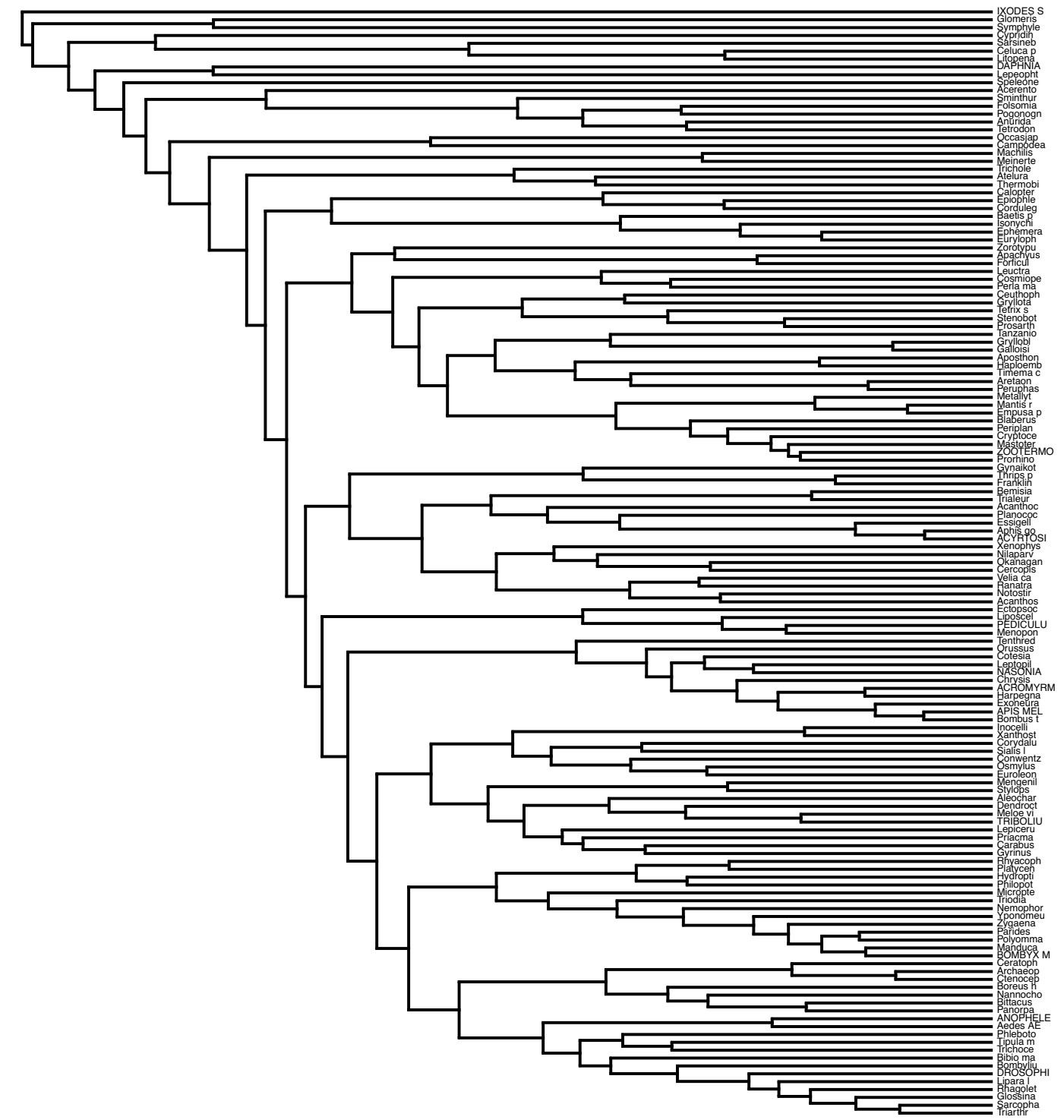

IXODES S
Glomeris
Symphyle
Cypridin
Sarsineb
Celuca p
Litopena
DAPHNIA
Lepeopht
Speleone
Acerento
Sminthur
Folsomia
Pogonogn
Anurida
Tetrodon
Occasjap
Campodea
Machilis
Meinerte
Trichole
Atelura
Thermobi
Calopter
Epiophle
Corduleg
Baetis p
Isonychi
Ephemera
Euryloph
Zorotypu
Apachyus
Forficul
Leuctra
Cosmiope
Perla ma
Ceuthoph
Gryllota
Tefrix s
Stenobot
Prosarth
Tanzanio
Gryllobl
Galloisi
Aposthon
Haploemb
Timema c
Aretaon
Peruphas
Metallyt
Mantis r
Empusa p
Blaberus
Periplan
Cryptoce
Mastoter
ZOOTERMO
Prorhino
Gynaikot
Franklin
Thrips p
Bemisia
Trialeur
Acanthoc
Planococ
Essigell
Aphis go
ACYRTOSI
Xenophys
Nilaparv
Okanagan
Cercopis
Velia ca
Ranatra
Notostir
Acanthos
Ectopsoc
Liposcel
PEDICULU
Menopon
Tenthred
Orussus
Cotesia
Leptopil
NASONIA
Chrysis
ACROMYRM
Harpegna
Exoneura
APIS MEL
Bombus t
Inocelli
Xanthost
Corydalu
Sialis l
Conwentz
Osmylus
Euroleon
Mengenil
Stylops
Aleochar
Dendroct
Meloe vi
TRIBOLIU
Lepiceru
Priacma
Carabus
Gyrinus
Rhyacoph
Platyceh
Hydropti
Philopot
Micropte
Triodia
Nemophor
Yponomeu
Zygaena
Parides
Polyomma
Manduca
BOMBYX M
Ceratoph
Archaeop
Ctenoceph
Boreus h
Nannocho
Bittacus
Panorpa
ANOPHELE
Aedes AE
Phleboto
Tipula m
Trichoce
Bibio ma
Bombyliu
DROSOPHI
Lipara l
Rhagolet
Glossina
Sarcopha
Triarthr

**Extended Data Fig. 1 | The insect wide phylogeny.** The insect wide phylogeny[15] used to conduct the ancestral state reconstruction of insect muscle type. The full name of every tip species can be found in raw data (Supplementary Data Table S4 – Species muscle type with sources). Node number labels are found in Extended Data Fig. 2.

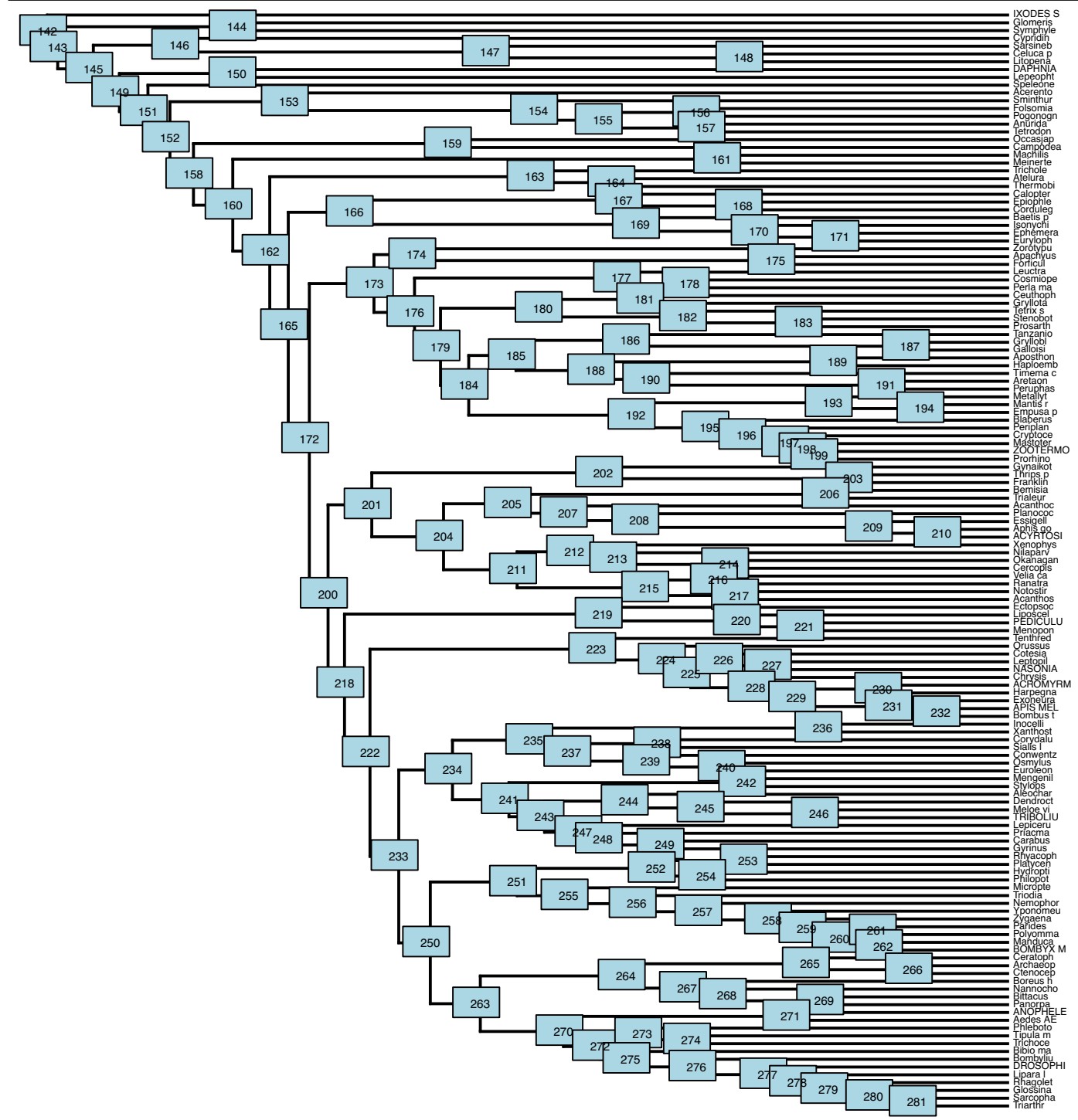

**Extended Data Fig. 2 | The insect wide phylogeny with node number labels.** The number of each node is overlayed on top of the insect wide phylogeny (Fig. 1c, Extended Data Fig. 1). These node numbers are referenced in the table containing the posterior probability of the ancestral state for each ancestral node of the phylogeny (Supplementary Data Table S5 - Ancestral state posterior probabilities per node).

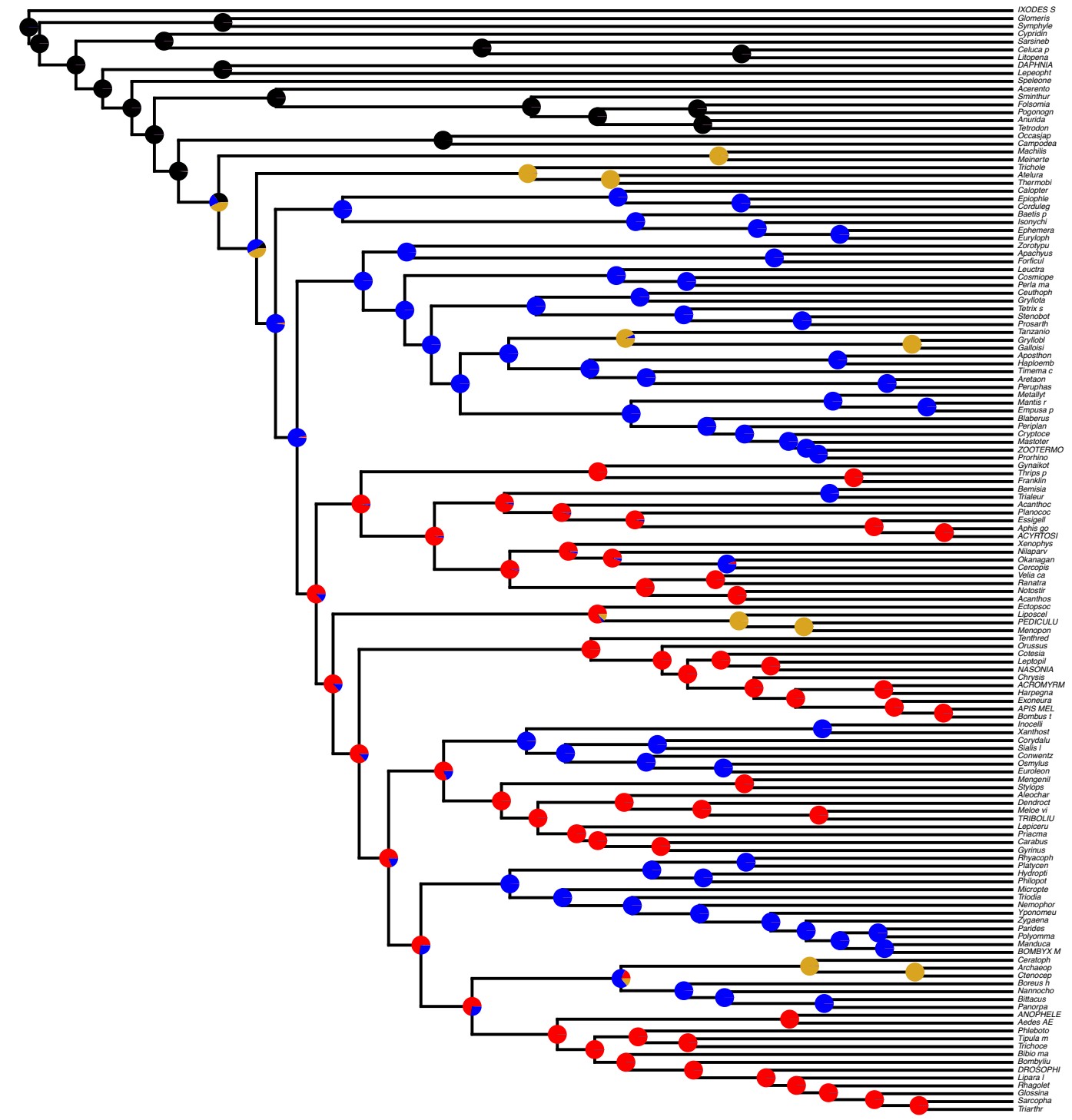

**Extended Data Fig. 3 | The posterior probability of muscle type at each ancestral node.** The pie chart at each ancestral node represents the posterior probability of muscle type under the equal rates ancestral state reconstruction found in Fig. 1c. Values are reported in Supplementary Data Table S2 - Ancestral state posterior probabilities per node. The node number for every node on the phylogeny is referenced in Extended Data Fig. 2. All colors are consistent with Fig. 1 of the main text.

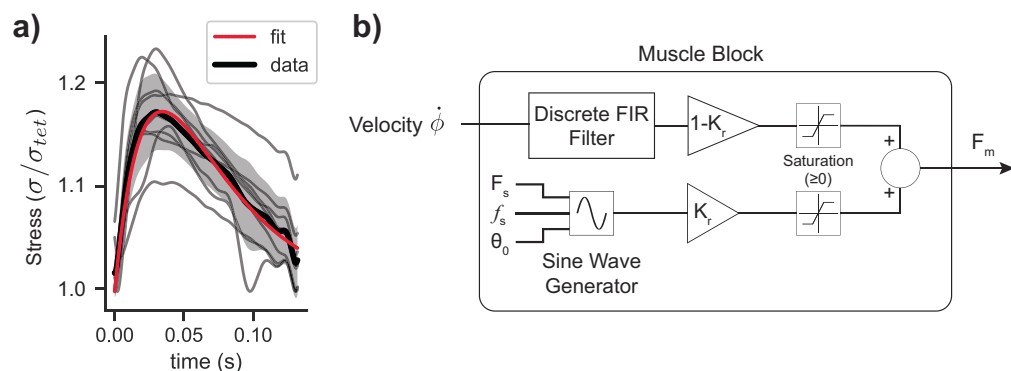

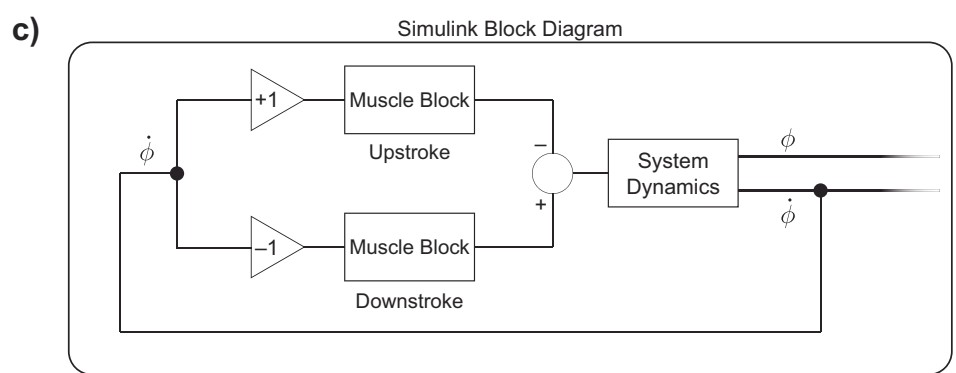

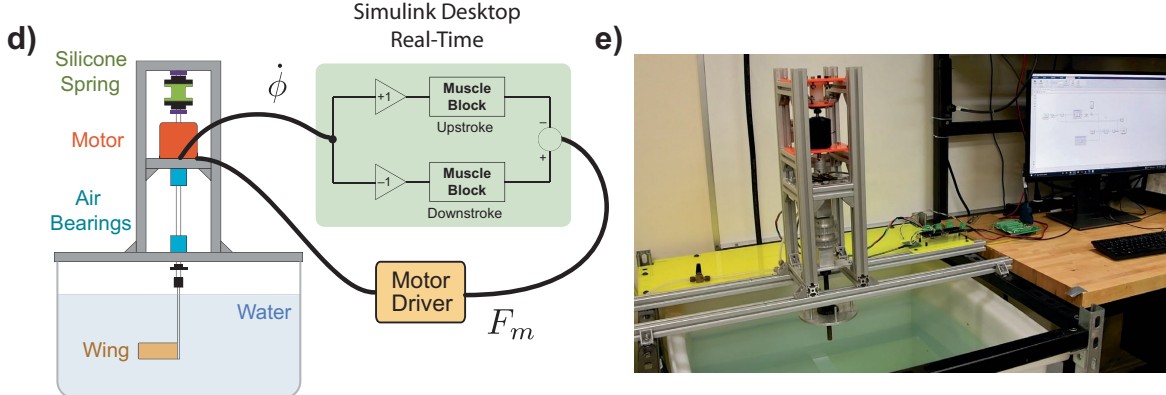

**Extended Data Fig. 4 | Details of simulation and robophysical implementation of delayed stretch activation.** a) Data from Fig. 2c replotted (black line is mean, shaded region is ±1 s.d., grey lines are individual traces. A two parameter model fit ($r_3$ and $r_4$) of phases 3 and 4 of the *M. sexta* delayed stretch activation response (Eq. 6). b) Diagram of the muscle block alone. Output is a weighted sum of asynchronous feedback and synchronous forcing, saturated such that it exerts force only in one direction. The sine generator has a phase, $\theta_0$, of 0 for the upstroke muscle and $\pi$ for the downstroke muscle so that together the two muscles generate sinusoidal forcing. c) Block diagram of simulation of antagonistic muscles under both delayed stretch activation and synchronous forcing. d) Schematic of the robophysical experiment. A dynamically scaled wing is immersed in a large water tank and is actuated by a brushless motor under torque control. An angular encoder measures the wing rotation and the calculated wing velocity is supplied to a Simulink simulation of delayed stretch activation. The combined output of delayed stretch activation ($F_{async}$) and the synchronous force ($F_{sync}$) is provided to the motor driver to actuate the wing. e) A photo of the experimental setup.

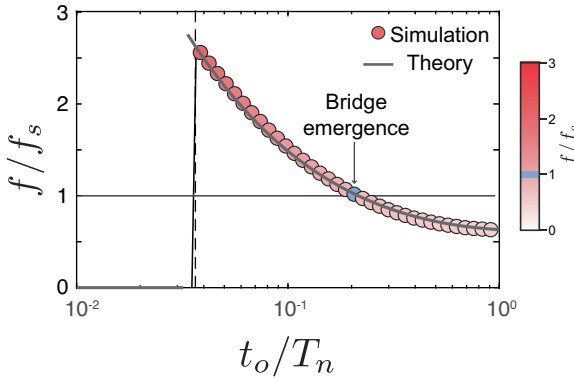

**Extended Data Fig. 5 | Emergent frequency ($f$) normalized by the synchronous frequency ($f_s$) in the asynchronous regime ($K_r = 0$).** As the time to peak force from delayed stretch activation ($t_0$) is increased the system undergoes a bifurcation in which steady wingbeats emerge. Dashed line is the prediction of this critical $t_0/T_n$ from analysis of a linearized system. The emergent wingbeat frequency in simulation (circles) decreases as $t_0$ is increased. A linearized analysis of Eqs. 1 & 3 (See SI for details) is able to capture the emergent wingbeat frequency. The colormap of simulation points matches the heatmap of Fig. 3c, f. The bridge emerges when the emergent frequency from delayed stretch activation matches the synchronous frequency.

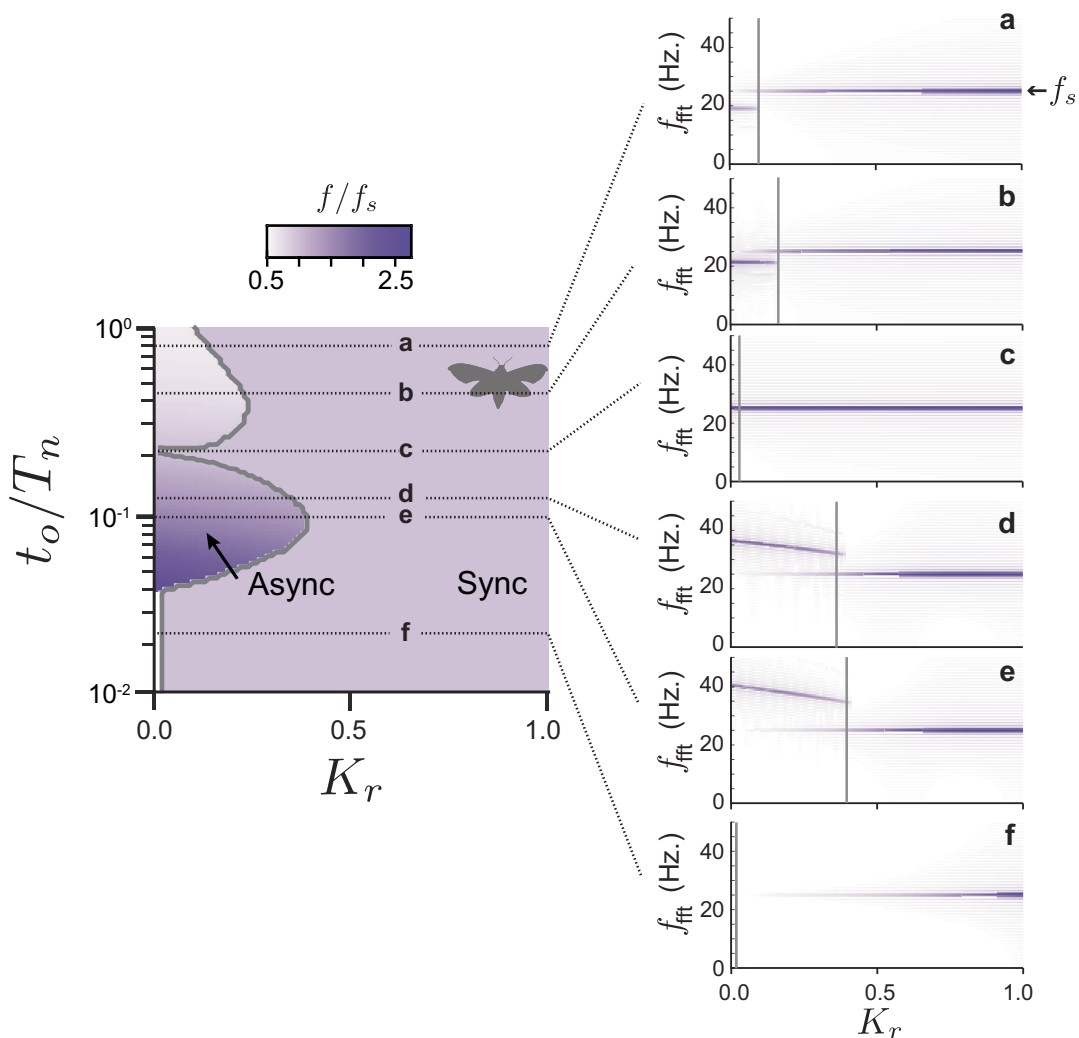

**Extended Data Fig. 6 | Emergent frequency and Fourier transform of wingbeats versus $K_r$ in simulation.** Left plot shows the normalized emergent frequency ($f/f_s$) from Fig. 3a using a continuous colormap. Six horizontal lines correspond to values of $t_0/T_n$ where we examined the Fourier transform of the emergent wingbeats. The six plots in the right column show heatmaps of the Fourier transform of wingbeat at each value of $K_r$. As $t_0$ decreases (from top to bottom), the emergent asynchronous frequency varies, and we see mixing of asynchronous and synchronous dynamics near the boundary between async and sync regions (shown in gray). The bridge between synchronous and asynchronous regimes occurs when the emergent async frequency is exactly equal to the synchronous driving frequency (Plot **c** in right column).

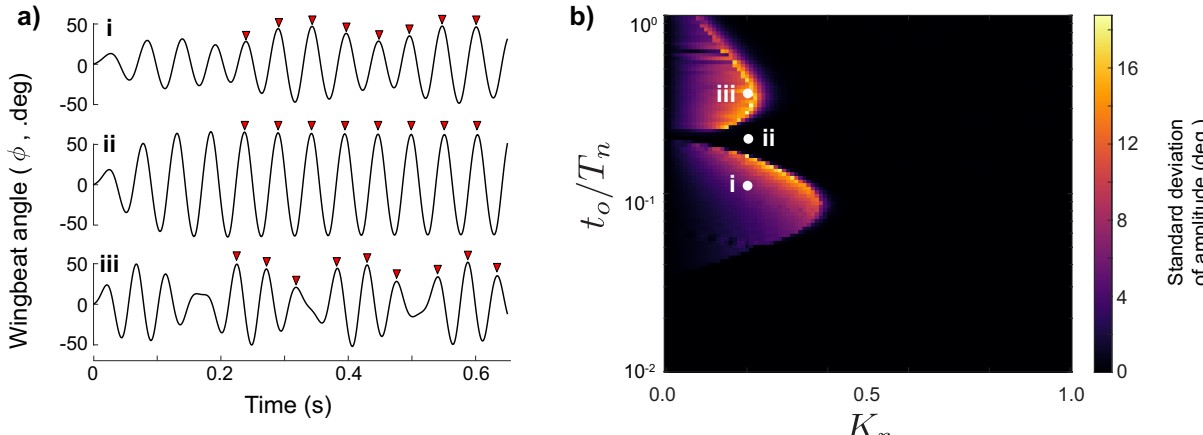

**Extended Data Fig. 7 | Wingbeat amplitude variation in *M. sexta* simulations.**
a) The three plots show the wingbeat angle versus time at three different values
of $t_0/T_n$, for a constant $K_r = 0.2$ (parameters for tests i, ii, and iii are shown in
panel b). After transient oscillations die out we measure the amplitude of
wingbeat peaks as a function of time (positive peaks are shown as red triangles).
The top plot is an example in the asynchronous regime, displaying moderate
amplitude fluctuations. The middle plot is the wing angle within the frequency
locking synchronous regime (on the "bridge") where the wing amplitude is
steady. Lastly, the bottom plot shows the wing motion in the asynchronous

regime below the frequency locking "bridge". b) For all combinations of $K_r$ and
$t_0/T_n$ we calculated the standard deviation of the wingbeat amplitudes which
we show as a heatmap. Brighter regions of the plot correspond to where large
stroke to stroke amplitude variation occurs (i.e. top and bottom plots in panel a).
When the wingbeat is steady the amplitude variation is small these appear as
the black regions. The boundaries between the synchronous and asynchronous
regimes exhibit large amplitude fluctuations, while the bridge connects the
synchronous and asynchronous regimes with smooth sinusoidal emergent
wingbeats.

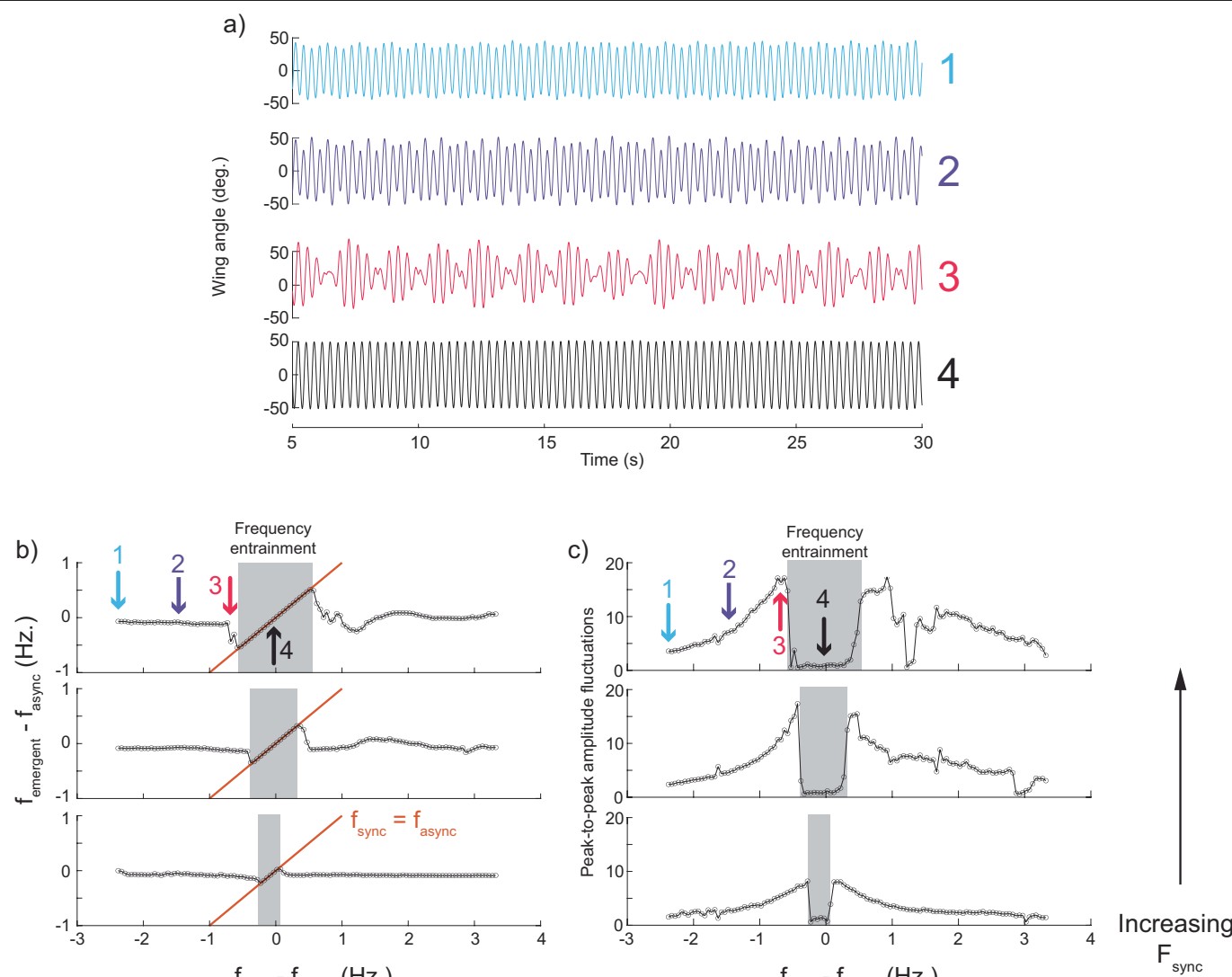

**Extended Data Fig. 8 | Synchronous entrainment of asynchronous oscillations experiment.** a) Emergent wingbeat angle versus time in experiments with a combination of asynchronous and synchronous actuation. The four plots correspond to varying the synchronous drive frequency compared to the emergent asynchronous frequency. The synchronous drive is increased from curves 1 through 4 as shown by the arrows in (b). As the synchronous drive frequency gets closer to the asynchronous frequency the wing motion exhibits large amplitude modulations due to a beat frequency between synchronous and asynchronous oscillations. However, when the synchronous drive is close enough the emergent wingbeat frequency entrains to the synchronous drive and the amplitude fluctuations disappear. b) The emergent wingbeat frequency compared to the driving wingbeat frequency. The gray region indicates frequency entrainment where the emergent frequency is exactly equal to the driving synchronous frequency. The three plots are of increasing synchronous forcing magnitude from bottom to top. c) The amplitude fluctuations increase as the synchronous frequency approaches the asynchronous frequency. However, when the driving frequency cross the Arnold tongue and the emergent frequency becomes entrained to the synchronous frequency, then the amplitude fluctuations disappear.

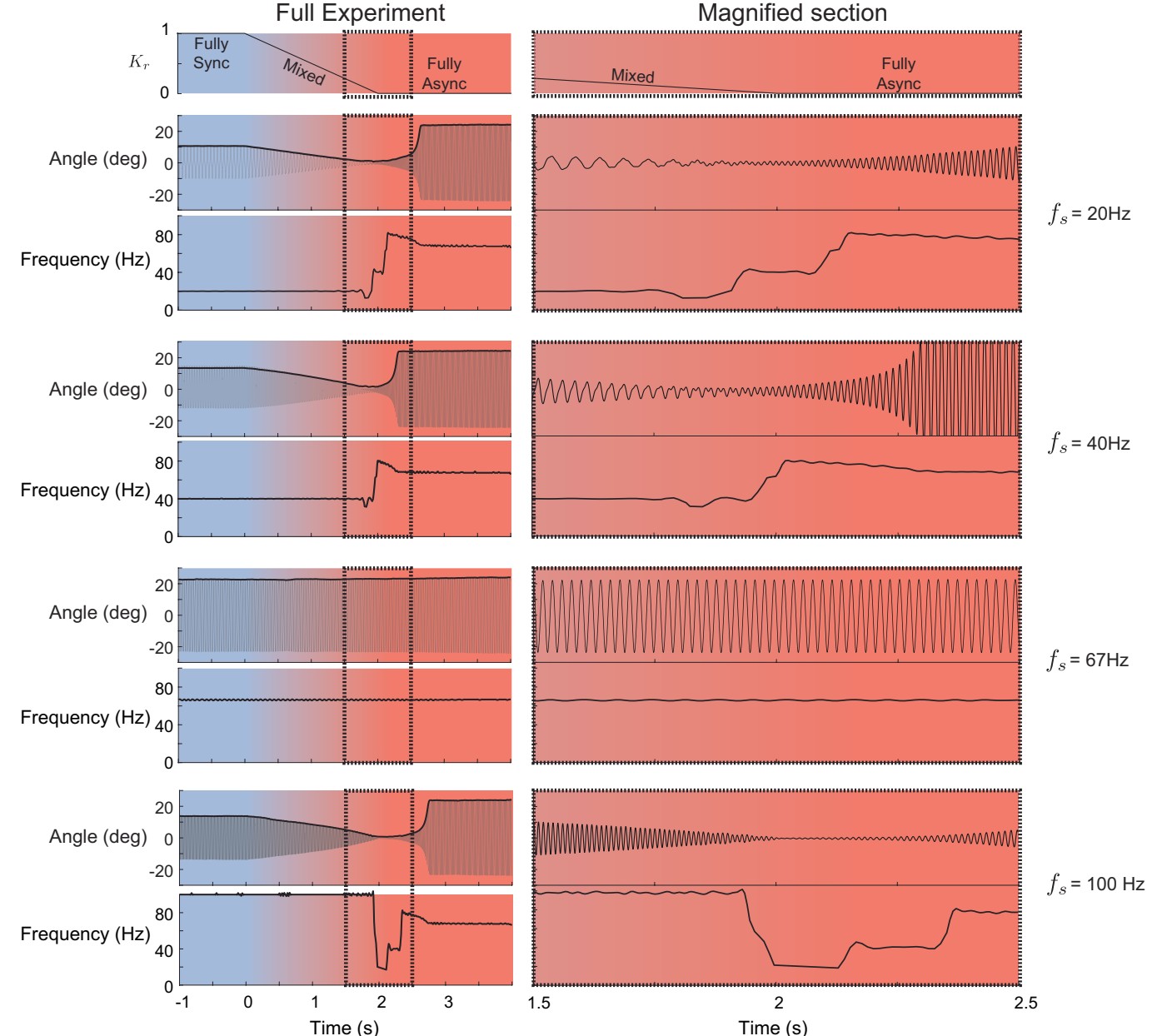

**Extended Data Fig. 9 | Synchronous to asynchronous transitions in the robobee wing.** Four tests at $f_s$ = [20, 40, 67, 100] Hz are shown in which the robobee is transitioned from synchronous to asynchronous forcing. Each experiment consisted of one second of synchronous flapping (blue region) at a particular frequency, followed by a 2 s transition in which $K_r$ was linearly increased from $K_r = 1$ to $K_r = 0$ (top plot) followed by 2 s of 100% asynchronous operation (red region). The wingbeat angle and frequency are plotted for each of the four experiments ($f_s$ is indicated on the right hand side). The left column shows the full time course of the experiments while the right column is a zoomed in region of the onset of fully asynchronous dynamics.

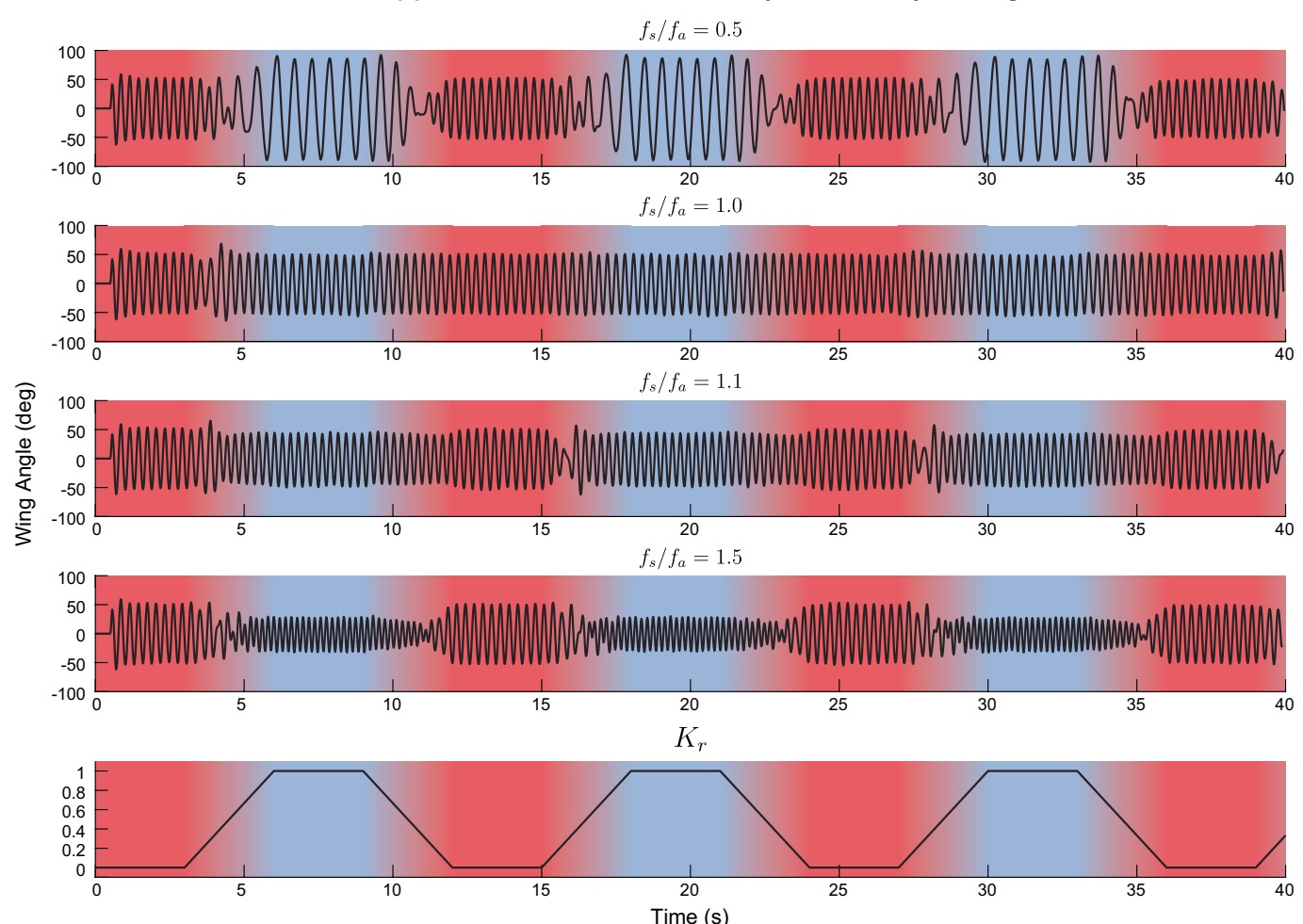

## Roboflapper Transitions Between Sync and Async Regimes

**Extended Data Fig. 10 | Transitions both ways across the synchronous to asynchronous bridge.** In experiments on the dynamically scaled roboflapper, we show that real-time transitions are possible from asynch to synch and synch to asynch. This is an extension of the insect-scale wing experiment from Fig. 3. Smooth transitions occur only when the synchronous and emergent asynchronous frequencies are matched, although there are signs of phase entrainment when the system goes from asynch to synch (second plot, 3–5 s). In all other cases, including when the frequencies are nearly matched but not quite, we see interference as the dynamics transition from one mode and frequency to another.

# Reporting Summary

## Statistics

For all statistical analyses, confirm that the following items are present in the figure legend, table legend, main text, or Methods section.

| n/a | Confirmed | |
|---|---|---|
| ☐ | ☒ | The exact sample size (*n*) for each experimental group/condition, given as a discrete number and unit of measurement |
| ☐ | ☒ | A statement on whether measurements were taken from distinct samples or whether the same sample was measured repeatedly |
| ☒ | ☐ | The statistical test(s) used AND whether they are one- or two-sided<br>*Only common tests should be described solely by name; describe more complex techniques in the Methods section.* |
| ☒ | ☐ | A description of all covariates tested |
| ☒ | ☐ | A description of any assumptions or corrections, such as tests of normality and adjustment for multiple comparisons |
| ☐ | ☒ | A full description of the statistical parameters including central tendency (e.g. means) or other basic estimates (e.g. regression coefficient) AND variation (e.g. standard deviation) or associated estimates of uncertainty (e.g. confidence intervals) |
| ☒ | ☐ | For null hypothesis testing, the test statistic (e.g. *F*, *t*, *r*) with confidence intervals, effect sizes, degrees of freedom and *P* value noted<br>*Give P values as exact values whenever suitable.* |
| ☒ | ☐ | For Bayesian analysis, information on the choice of priors and Markov chain Monte Carlo settings |
| ☒ | ☐ | For hierarchical and complex designs, identification of the appropriate level for tests and full reporting of outcomes |
| ☒ | ☐ | Estimates of effect sizes (e.g. Cohen's *d*, Pearson's *r*), indicating how they were calculated |

*Our web collection on statistics for biologists contains articles on many of the points above.*

## Software and code

Policy information about availability of computer code

| Data collection | Matlab and Simulink (R2020, Mathworks) was used for all data collection as well as robophysical and robot platforms. |
|---|---|
| Data analysis | All phylogenetic analyses were done using R studio (v. 1.1.383) using R (v. 4.0.2), with the PhyTools and  corHMM packages. No special data analysis was required for the physiology data. Plotting was done with Matlab (R2020). All code for implementing the simulations, robophyiscal model and robotic platform are available in the github repository: https://github.com/agilesystemslab/synch_asynch_sim |

For manuscripts utilizing custom algorithms or software that are central to the research but not yet described in published literature, software must be made available to editors and reviewers. We strongly encourage code deposition in a community repository (e.g. GitHub). See the Nature Portfolio guidelines for submitting code & software for further information.

## Data

Policy information about availability of data

All manuscripts must include a data availability statement. This statement should provide the following information, where applicable:
- Accession codes, unique identifiers, or web links for publicly available datasets
- A description of any restrictions on data availability
- For clinical datasets or third party data, please ensure that the statement adheres to our policy

Muscle ultrastructure data that was collected from prior literature is located in supplementary data table 1. Raw physiological data for the muscle experiments are available at the Georgia Tech SmartTech data repository: http://hdl.handle.net/1853/66777

# Research involving human participants, their data, or biological material

Policy information about studies with [human participants or human data](). See also policy information about [sex, gender (identity/presentation), and sexual orientation]() and [race, ethnicity and racism]().

| | |
|---|---|
| Reporting on sex and gender | N/A |
| Reporting on race, ethnicity, or other socially relevant groupings | N/A |
| Population characteristics | N/A |
| Recruitment | N/A |
| Ethics oversight | N/A |

Note that full information on the approval of the study protocol must also be provided in the manuscript.

# Field-specific reporting

Please select the one below that is the best fit for your research. If you are not sure, read the appropriate sections before making your selection.

☒ Life sciences   ☐ Behavioural & social sciences   ☐ Ecological, evolutionary & environmental sciences

For a reference copy of the document with all sections, see [nature.com/documents/nr-reporting-summary-flat.pdf](http://nature.com/documents/nr-reporting-summary-flat.pdf)

# Life sciences study design

All studies must disclose on these points even when the disclosure is negative.

| | |
|---|---|
| Sample size | Sample size of 9 independent individuals from the same source colonies were chosen based on prior muscle physiological experiments such as: Tu & Daniel 2004, Journal of Experimental Biology; Wang, et al. 2018, Biophysical Journal; Josephson, 1997, Journal of Experimental Biology |
| Data exclusions | No data were excluded from the muscle physiology experiments. |
| Replication | All experimental protocols were identical and replicated across each of 9 preparations. No further replication was conducted because other colony sources were not available and results were consistent across the individuals analyzed. |
| Randomization | Twitch responses were characterized first in all preparations to determine that stimulation was effective. All other measurements were taken in a continuous ramp & hold trial, so there was no need for randomization. |
| Blinding | The experiments were not blinded because there were not multiple group allocations of data being collected. The researcher also had to monitor the data during experiment to determine if stimulation of the muscle produced a force response. |

# Reporting for specific materials, systems and methods

We require information from authors about some types of materials, experimental systems and methods used in many studies. Here, indicate whether each material, system or method listed is relevant to your study. If you are not sure if a list item applies to your research, read the appropriate section before selecting a response.

## Materials & experimental systems

| n/a | Involved in the study |
|---|---|
| ☒ | ☐ Antibodies |
| ☒ | ☐ Eukaryotic cell lines |
| ☒ | ☐ Palaeontology and archaeology |
| ☐ | ☒ Animals and other organisms |
| ☒ | ☐ Clinical data |
| ☒ | ☐ Dual use research of concern |
| ☒ | ☐ Plants |

## Methods

| n/a | Involved in the study |
|---|---|
| ☒ | ☐ ChIP-seq |
| ☒ | ☐ Flow cytometry |
| ☒ | ☐ MRI-based neuroimaging |

# Animals and other research organisms

Policy information about studies involving animals; ARRIVE guidelines recommended for reporting animal research, and Sex and Gender in Research

| | |
|---|---|
| Laboratory animals | Adult Manduca sexta (2-6 days post eclosion) from two colonies (University of Washington and Case Western Reserve Univ.) |
| Wild animals | No wild animals used |
| Reporting on sex | Both sexes were used in this study (6 females and 3 males). |
| Field-collected samples | No Field collected samples were used. |
| Ethics oversight | All animals used were invertebrates and not governed by IACUC guidelines. |

Note that full information on the approval of the study protocol must also be provided in the manuscript.

