## [Peer Review File · Nature]

Manuscript Title: Bridging two insect flight modes in evolution, physiology, and robophysics

Reviewer Comments & Author Rebuttals

Reviewer Reports on the Initial Version:

Referees' comments:

Referee #1 (Remarks to the Author):

This is an exceptionally interesting study that explores the evolution of transitions between synchronous and asynchronous flight systems in insects, and employ some novel robotic models to demonstrate that these seemingly disparate flight neuro-mechanical systems are actually based on the same dynamics and that transitions between the two can be achieved more easily than previously surmised. This conclusion matches with their ancestral state reconstructions, which demonstrate multiple transitions (principally reversions) between these states. It is a unique exploration of one of the key features in the history of insect evolution and the refinements and specialization of insect flight.

In terms of the methods, my only real concern is the undefended position of assuming an equal rates model for exploring the evolution of these traits across insect phylogeny. This seems to be a naive assumption and I encourage the authors to explore this instead through the application of a relaxed clock model with uncorrelated rates in a Bayesian framework. This is likely to be a far more realistic approximation of the probability of given transitions. Or, at the minimum, some defense of the underlying assumptions in applying an equal rates model, most of which would seem to be violated by the system they discuss. Otherwise, aside from this one concern, the validity of their methods and analyses are certainly sound.

The citations presented appropriately credit the necessary sources, and the writing is clear and easily understood by a broad readership, with conclusions that do not overreach from the results of their work.

I encourage publication once the concerns regarding the ancestral state reconstruction are resolved. I look forward to seeing this work in print.

Referee #2 (Remarks to the Author):

This is a very interesting manuscript on an important (but in recent times, neglected) topic, viz. how the two fundamental mechanisms of generating insect wingbeat oscillations in flight have evolved, and how they interact. These two mechanisms are referred to as "synchronous" or "asynchronous", respectively, according to whether neural activation maps in a one-to-one fashion to muscle activity. Asynchronous musculature is typically associated with high power output requirements at high wingbeat frequencies, making its origin a key transition in the evolution of insect flight.

The authors use formal ancestral state reconstruction (ASR) methods to identify the likely number of origins and/or losses of asynchronous flight musculature across taxa. This is an important contribution, because several previous works have drawn or repeated the conclusion that asynchronous flight musculature has evolved multiple times in insects on the informal basis that its occurrence appears scattered across the phylogeny. In contrast, the authors' formal analyses provide strong statistical support for the opposite conclusion, that asynchronous musculature evolved once and has subsequently been lost in several major taxa.

The authors follow their evolutionary analyses with a more detailed analysis of the muscle physiology of several species. This analysis shows that delayed stretch activation (dSA; a necessary property of asynchronous muscles) is present in species identified as secondarily synchronous by their ASR analysis (albeit to a lesser degree than in species classified as asynchronous), but is lacking in species for which synchronous musculature is the ancestral state.

Finally, the authors use a dynamics framework to model how dSA can coexist with synchronous neural activation in secondarily synchronous flight muscles, and by extension to unify our understanding of the dynamics of synchronous and asynchronous musculature. They do so through three parallel analyses involving computer simulation, robophysical modelling, and an insect-scale robotic flapper. This is a useful conceptual contribution, although the main result – that dSA can coexist with synchronous neural activation provided that the frequency of the synchronous activation matches the natural frequency that the dSA generates – is not in itself surprising.

This is a well written paper, but I would like to see the authors address the following comments:

General comments:

1. The results of the dynamics analysis as presented seem largely unsurprising, and I would therefore have liked to see some more detailed discussion of the dynamics they capture in order to maximise the value of the analysis. The authors refer at one point to an “apparent Arnold tongue”, but this jargon is unlikely to provide much dynamical insight for readers not already familiar with what is meant by this description of the appearance of a visualization – even with the additional explanation at Lines 184-186. On the other hand, there is none of the discussion that I think I might have expected to see on what look to be period-doubling bifurcations visible in Fig. 3i, or indeed of the dynamical meaning of the pink areas of Fig. 3c, which represent the region of the parameter space where variation in dSA drives variation in the frequency of asynchronous muscle activity. Surprisingly, the word “bifurcation” appears nowhere in the manuscript at present. This more detailed discussion would make it possible to assess the correctness or otherwise of the authors’ identification of an “Arnold tongue” in Fig. 3c, where their use of the word “apparent” seems to imply hesitancy on their part. Some further discussion of the variations in power output in a system being driven at or near resonance would also have been helpful in relation to the “bridge” that the authors describe, in grounding readers’ understanding of the phenomena that these plots describe.

2. The dichotomous shading of Fig. 3c and f seems highly arbitrary, and risks giving the appearance of bifurcations between the regions of the graph where none may in fact be present. It looks to me as if the boundaries between the blue region and the dark pink region may indeed represent jumps in emergent wingbeat frequency (i.e. bifurcations), but it seems plausible that the boundaries between the blue region and the pale pink represent smooth transitions. Whilst I can see the merit of the panel coloration that is shown, I think it is essential to complement this with a continuous colormap to avoid this kind of issue.

3. The location of the Manduca icon on Figure 3 seems to suggest that it sits far away from the “bridge” that the authors describe, so I can see little evidence in the results that Manduca is exploiting any amplification. This doesn’t necessarily detract from the other findings, but it does raise a question on the evolutionary importance of the resonance “bridge” that the authors have identified in the dynamics. For instance, another way of looking at their results is that Fig. 3c, f demonstrates that it is perfectly straightforward to switch from asynchronous to synchronous dynamics (which is the evolutionary transition that the authors find has occurred multiple times on the phylogeny), simply by decreasing dSA magnitude. It is then an open question whether it would be beneficial to do so by being on the “bridge” (i.e. by exploiting resonance), although the one datapoint shown on the figure may suggest otherwise.

4. In fact, there may be very good reasons to avoid being on the “bridge”. A good reason for using synchronous musculature, for example, is to have control of wingbeat frequency, which will be difficult to effect consistently at frequencies close to the resonance. This may explain why *Manduca* sits far from the “bridge”, but does beg the question of whether the “bridge” matters after all, and even whether the low dSA magnitude in *Manduca* has any functional significance at all. Perhaps, since the dSA magnitude in *Manduca* appears to be low enough that there is neither competition nor beneficial interaction between the synchronous and asynchronous dynamics, this has simply been driven to a point at which it is no longer under selection?

5. Although the first two columns of Fig. 3 explore the parameter space quite thoroughly, albeit for a fixed synchronous forcing frequency, the last column of Fig. 3 presents only two different synchronous forcing frequencies. I appreciate that it might be difficult or impossible to conduct new experiments at this point, but it would, I think, have been informative to have explored the effect of perturbing the synchronous forcing frequency to a small degree around the natural frequency of 67 Hz. This would provide a useful test of the real-world robustness of the findings, and would shed some light on the extent to which the “bridge” that the authors have identified is indeed a bridge rather than a tightrope that evolution must tread. At present, the results are stated in the figure legend as “Transitions are smooth when $f_s = f_a$... but interference causes oscillation amplitudes to drop when $f_s \neq f_a$ ” and similarly at lines 242-245. This needs rewording, because it would be reasonable to expect that small perturbations of f_s from $f_s = f_a$ (i.e. f_s not equal to f_a sensu stricto) will still allow the natural dynamics to be driven at the forcing frequency, which is what the vertical extent of the “bridge” in the other two panels presumably shows. At the least, I would suggest replacing the equality sign with an “approximately equal to” sign to reflect this.

Specific comments:

6. Lines 101-192. “... while maintaining high-power wing strokes and providing a plausible path for the many subsequent shifts between asynchrony and synchrony”. An alternative interpretation of

7. Line 47. The meaning of the percentages here needs to be explained, as it relates to some quite involved details of the ASR analysis. If it is not possible to explain succinctly, then I would suggest removing the percentage figures, and leaving the figure to justify the conclusion that “there has most likely been only one evolution of flight muscle asynchrony at the order level”.

8. Line 48. Consider expanding the sentence “Although asynchrony was thought to have evolved 7-10 times throughout insect flight (2,3,6), only recently has an insect-wide phylogeny enabled interorder resolution” to make clear that the conclusions of Refs 2, 3, 6 are not based on any formal phylogenetic analysis. That is, the reason why the authors conclude the opposite here is because they have done the kind of analysis that was required to draw a conclusion of this kind in the first place; not because the assumed phylogeny has changed from an earlier analysis, or because of some fine difference in method or assumptions.

9. Line 66. the statement “locusts evolved before the first synchronous fliers” would be better stated in terms of ancestral states, especially as “locust” has a specific meaning in relation to swarming that the fossil record does not capture.

10. Line 90. “We found that the dSA rate, r_3 , in *M. sexta* DLMs fits on the asynchronous scaling line for its wingbeat frequency of 25 Hz”. By no stretch of the imagination does the data point shown by the star in Fig. 2 “fit on the ... scaling line”. For one thing, the line drawn on the figure has not even been extrapolated to frequencies below about 30 Hz, and if it had been, then the predicted value of r_3 would have been close to (and perhaps even less than) zero, rather than of order 100 as shown for *Manduca*. I agree that the datapoint is consistent with the broad scaling

relationship shown in the figure, but the data are too variable to describe this as a “scaling line”.

11. Lines 94-96 are key in showing that the magnitude of dSA in hawkmoths is an order of magnitude lower than in asynchronous beetles, waterbugs and flies, because this is what ultimately gives physiological meaning to the parameter K_r in the dynamics analysis. I wonder whether it might be worth making this connection further down when the parameter K_r is introduced in Eq. 1?

12. Line 190. “transitions are possible, especially on evolutionary timescales that could span the bridge”. Again, it might be worth referring to variation in the dSA magnitude here, to explain what is meant by the currently somewhat vague “transitions ... on evolutionary timescales that could span the bridge”.

Referee #3 (Remarks to the Author):

Since the seminal work of Pringle, I have not seen a more compelling and exciting mechanistic study of insect muscle function. Having said this, it looks like the model switching between synchronous and asynchronous muscle function underpinning this work has been constructed without a biophysical representation in the context of actual insect functional morphology. Reading the manuscript and its mathematical basis in detail, I concluded that this is indeed the case and cannot be resolved as it is a central assumption in/simplification of the model. Based on the literature available and given the complexity of this system and the new mathematical framework the authors present, the best revision of the manuscript maybe to point this out explicitly. Further, the authors could replace the current speculative last paragraph(s) of the manuscript (that do not strengthen the manuscript) with new ones discussing how the actual embodiment of their model could be established by future insect muscle research. The new last paragraph should give some specific tangible pointers for this future work motivated and supported by specific findings of your research. That would address the biggest weakness of the current write-up of the otherwise visionary research and make it more robust as a Nature paper. As the authors probably realize themselves, their approach has several weaknesses that can be easily criticized because they do not present the necessity/logic for these simplifications clearly in the manuscript. By being open about this in the concluding paragraphs and more informative about why their assumptions and approach is the best way to make progress given the missing information in the literature, this otherwise visionary manuscript will more robustly influence the work of colleagues in the field and generate more citations accordingly the next decade.

A somewhat smaller, but pertinent, modelling limitation that is not outlined in the manuscript is that neither the mathematical model, nor the robophysical model, nor the micro flapping robot model includes a mechanistic model that explicitly translates muscle/actuator activation spikes into the muscle/actuator dynamics. There seems to be / is no asynchronous versus synchronous neural drive model based on the representative schematic insect activation pattern / pulse train shown in figure 1a. This is a large simplification that should be clear from the start in the manuscript and repeated in the last paragraph with the outline of a roadmap for future work that resolves this: including realistic activation patterns that results in the emergent dynamic behavior presented here.

Please find my detailed comments below, in the interest of reducing my review writeup time I am listing them trusting the authors will make the effort to understand their merit and resolving them productively. They may come across as quite critical, but both the authors and I know that any study trying to advance our mechanistic understanding of insect muscle will encounter most of them. Hence, I see this manuscript as a mechanistically substantiated roadmap that can guide the field for the next decade if the authors make the effort to point these issues out and write more balanced discussions and conclusions. The order of the comments is mostly chronological as I read

the manuscript page by page.

Finally, please note none of the comments are for me, they are for helping the Nature readership understand your work. So please do not give me an explanation while not giving one for the reader in the manuscript. I am not debating the validity of your research; I am giving pointers for substantiating it fully and making the work clear for the entire multidisciplinary readership.

1) Throughout the manuscript no acronyms should be used. Even if wording is often repeated, it impedes the multidisciplinary readership of Nature because the reader is forced to memorize something that is at best field specific and generally unimportant to understanding the science when it is simply written out. Excellent writing makes acronyms obsolete and shortens the manuscript simultaneously. Acronyms should also not be used in figure captions and in the figures themselves, variable symbols should be replaced with words representing what is plotted whenever possible: I noticed this was possible in virtually all figure panels, there is sufficient space along the axis. This makes the manuscript much more readable to all muscle physiologists, neuroscientists, biomechanists, evolutionary biologists, entomologists etc. who will all find this work of interest. It also helps physicists and engineers not familiar with this type of biophysical research and the associated nomenclature.

---Examples:

"dSA" I found it a drag to read and memorize, unhelpful.

Fig 1b "X" "F" are unhelpful etc.

Fig 2, the labels 2c,d,e vertical are cryptic: but notice the horizontal labels are great (except for in 2e dSA and Tet., Twitch is good again).

Fig. 3 all labels. Variables in a,d are helpful for introducing them but fonts are small and it would be great to also add a word explaining them.

Etc. I am not repeating these below in the figure comments, they are all centralized here.

2) "found that there has most likely been" caveats should be outlined in the methods and the reader should be alerted with "(but see methods)"
e.g. since entomologist typically estimate only 10% of insect species have been discovered and of those about a million species almost none have been studied there are caveats. Also, the reconstruction of the phylogeny tree is based on selected sequence snippets that do not code for muscle properties as typical for almost all phylogenetic studies. These are limitations for phylogenetic studies generally, and acceptable according to several fields, but they need to be listed.

3) " These patterns suggest that after the evolution of asynchrony, the physiological properties associated with asynchrony, such as dSA, could be conserved in secondarily synchronous fliers. " it just shows the most likely scenario supported by the particular assumptions underpinning the phylogenetic tree construction and the way evolution was propagated. This is fine, but not a certain statement, the manuscript is not clear on this. The biophysical framework in combination with the phylogenetic tree are an important advance regardless.

4) Figure 1:

1a) what parameters are plotted, show horizontal and vertical axes

1b) color parameters according to trace. Do we need parameters here, these axes should be labeled with informative names to increase accessibility. Add a horizontal scale for all these traces.

Caption:

- " The evolutionary history " should be "Current..."

- " actuation reveals " should be "suggests"

- It is great style that the acronym is introduced again in the caption of the first figure, but a readable nature paper in our small multidisciplinary field requires writing without acronyms and maintaining length constraints by cutting non-essential text of which there still is quite a bit (e.g. the last few paragraphs are highly speculative and not the most informative given the new research road map this new model dictates, which is more worthy to outline briefly in a single last

paragraph).

5) Figure 2:

2c) the legend is unclear and not all traces indicated, what is what?

Caption:

-replace acronyms with text (also DLMS, it is unreadable)

- r_3 is poor naming since it is the wing radius variable for aerodynamic power required to beat the insect wing - rename or don't use a parameter symbol and instead write out the name. This is a problem throughout that needs to be resolved systematically.

- " previously-established linear relationship between " provide reference here.

6) " The timing of the dSA response are typically characterized by fitting a sum of three exponents¹⁷ representing three phases: " show in a figure, fig 1b would probably be good fit.

7) " where the spring represents parallel elasticity from the deformable exoskeleton " is this a linear model, clarify here. What is the justification for assuming a linear thorax spring system? The reader needs it.

8) " we modeled dSA force as the convolution of the stretch-hold-release-hold response with velocity " please define for a general audience: e.g. wikipedia is a lot clearer: the integral of the product of the two functions after one is reversed and shifted. Why is a convolution model appropriate, what is the logic? Please include it concisely. In the discussion of results, are there limitations of this model for the flight muscle apparatus? In terms of solving the underlying coupled differential equation, please clarify. Equation 3 is a lot clearer than writing about a convolution since that is just the solution of this particular system with its assumptions.

9) " The muscle strain rate scales with wing angular velocity by a factor of LT , where L is the resting length of the muscle, and T is the transmission ratio, $LT \cdot \dot{\epsilon}(t) = \dot{\phi} / 131 (t)$. To " Where does this come from and how come the transmission ratio is a factor here. Given this is strain rate, non-dimensional length change rate, how do we get per second? Aren't there scaling parameters missing here. Especially considering the transmission ratio is wing stroke amplitude / muscle amplitude. This should be resolved for any reader without assuming knowledge about traditions. Also, in some fields this is considered wrong/erroneously and unphysical so keep in mind such idiosyncratic traditions do not cross multidisciplinary borders because they have no generalizable logic.

10) Figure 3:

1a) symbols too small; is F_{aero} applied at r_2 (aka the radius of gyration in insect wing aerodynamics, the radius where the net force acts)? If not resolve everywhere and clarify where the force acts in your model.

1b,e) unusual power units that are unhelpful, why not Watts or non-dimensional?

1c,f) the short blue segment on the color bar is really unclear and unexpected/unusual and thus confusing to most readers. Why not plot a more traditional plot with different curves for different f/f_s values so we can see which combinations give 1? Also, how close do we need to be to one? Are the values deviating from 1 far away, it seems most of the space has the frequency match, but it is the power that is the issue.

1h,1) clarify color code, looks like this is blue synch and red asynch and that the transition is from one to the other via the gain. Again, how is the gain controlling mode switching implemented in the robot? Clarify, I only understood after reading the methods that basically the actuator is controlled with a voltage pulse generated by the model in 1a: Meaning that this is not an emergent property and simply a playback of the model in a robot showing the model ignores friction in the robot and possibly in the insect. Because the robot does not embody the model in 1a) it is also not an independent proof of its validity, it just shows the model can control the flapper feed-forward. Yet this is nowhere clear in the manuscript and even somewhat hidden in the methods. This should be communicated upfront in the caption and in the main text with a reference for details in the

methods. The methods should be revised to be clearer on this as well. The final paragraph should outline how the model could be embodied in the robot, so the behavior is naturally emergent from the electromechanical system in closed loop.

Caption:

- "stretch activated feedback via a dSA dynamics model (Eq. 8-11), and combined synchronous and asynchronous muscle force with "gain knob" K_r " make specific what the parameter is and captures. The definition used in the methods should be written here since it is central to the scientific idea that this paper discusses: " K_r describes the ratio of synchronous and stretch-activated forcing"

A critical aspect missing in this manuscript is a biophysical explanation for what K_r actually is and how it is represented in the flapping wing apparatus. If this is not known, it is just a model that shows that by linearly combining async and sync forcing both can be represented. Given the same knob is used in the experiments it would be key for proving the main idea to pinpoint this mechanism in the insect muscle physiology.

Also, it is unclear if the insect aeromechanics dynamical system is typically a periodic limit cycle system, there is evidence for it being less regular or even chaotic (strange attractor) in some flapping wing regimes. I am not concerned about this, but this context would be valuable to add in the road map for future work.

11) " as a quasi-steady term proportional to velocity squared, " fine but do know the coefficients can vary during the cycle and that they don't always fall on a limit cycle. They can also fall on a strange attractor. I don't think this invalidates the idea in this paper, it is just that more research will be needed in the future as usual at the cutting edge.

12) "(T) that transforms linear muscle displacement into rotational wing movement. " define in the main text since its an important factor.

13) "rates, we reparameterized r_3 as t_0 , which" reparameterization is completely unnecessary and highly confusing and thus unacceptable in a research paper. Choose one distinct parametrization and stick with it. Also, if there are difficulties with this, refer to methods and provide clarifications for specific choices there.

14) "which is the time delay to peak dSA force (Fig. 1b)." visually indicate the time parameters in 1b and use normal wording not only variable names, so the graph can be read by all Nature readers.

15) "scales. T_n is determined by the mechanics alone (I , kl , and Γ ; see Methods)." give eqn here, since it the characteristic frequency / representative period of the system (depending on the most useful perspective to explain the main point of the paper to all readers).

16) "a narrow ratio of muscular and mechanical time constants" clarify: what is the critical constraint mathematically and how does it relate to muscle mechanics and aeromechanics.

17) "Although changing K_r is the most obvious transition, moving along the t_0/T_n axis can also drive the system from synchrony and asynchrony, but the power output quickly diminishes in either direction." draw both paths in the model figure: 3a-c

18) "Around $t_0/T_n \approx 0.2$, we find" is this just math? Can this be predicted? What does this tell us about the interaction between muscle mechanics and aeromechanics? Clarify for the reader.

19) "especially on evolutionary timescales that could span the bridge, while" for this we would like to know the biophysical mechanism that represents the knob as well as the presence of both synchronous and asynchronous muscle control. It helps the reader to remind them of the in vivo evidence you found so this doesn't become a purely mathematical construct. If it is just math then write this and don't speculate, if it is unknown, it is future research for your roadmap paragraph.

20) "subsequent shifts between asynchrony and synchrony (1c)" are you thus suggesting there exist insects today (given there are ~10 million species) that are on the crossbridge in the middle or near the middle? How could this be tested in future research? Evolution is ongoing, there is no evidence insect muscle evolution converged into a final state.

21) "To test this hypothesis that insects can realize both synchronous and asynchronous oscillations simply by changing a ratio of timescales and a gain coefficient,"
" ratio of timescales " this can be understood
" a gain coefficient " again how is this represented in vivo? The manuscript needs to be clear on this.

22) "This arrangement mimics the indirect actuation of both kinds of insect flight muscle which act via the deformation of the thorax to sweep the wings back and forth." shouldn't this information come early when describing the differential equation model, the reader is completely surprised when it comes this late.

23) " when t_o/T_n exceeds ≈ 0.3 , possibly due to the dSA force failing to overcome friction " so how do we know that the high aerodynamic drag doesn't impede this in insects? And doesn't the thorax also have frictional losses? Does the model predict this and may this constrain the discussion? Also, insect flight is energetically so cheap, simply due to the square cube law, that insects do not need to be as efficient. So the assumption that friction plays no role cannot easily be supported. Given the problems the Wood lab uncovered with micro scale mechanism friction and given there is a scaling paper on this issue and how it impacts micro scale flight by Hawkes, this requires clearer information for the reader.

24) Line 224-228: This observation would be highly informative in the introduction for motivating the study instead of presenting it at the end.

25) " and asynchronous modes as K_r varies, as we predict insects " how was this achieved in the robot? What is the design that makes this possible? Is it off-board or on-board in the sense is it tiny enough to fly? I only found out when reading the methods: all of this information is critical for the reader to understand this work and needs to be communicated in the main manuscript.

26) " Then, we changed K_r linearly from 1 to 0 over two seconds. " how? Clarify this here for the reader.

27) " range of (K_r Fig. 3h). " typo, parenthesis should start before Fig.

28) 247-259: too speculative, unclear what benefit a robot would have to switch. Isn't the single one-line advantage that the robot wing flapping control system does not need to pulse each stroke as in insects. This is also what instigated this research so this is a nice closer. Otherwise, it is speculation without significant merit or proof for merit. Less is more. And the space can be used to clarify the rest of the highly informative and valuable work.

29) " , and we might find these features to lie on a continuum." what is the functional morphological or developmental evidence for this? Show it or remove.

30) 274-277: too speculative, I would prefer a discussion of how this could be observed by studying current species diversity, with 1mln counted species and an estimated 10mln total it seems like there should be evidence for these claims in extant species. How can this be found, what should entomologist test / study? Construct and present a research roadmap in the last paragraph.

31) " Still, the physiological properties and dynamics of indirect spring-wing systems are not necessarily dichotomous." unhelpful summary statement of a final paragraph.

32) " (LaMSA)" please leave acronyms out - also not used again in the main text. The small subfield using LaMSA can also connect the acronym to " latch-mediated spring actuation " and this cryptic wording could be replaced with something clear to a biologist not knowledgeable about biomechanics. Make specific how the latch is mediating. Is this not simply a spring triggered to expand by a latch? Why not keep it simple?

33) " with a blended actuation strategy enables " the ms is unclear on the in vivo mechanism representing this. Resolve this for the reader.

34) " both time-periodic and stretch-activated forcing " please more discussion on this mechanism instead of all the speculation on less important implications that are not well substantiated.

35) " Via a set of simulation, robophysical, and robotic experiments, we have developed a unifying spring-wing framework that describes synchronous-asynchronous evolutionary transitions in insects and generalizes to robotics. " obsolete at the end, this is a summary sentence that better serves as an opening of a discussion or conclusion section.

Specific methods section comments:

36) " Iterative force tuning procedure K_r describes the ratio of synchronous and stretch-activated forcing, but does not capture the absolute magnitude of each. " this paragraph seems to explain some of the issues that may also pertain to why the gain K_r is not clearly defined based on in vivo parameters. And why the mechanism isn't mentioned. Please clarify these issues for the reader.

37) "Calculation of K_r for *M. sexta* To calculate K_r for *M. sexta*, we use the following formula, with $F_s = 2721$ mN and $F_a = 320$ mN. " This is key, did I miss this in the main text? where do the input values come from?

Statistics statement:

replication-please provide sample sizes:

N individuals

n repetitions per N

I assume these have then been conducted once. But this is not clear

randomization-report statistics on this for all the trials, were all stimulations effective, what was the definition used to assess, those kind of insights.

Author Rebuttals to Initial Comments:

Referees' comments:

Referee #1 (Remarks to the Author):

This is an exceptionally interesting study that explores the evolution of transitions between synchronous and asynchronous flight systems in insects, and employ some novel robotic models to demonstrate that these seemingly disparate flight neuro-mechanical systems are actually based on the same dynamics and that transitions between the two can be achieved more easily than previously surmised. This conclusion matches with their ancestral state reconstructions, which demonstrate multiple transitions (principally reversions) between these states. It is a unique exploration of one of the key features in the history of insect evolution and the refinements and specialization of insect flight.

In terms of the methods, my only real concern is the undefended position of assuming an equal rates model for exploring the evolution of these traits across insect phylogeny. This seems to be a naïve assumption and I encourage the authors to explore this instead through the application of a relaxed clock model with uncorrelated rates in a Bayesian framework. This is likely to be a far more realistic approximation of the probability of given transitions. Or, at the minimum, some defense of the underlying assumptions in applying an equal rates model, most of which would seem to be violated by the system they discuss. Otherwise, aside from this one concern, the validity of their methods and analyses are certainly sound.

The citations presented appropriately credit the necessary sources, and the writing is clear and easily understood by a broad readership, with conclusions that do not overreach from the results of their work.

I encourage publication once the concerns regarding the ancestral state reconstruction are resolved. I look forward to seeing this work in print.

We thank the reviewer for the very positive and encouraging review. The reviewer brings up an excellent point that equal transition rates between traits and constant transition rates over (evolutionary) time can significantly impact the reconstruction of ancestral states. As the reviewer likely knows, in past papers this has been addressed with hidden rates models where multiple rate classes are used and transitions between the rate classes are considered within the transition rate matrix.

One challenge with unequal rates and hidden rate transitions is that they introduce more complexity to the model. We thought the limited number of transitions across the tree (~10) would be overfit by more complex models which was our initial motivation to keep things simpler, but we appreciate the reviewer's prompting to take a more rigorous look at this. We have done several things to address this.

First we compare an equal rates model with a single rate class with all rates differing. We also consider these models under both the case where wingless and unknown insects are coded as

distinct trait states and where these states are left ambiguous. In these cases ancestral state reconstructions are consistent in showing the same patterns across insect orders with a single transition to asynchrony at the node comprising the holometabolous insects + Hemiptera + Thysanoptera + Psocodea (Node 200). The resolution within Hemiptera could change and indeed we think that this clade deserves a lot more attention because there are more transitions and not extensive sampling compared to the diversity of species present (we emphasize this more now in the discussion as also prompted by Reviewer 3). We have added a statement to the results that these reconstructions are consistent across different rates models at least at the order level. The all rates differing model actually is the most favored by sample size corrected Akaike Information Criterion (AICc) and shows 100% reconstruction of Node 200 as the only origin of asynchrony.

However, we also appreciate that the number of transitions is not the only factor that determines the fit. Sampling matters as well and here we have used to our knowledge every available measurement of muscle ultrastructure, but obviously not all insects species or even families are sampled. We therefore have focused the discussion at the ordinal level. We also still report the ER model in the paper because it is more conservative and better reflects that there is some uncertainty given sampling limitations.

Second, we consider the possibility of transitions with the rates themselves (hidden rate models as a subclass of hidden markov models) using the *corHMM* package in R. Since we were already working in a MCMC framework we utilize this package, and at least one prior explicit comparison has shown the results are consistent with a Bayesian approach implemented in *BEAST* (King & Lee, 2015, *Sys Bio.*). We found that considering multiple rate classes produced overfit models as judged by AICc and by examining the reconstruction which showed many transitions occurring along several long branch segments. It is important to acknowledge that this doesn't reject these models, it just shows that they are less favorable.

Some of the less favored models produced consistent reconstructions at the order level with one transition to asynchronous and multiple reversions to synchrony. Others were more ambiguous and we have revised the paper to acknowledge that other models are possible, even if less favored. To address these points, we have added a methodological description of the different models and their results to the Supplementary Information. We have added a short section in the main paper pointing out the robustness of our conclusions to these different evolutionary models and pointing to the new supplement.

Finally, one of the strengths of our approach is that we use the phylogenetic conclusions to prompt a deeper look at the physiology in putative secondarily synchronous species. Finding delayed stretch activation in *Manduca* (and not in locusts as has been previously shown), further supports an asynchronous ancestry to lepidopterans.

Here is the expanded discussion in the first section of the main document addressing these points. Please also see the new Supplementary Information, part IIA.

“We first assumed an equal rates model of evolution and utilized a Markov chain Monte Carlo approach to estimate the state of ancient insects. In the most likely reconstruction, there have been many independent reversions back to synchronous flight muscle from the single origin of asynchrony, which we will call secondarily synchronous flyers (Fig. 1c, Extended Data Figure EDF3). We found that Mecoptera, Lepidoptera, Neuroptera, Megaloptera, and Raphidioptera are all most likely secondarily synchronous orders. This pattern of transitions is consistent across alternative models of evolution. The best fit model actually had all transitions unequal with ambiguous coding of wingless species and produced a 100% reconstruction of a single asynchronous origin at Node 200. Even if muscle structural data is available across most orders, we still only have samples from a small number of all insect species. We, therefore, show the more conservative equal rates model 1c. Incorporating heterogenous rates across the phylogeny^{16,17} did not produce better model fits (See Supporting Information IIA). Ancestral state reconstruction can change with more sampling and different phylogenetic reconstructions, but the current best evidence supports a single origin of asynchrony at the order level. Most importantly, our analysis raises the possibility that the physiological properties associated with asynchrony, such as delayed stretch activation, could be conserved in secondarily synchronous fliers. If so, this would provide evidence that both modes can co-occur across the phylogeny even if the muscle ultrastructure appears as a specific type.”

Thanks again for this suggestion, the paper is stronger for it.

Referee #2 (Remarks to the Author):

This is a very interesting manuscript on an important (but in recent times, neglected) topic, viz. how the two fundamental mechanisms of generating insect wingbeat oscillations in flight have evolved, and how they interact. These two mechanisms are referred to as “synchronous” or “asynchronous”, respectively, according to whether neural activation maps in a one-to-one fashion to muscle activity. Asynchronous musculature is typically associated with high power output requirements at high wingbeat frequencies, making its origin a key transition in the evolution of insect flight.

The authors use formal ancestral state reconstruction (ASR) methods to identify the likely number of origins and/or losses of asynchronous flight musculature across taxa. This is an important contribution, because several previous works have drawn or repeated the conclusion that asynchronous flight musculature has evolved multiple times in insects on the informal basis that its occurrence appears scattered across the phylogeny. In contrast, the authors’ formal analyses provide strong statistical support for the opposite conclusion, that asynchronous musculature evolved once and has subsequently been lost in several major taxa.

The authors follow their evolutionary analyses with a more detailed analysis of the muscle physiology of several species. This analysis shows that delayed stretch activation (dSA; a necessary property of asynchronous muscles) is present in species identified as secondarily synchronous by their ASR analysis (albeit to a lesser degree than in species classified as asynchronous), but is lacking in species for which synchronous musculature is the ancestral state.

Finally, the authors use a dynamics framework to model how dSA can coexist with synchronous neural activation in secondarily synchronous flight muscles, and by extension to unify our understanding of the dynamics of synchronous and asynchronous musculature. They do so through three parallel analyses involving computer simulation, robophysical modelling, and an insect-scale robotic flapper. *This is a useful conceptual contribution, although the main result – that dSA can coexist with synchronous neural activation provided that the frequency of the synchronous activation matches the natural frequency that the dSA generates – is not in itself surprising*.

We sincerely thank the reviewer for their careful reading of the manuscript and the concise summary above. We agree that the role of power muscle actuation in insect flight—and specifically the differences between synchronous and asynchronous actuation—have been less explored recently. With the help and encouragement from this review we hope to bring some of these issues back to light.

We do think that the dynamics framework offers a critical element to this manuscript. It is known from work on the periodic forcing of limit-cycle systems that periodic and self-excited oscillations can co-exist, but there is a rich class of behavior of these systems with respect to entrainment, bifurcations, and interference. This is important behavior in the region where synchronous and asynchronous modes are mixed. Therefore, we cautiously agree with the reviewer “that dSA can

coexist with synchronous neural activation provided that the frequency of the synchronous activation matches the natural frequency that the dSA generates”, and that from a pure mathematical context this is unsurprising, but still an active area of research in dynamical systems. However, from a biological context this result is important because it shows that this class of systems applies in an interesting and important way to the patterns of evolution and biomechanics of insect flight.

In response to the reviewer’s specific comments below we have now significantly increased and clarified the discussion of these dynamics.

This is a well written paper, but I would like to see the authors address the following comments:

General comments:

1. The results of the dynamics analysis as presented seem largely unsurprising, and I would therefore have liked to see some more detailed discussion of the dynamics they capture in order to maximise the value of the analysis. The authors refer at one point to an “apparent Arnold tongue”, but this jargon is unlikely to provide much dynamical insight for readers not already familiar with what is meant by this description of the appearance of a visualization – even with the additional explanation at Lines 184-186. On the other hand, there is none of the discussion that I think I might have expected to see on what look to be period-doubling bifurcations visible in Fig. 3i, or indeed of the dynamical meaning of the pink areas of Fig. 3c, which represent the region of the parameter space where variation in dSA drives variation in the frequency of asynchronous muscle activity. Surprisingly, the word “bifurcation” appears nowhere in the manuscript at present. This more detailed discussion would make it possible to assess the correctness or otherwise of the authors’ identification of an “Arnold tongue” in Fig. 3c, where their use of the word “apparent” seems to imply hesitancy on their part. Some further discussion of the variations in power output in a system being driven at or near resonance would also have been helpful in relation to the “bridge” that the authors describe, in grounding readers’ understanding of the phenomena that these plots describe.

The reviewer makes many helpful points in this comment. We will individually address them below:

- a. “The authors refer at one point to an “apparent Arnold tongue”, but this jargon is unlikely to provide much dynamical insight for readers not already familiar with what is meant by this description of the appearance of a visualization – even with the additional explanation at Lines 184-186.”

We agree with the reviewer that the term “Arnold tongue” is not descriptive or inclusive for audiences unfamiliar with forced limit-cycle oscillators. We have decided to remove the direct reference to an Arnold tongue and directly describe the phenomenon instead. We now refer to the phenomena of 1 to 1 frequency entrainment ($f = f_s$) as “synchronous frequency entrainment” in the manuscript.

The updated sentences are as follows:

In Figure 3 caption

“Emergent wingbeat frequencies (f) normalized by the synchronous drive frequency (f_s). Blue indicates regions where synchronous frequency entrainment occurs ($f/f_s = 1$). The red regions indicate where the asynchronous dynamics dominate and the forcing is predominantly stretch activated ($f \neq f_s$).”

And in the main text

“As the synchronous forcing becomes stronger relative to the asynchronous dynamics, we see that a larger region near $t_0/T_n = 0.2$ becomes entrained to the synchronous frequency³⁵. Entrainment is the process where a self-excited oscillating system is forced to oscillate exactly at the frequency of an external driving frequency (a phenomenon called an Arnold tongue³⁵).”

To further test that the phenomena we observe represents a frequency locking bifurcation, which is a hallmark of an Arnold tongue, we have performed new experiments, new discussion into the text, and a new Extended Data Figure. Briefly, we performed new experiments to sweep the driving synchronous frequency, and the synchronous force magnitude, while keeping the asynchronous properties the same. This experiment allowed us to observe the “textbook” phenomena of an Arnold tongue: 1) frequency entrainment when the driving frequency, f_s , is near the asynchronous frequency, and 2) a widening of the frequency locking regime as the synchronous force is increased. We include here the new experimental figure (Fig. EDF8) that has been added to the supplemental material.

Figure EDF8: Synchronous entrainment of asynchronous oscillations experiment. a) Emergent wingbeat angle versus time in experiments with a combination of asynchronous and synchronous actuation. The four plots correspond to varying the synchronous drive frequency compared to the emergent asynchronous frequency. The synchronous drive is increased from curves 1 through 4 as shown by the arrows in (b). As the synchronous drive frequency gets closer to the asynchronous frequency the wing motion exhibits large amplitude modulations due to a beat frequency between synchronous and asynchronous oscillations. However, when the synchronous drive is close enough the emergent wingbeat frequency entrains to the synchronous drive and the amplitude fluctuations disappear. b) The emergent wingbeat frequency compared to the driving wingbeat frequency. The gray region indicates frequency entrainment where the emergent frequency is exactly equal to the driving synchronous frequency. The three plots are of increasing synchronous forcing magnitude from bottom to top. c) The amplitude fluctuations increase as the synchronous frequency approaches the asynchronous frequency. However, when the driving frequency cross the Arnold tongue and the emergent frequency becomes entrained to the synchronous frequency, then the amplitude fluctuations disappear.

Furthermore, this experiment highlighted the large amplitude fluctuations that occur near the boundary of the asynchronous and frequency entrained regimes (Fig, EDF8a). This transitional regime would be extremely detrimental to any flying robot or insect, and thus highlights why one would want to be either frequency locked (i.e. on the bridge), or far away from the locking boundary. To quantify the non-steadiness of this beating phenomena we measured the standard deviation of the peaks of the waveform. We plot this non-steadiness measure from the

experiments in the new supplemental figure EDF8c, and in an additional supplemental figure we have computed this same non-steadiness metric for our simulations which we include below.

Figure EDF7: Wingbeat amplitude variation in Manduca simulations. a) The three plots show the wingbeat angle versus time at three different values of t_0/T_n , for a constant $K_r = 0.2$ (shown by the white line in the heatmap). After transient oscillations die out we measure the amplitude of wingbeat peaks as a function of time (positive peaks are shown as red triangles). The top plot is an example in the asynchronous regime, displaying moderate amplitude fluctuations. The middle plot is the wing angle within the frequency locking synchronous regime (on the “bridge”) where the wing amplitude is steady. Lastly, the bottom plot shows the wing motion in the asynchronous regime below the frequency locking “bridge”. b) For all combinations of K_r and t_0/T_n we calculated the standard deviation of the wingbeat amplitudes which we show as a heatmap. Brighter regions of the plot correspond to where large stroke to stroke amplitude variation occurs (i.e. top and bottom plots in panel a). When the wingbeat is steady the amplitude variation is small these appear as the black regions. The boundaries between the synchronous and asynchronous regimes exhibit large amplitude fluctuations, while the bridge connects the synchronous and asynchronous regimes with smooth sinusoidal emergent wingbeats.

This new analysis of the simulation results clearly shows that regimes near the boundary of the bridge exhibit highly fluctuating wingbeats. Thus, in addition to lower power wingbeats occurring off the bridge, the unsteadiness of the emergent wingstroke dynamics in the regimes just off of the bridge is another factor suggesting that being on the bridge is important for physically realistic, and feasible wingbeats.

We thank the reviewer for prompting these new experiments and we hope they help illuminate more of the dynamical phenomena in this system.

- b. On the other hand, there is none of the discussion that I think I might have expected to see on what look to be period-doubling bifurcations visible in Fig. 3i,

We see the phenomena the reviewer is pointing out in Fig. 3i, in the 20 Hz forcing experiment there appears to be two jumps in frequency near the region where synchronous forcing goes to zero (at time $t = 2$ s). This appearance of frequency “jumps” however is not a signature of period doubling in these experiments since period doubling is a bifurcation that causes the oscillations to change character *in the steady-state*. The frequency jumps in Fig. 3i are transient and only occur during the transition from synchronous to asynchronous dynamics. To elaborate on this we have provided a new Extended Data Figure (Figure EDF10, reproduced below) which shows the details of the wing position versus time during the transition which shows that the frequency jumps are associated with transient behavior of the system during crossover from synchronous to asynchronous.

Figure EDF10: **Synchronous to asynchronous transitions in the robobee wing.** Four tests at $f_s = [20, 40, 67, 100]$ Hz are shown in which the robobee is transitioned from synchronous to asynchronous forcing. Each experiment consisted of one second of synchronous flapping (blue region) at a particular frequency, followed by a 2 second transition in which K_r was linearly increased from $K_r = 1$ to $K_r = 0$ (top plot) followed by 2 seconds of 100% asynchronous operation (red region). The Wingbeat angle and frequency are plotted for each of the four experiments (f_s is indicated on the right hand side). The left column shows the full time course of the experiments while the right column is a zoomed in region of the onset of fully asynchronous dynamics.

In addition to the above changes we have added to the figure caption to further clarify that amplitude and frequency modulation occurs only during the transition periods between fully synchronous and fully asynchronous. The new text is below:

“Transitions are smooth when synchronous and asynchronous frequencies are approximately equal (blue and red markers, respectively). However, when the synchronous frequency substantially differs from the asynchronous frequency, interference between synchronous and asynchronous dynamics causes the oscillation amplitude and emergent frequency to fluctuate during the transition regime.”

- c. or indeed of the dynamical meaning of the pink areas of Fig. 3c, which represent the region of the parameter space where variation in dSA drives variation in the frequency of asynchronous muscle activity.

This is a great point and we thank the reviewer for prompting this response. We have followed your advice and now performed a thorough analysis of the dynamics in the asynchronous regime ($K_r = 0$). First, we examined the emergent wingbeat dynamics from the simulation and observed that as t_0/T_n increases, the system undergoes a bifurcation from no limit-cycle oscillations (low t_0/T_n) to emergent limit-cycle wingbeat oscillations (large t_0/T_n).

To better understand these dynamics and how limit-cycle oscillations emerge we performed a linear stability analysis of the coupled system (delayed stretch activation and spring-wing mechanics). In the Supplementary Information document we provide a full analysis of the linearized dynamics and we are able to demonstrate that asynchronous wingbeats occur through a Hopf bifurcation in which a pair of complex conjugate eigenvalues for the linearized system go unstable (gain a positive real component).

In a new Extended Data Figure (see below) we now compare the simulation and linearized analysis results and find excellent agreement. From the linearized system we are able to calculate in closed form the critical t_0/T_n where oscillations emerge, and oscillation frequency of the linearized system. Lastly, this analysis also demonstrates that the bridge emerges exactly at the location where the asynchronous oscillation frequency is equal to synchronous driving frequency.

We thank the reviewer for this suggestion as it has greatly improved our understanding of the phenomena.

Figure EDF5: **Emergent frequency (f) normalized by the synchronous frequency (f_s) in the asynchronous regime ($K_r = 0$).** As the time to peak force from delayed stretch activation (t_0) is increased the system undergoes a bifurcation in which steady wingbeats emerge. Dashed line is the prediction of this critical t_0/T_n from analysis of a linearized system. The emergent wingbeat frequency in simulation (circles) decreases as t_0 is increased. A linearized analysis of Equations 1 & 2 (See SI for details) is able to capture the emergent wingbeat frequency. The colormap of simulation points matches the heatmap of Fig. 3c,f. The bridge emerges when the emergent frequency from delayed stretch activation matches the synchronous frequency.

- d. Surprisingly, the word “bifurcation” appears nowhere in the manuscript at present. This more detailed discussion would make it possible to assess the correctness or otherwise of the authors’ identification of an “Arnold tongue” in Fig. 3c, where their use of the word “apparent” seems to imply hesitancy on their part. Some further discussion of the variations in power output in a system being driven at or near resonance would also have been helpful in relation to the “bridge” that the authors describe, in grounding readers’ understanding of the phenomena that these plots describe.

This is a good point, and one we have corrected in the updated manuscript. We now reference two bifurcations in the dynamical phenomena: 1) the emergence of asynchronous oscillations through a bifurcation (as discussed in the previous comment response), and 2) an entrainment bifurcation where the system undergoes a sharp transition to synchronous frequency entrainment. The new text for these changes are as follows:

“As the time to reach peak force of the asynchronous muscle (t_0) is increased we observe a bifurcation in the steady-state wingbeat amplitude and frequency as oscillations first appear when t_0/T_n crosses a critical value (see Extended Data Figure Fig. EDF5).”

“As we move away from the bridge along the t_0/T_n axis, there is a bifurcation and the the asynchronous and synchronous frequencies diverge, ending the entrainment, and leading to emergent asynchronous oscillations (Extended Data Figure EDF6).”

To better illustrate that the boundaries between the synchronous and asynchronous regime reflect an Arnold tongue we have added a new Extended Data figure (EDF6) that illustrates the spectral content of our simulations. The figure shows that for values of t_0/T_n that are not on the bridge, there is a mixing of synchronous and asynchronous frequencies until, at critical values of K_r , the asynchronous frequency completely disappears. Near these boundary regions the interference of these dynamics leads to large fluctuations in amplitude as discussed in the previous comment. We believe that this new figure, and the amplitude fluctuation figure (EDF7) provide ample insight into what is going on at these boundary regions between synchronous and asynchronous regimes, and the relevance of the bridge which connects synchronous and asynchronous regimes with smooth sinusoidal wingbeats.

Figure EDF6: **Emergent frequency and Fourier transform of wingbeats versus K_r in simulation.** Left plot shows the normalized emergent frequency (f/f_s) from Fig. 3a using a continuous colormap. Six horizontal lines correspond to values of t_0/T_n where we examined the Fourier transform of the emergent wingbeats. The six plots in the right column show heatmaps of the Fourier transform of wingbeat at each value of K_r . As t_0 decreases (from top to bottom), the emergent asynchronous frequency varies, and we see mixing of asynchronous and synchronous dynamics near the boundary between async and sync regions (shown in gray). The bridge between synchronous and asynchronous regimes occurs when the emergent async frequency is exactly equal to the synchronous driving frequency (Plot c in right column).

2. The dichotomous shading of Fig. 3c and f seems highly arbitrary, and risks giving the appearance of bifurcations between the regions of the graph where none may in fact be present. It looks to me as if the boundaries between the blue region and the dark pink region may indeed represent jumps in emergent wingbeat frequency (i.e. bifurcations), but it seems plausible that the boundaries between the blue region and the pale pink represent smooth transitions. Whilst I can see the merit of the panel coloration that is shown, I think it is essential to complement this with a continuous colormap to avoid this kind of issue.

This is a very important point and we have now included a continuous colormap version of Fig. 3c in the Extended Data Figures (EDF6, shown above in previous comment response). In addition to the frequency plot, we have also included a new plot which shows the fluctuation in peak-to-peak amplitude that occurs near the boundary regions (EDF6). These new figures help to illustrate that locations off the bridge undergo a distinct change in emergent oscillations, reflecting a bifurcation between the synchronous entrainment regime, and the asynchronous regime.

3. The location of the *Manduca* icon on Figure 3 seems to suggest that it sits far away from the “bridge” that the authors describe, so I can see little evidence in the results that *Manduca* is exploiting any amplification. This doesn’t necessarily detract from the other findings, but it does raise a question on the evolutionary importance of the resonance “bridge” that the authors have identified in the dynamics. For instance, another way of looking at their results is that Fig. 3c, f demonstrates that it is perfectly straightforward to switch from asynchronous to synchronous dynamics (which is the evolutionary transition that the authors find has occurred multiple times on the phylogeny), simply by decreasing dSA magnitude. It is then an open question whether it would be beneficial to do so by being on the “bridge” (i.e. by exploiting resonance), although the one datapoint shown on the figure may suggest otherwise.

The reviewer makes many good points in this comment. We will individually address them below:

- a. The location of the *Manduca* icon on Figure 3 seems to suggest that it sits far away from the “bridge” that the authors describe, so I can see little evidence in the results that *Manduca* is exploiting any amplification.

We do not believe that *Manduca* necessarily benefits from any delayed stretch activation amplification and we make no claim as such. The relative scale of delayed stretch activation and twitch force in the muscle suggests that it could only have a very small effect on muscle behavior.

- b. This doesn’t necessarily detract from the other findings, but it does raise a question on the evolutionary importance of the resonance “bridge” that the authors have identified in the dynamics. For instance, another way of looking at their results is that Fig. 3c, f demonstrates that it is perfectly straightforward to switch from asynchronous to synchronous dynamics (which is the evolutionary transition that the authors find has occurred multiple times on the phylogeny), simply by decreasing dSA magnitude.

We argue that being on the bridge is important because that is the only place in the parameter space where it's possible to have combined asynchronous and synchronous forcing while still producing high amplitude, smooth, sinusoidal wingbeats. Our new Extended Data Figures help to support this by showing how “erratic” the wingbeats are when slowly transitioning between synchronous and asynchronous states off of the bridge.

4. In fact, there may be very good reasons to avoid being on the “bridge”. A good reason for using synchronous musculature, for example, is to have control of wingbeat frequency, which will be difficult to effect consistently at frequencies close to the resonance. This may explain why *Manduca* sits far from the “bridge”, but does beg the question of whether the “bridge” matters after all, and even whether the low dSA magnitude in *Manduca* has any functional significance at all. Perhaps, since the dSA magnitude in *Manduca* appears to be low enough that there is neither competition nor beneficial interaction between the synchronous and asynchronous dynamics, this has simply been driven to a point at which it is no longer under selection?

We totally agree with the reviewer that synchronous forcing provides opportunities for control that asynchronous forcing does not. For example, as the reviewer notes, modulation of frequency may be challenging in asynchronous insects whereas synchronous insects such as *Manduca* can modulate wingbeat frequency over short transient periods (See Gau et. al. 2021; Ref 42 in manuscript). Furthermore, being “off resonance” may also be useful because frequency modulation can be performed without large changes in actuation power requirements (see reference below).

42. Gau, J., Gemilere, R., Fm Subteam, L.-V., Lynch, J., Gravish, N., & Sponberg, S. (2021). Rapid frequency modulation in a resonant system: aerial perturbation recovery in hawkmoths. *Proceedings. Biological Sciences / The Royal Society*, 288(1951), 20210352.

We do not think that the delayed stretch activation present in *Manduca* necessarily contributes much to flight and indeed in our simulation based on *Manduca* parameters there is very little influence of stretch activation on the wingstroke and the frequency is fully determined by the neural driving frequency (i.e. synchronous). The reviewer's insights here are precisely what we think is happening. However, we realize this may not have been clear in our earlier draft and so we have revised the two points in the modeling section to say that there is no evidence that the residual delayed stretch activation contributes functionally to *Manduca* (although this worthy of further investigation!) and to discuss that *Manduca* is not on the bridge.

In the discussion of *M. sexta*'s location in the simulation:

“As expected, while M. sexta does have delayed stretch activation ($K_r = 0.88$, $t_0/T_n = 0.54$), it is firmly in the synchronous regime. Its wingstrokes are largely unaffected by delayed stretch activation as in the real moth (Fig. 3b). Delayed stretch activation, while present in M. sexta, has been reduced to a point where it is less consequential at least at steady-state, although it may still play a role under perturbed conditions with faster strains and frequency modulation⁴².”

In the discussion of the bridge:

*“Matching muscular and mechanical timescales is evidently a critical requirement for both synchronous and asynchronous power production. However, variation in the strength of the delayed stretch activation response (changing K_r), its timescale (t_0), or the resonant mechanics of the thorax and wings (T_n) could enable smooth, gradual transitions across the bridge, especially over evolutionary timescales. Biologically these parameters will be closely tied to the molecular components of the delayed stretch activation, such as the crossbridging binding, calcium responsiveness, and troponin isoforms mentioned earlier. Evolutionary transitions need not necessarily be smooth, however our model and analysis reveals the existence of a pathway for gradual transitions between a fully synchronous and asynchronous regime even while maintaining high-power wing strokes. This bridge may have facilitated the many subsequent shifts between asynchrony and synchrony (Fig. 1c). However clades like lepidoptera, which appear uniformly synchronous, may have subsequently specialized away from the bridge, reflected by *M. sexta*’s location in the model simulation.”*

Evolution certainly would not need to move across the bridge gradually, but our results demonstrate that there is a path between synchrony and asynchrony where the emergent asynchronous frequency matches the synchronous frequency. We have also clarified the discussion of this point also in response to Reviewer 3.

5. Although the first two columns of Fig. 3 explore the parameter space quite thoroughly, albeit for a fixed synchronous forcing frequency, the last column of Fig. 3 presents only two different synchronous forcing frequencies. I appreciate that it might be difficult or impossible to conduct new experiments at this point, but it would, I think, have been informative to have explored the effect of perturbing the synchronous forcing frequency to a small degree around the natural frequency of 67 Hz. This would provide a useful test of the real-world robustness of the findings, and would shed some light on the extent to which the “bridge” that the authors have identified is indeed a bridge rather than a tightrope that evolution must tread. At present, the results are stated in the figure legend as “Transitions are smooth when $f_s = f_a$... but interference causes oscillation amplitudes to drop when $f_s \neq f_a$ ” and similarly at lines 242-245. This needs rewording, because it would be reasonable to expect that small perturbations of f_s from $f_s = f_a$ (i.e. f_s not equal to f_a sensu stricto) will still allow the natural dynamics to be driven at the forcing frequency, which is what the vertical extent of the “bridge” in the other two panels presumably shows. At the least, I would suggest replacing the equality sign with an “approximately equal to” sign to reflect this.

The reviewer makes many good points in this comment. We will individually address them below.

- a. Although the first two columns of Fig. 3 explore the parameter space quite thoroughly, albeit for a fixed synchronous forcing frequency, the last column of Fig. 3 presents only two different synchronous forcing frequencies. I appreciate that it might be difficult or impossible to conduct new experiments at this point, but it would, I think, have been informative to have explored the effect of perturbing the synchronous forcing frequency to a small degree around the natural frequency of 67 Hz. This would provide a useful test of

the real-world robustness of the findings, and would shed some light on the extent to which the “bridge” that the authors have identified is indeed a bridge rather than a tightrope that evolution must tread.

We were able to add additional observations that were made using the robot at 40 and 100 Hz synchronous driving frequencies (see previous comment about period doubling, and Extended Data Figure EDF10). However, we were not able to perform further experiments on this small robot because as we have discussed above, perturbing the synchronous frequency near the asynchronous frequency results in large amplitude fluctuations which cause damage to the small robot.

However, we were able to perform the suggested experiments using our dynamically scaled robophysical system which is much more robust to amplitude fluctuations. To address the question of how small perturbations in synchronous frequency influence the emergent dynamics we did the following experiment:

- 1) We chose asynchronous parameters that established steady, high-amplitude, limit-cycle oscillations at an emergent asynchronous frequency, f_{async} (without any synchronous forcing).
- 2) We chose a magnitude of synchronous forcing magnitude, F_{sync} , that was below the nominal asynchronous forcing magnitude.
- 3) We applied a synchronous driving force with a synchronous frequency, f_{sync} , that was chosen between the range of $f_{sync} = [f_{async} - \delta, f_{async} + \delta]$.
- 4) We measured the emergent wing motion of the combined synchronous and asynchronous system, and we calculated the emergent frequency, amplitude, and the peak-to-peak amplitude fluctuations.
- 5) We swept across the frequency range of synchronous driving at three different synchronous forcing amplitudes, producing 315 experiments of synch + async observations.

This experiment allowed us to examine the following questions, with answers described below:

- 1) Do the emergent oscillations of the asynchronous system become frequency entrained by the synchronous signal?
- 2) How does synchronous forcing amplitude influence the entrainment properties of the emergent dynamics?
- 3) What does the wing motion look like near the entrainment region, are there potential downsides to being near the entrainment region?

We observe the presence of an Arnold tongue in this experiment (and also in simulation using Hawkmoth parameters). The Arnold tongue by the region near $f_{sync} = f_{async}$ where the emergent frequency is locked to the synchronous driving frequency. The range over which this locking occurs increases as the magnitude of synchronous driving is increased. These experiments clearly show the sharp transition between entrained (blue) and asynchronous (red) regimes indicative of a bifurcation. Most notably for our biological interpretation of these dynamics, the wingbeats near the transition boundary between synchronously

entrained and asynchronous oscillations outside show extremely disrupted peak-to-peak amplitude. This can be seen in Extended Data Figures EDF6, EDF7, and EDF8.

Figure EDF8: Synchronous entrainment of asynchronous oscillations experiment. a) Emergent wingbeat angle versus time in experiments with a combination of asynchronous and synchronous actuation. The four plots correspond to varying the synchronous drive frequency compared to the emergent asynchronous frequency. The synchronous drive is increased from curves 1 through 4 as shown by the arrows in (b). As the synchronous drive frequency gets closer to the asynchronous frequency the wing motion exhibits large amplitude modulations due to a beat frequency between synchronous and asynchronous oscillations. However, when the synchronous drive is close enough the emergent wingbeat frequency entrains to the synchronous drive and the amplitude fluctuations disappear. b) The emergent wingbeat frequency compared to the driving wingbeat frequency. The gray region indicates frequency entrainment where the emergent frequency is exactly equal to the driving synchronous frequency. The three plots are of increasing synchronous forcing magnitude from bottom to top. c) The amplitude fluctuations increase as the synchronous frequency approaches the asynchronous frequency. However, when the driving frequency cross the Arnold tongue and the emergent frequency becomes entrained to the synchronous frequency, then the amplitude fluctuations disappear.

- b. At present, the results are stated in the figure legend as “Transitions are smooth when $f_s = f_a$... but interference causes oscillation amplitudes to drop when $f_s \neq f_a$ ” and similarly at lines 242-245. This needs rewording, because it would be reasonable to expect that small perturbations of f_s from $f_s = f_a$ (i.e. f_s not equal to f_a sensu stricto) will still allow the natural dynamics to be driven at the forcing frequency, which is what the vertical extent of the “bridge” in the other two panels presumably shows. At the least, I would suggest replacing the equality sign with an “approximately equal to” sign to reflect this.

We have reworded this as the reviewer suggests. We have taken the reviewer's suggestion to replace the equality sign with an "approximately equal to" statement. Furthermore, we use words rather than symbols as suggested by Reviewer 3. The updated sentence is below.

Caption of Fig. 3:

"Transitions are smooth when synchronous and asynchronous frequencies are approximately equal (blue and red markers, respectively). However, when the synchronous frequency substantially differs from the asynchronous frequency, interference between synchronous and asynchronous dynamics causes the oscillation amplitude and emergent frequency to fluctuate during the transition regime."

Specific comments:

6. Lines 101-192. "... while maintaining high-power wing strokes and providing a plausible path for the many subsequent shifts between asynchrony and synchrony". An alternative interpretation of

The reviewer's comment here is incomplete, but there was a further comment on this section below. We have revised this sentence and the surrounding paragraph to better link the specific parameters, the bridge in the simulated dynamics, and the biological basis for these parameters (see below). We suspect the reviewer may have been thinking that the bridge also has the corollary of making it hard to smoothly transition if an organism has evolved away from it. We think both are indeed possible and in fact may explain why there are multiple transitions in some orders (e.g. Hemiptera) but not in others (e.g. Diptera, Lepidoptera). We now elaborate this point in the discussion of these clades.

In simulation section:

However clades like lepidoptera, which appear uniformly synchronous, may have subsequently specialized away from the bridge, reflected by M. sexta's location in the model simulation.

In the final part of the paper:

Supporting this, we see multiple asynchronous-synchronous transitions in the earlier diverging orders like Hemiptera (Fig. 1c). This also suggests that hemipterans and other orders with multiple transitions may have muscle physiological parameters closer to the bridge in parameter space, thereby enabling more frequent transitions.

7. Line 47. The meaning of the percentages here needs to be explained, as it relates to some quite involved details of the ASR analysis. If it is not possible to explain succinctly, then I would suggest removing the percentage figures, and leaving the figure to justify the conclusion that "there has most likely been only one evolution of flight muscle asynchrony at the order level".

Thank you for this suggestion. We have changed the parenthetical reporting of percentages to a succinct, but more thorough sentence and call out the longer discussion in the methods section.

“There is an 86% likelihood of a single transition from synchronous to asynchronous in the ancestor of the clade of Thysanoptera + Hemiptera + Psocodea + holometabolous (Node 200) occurring 407 million years ago (Fig. 1c, Extended Data Figure 1-3, Supp. Data Table 1, 2).”

8. Line 48. Consider expanding the sentence “Although asynchrony was thought to have evolved 7-10 times throughout insect flight (2,3,6), only recently has an insect-wide phylogeny enabled interorder resolution” to make clear that the conclusions of Refs 2, 3, 6 are not based on any formal phylogenetic analysis. That is, the reason why the authors conclude the opposite here is because they have done the kind of analysis that was required to draw a conclusion of this kind in the first place; not because the assumed phylogeny has changed from an earlier analysis, or because of some fine difference in method or assumptions.

This is a great suggestion, thank you. Indeed some of those prior references even indicated that the conclusions would be improved with a phylogenetic analysis and so it is good to acknowledge this difference. We have revised the sentence to:

“Although asynchrony was thought to have evolved 7-10 times throughout insect flight^{2,3,6}, earlier analyses were not done using phylogenetic ancestral state reconstruction.”

9. Line 66. the statement “locusts evolved before the first synchronous fliers” would be better stated in terms of ancestral states, especially as “locust” has a specific meaning in relation to swarming that the fossil record does not capture.

Thanks for this suggestion. We have revised the statement to read:

“However, because orthopterans (including the ancestors to modern day locusts) diverged from other insects before the first asynchronous fliers...”

10. Line 90. “We found that the dSA rate, r_3 , in *M. sexta* DLMs fits on the asynchronous scaling line for its wingbeat frequency of 25 Hz”. By no stretch of the imagination does the data point shown by the star in Fig. 2 “fit on the ... scaling line”. For one thing, the line drawn on the figure has not even been extrapolated to frequencies below about 30 Hz, and if it had been, then the predicted value of r_3 would have been close to (and perhaps even less than) zero, rather than of order 100 as shown for *Manduca*. I agree that the datapoint is consistent with the broad scaling relationship shown in the figure, but the data are too variable to describe this as a “scaling line”.

We agree with the reviewer’s more cautious re-wording of the comparison between r_3 and wingbeat frequency. We have changed the text to the following:

*“We found that the relationship between delayed stretch activation rate, r_3 , and wingbeat frequency of 25 Hz in *M. sexta* is consistent with the broad scaling relationship observed by Molloy across asynchronous insects^{17,22} (Fig. 2d, see Methods), which suggests that a hawkmoth could be asynchronous (Fig. 2c).”*

11. Lines 94-96 are key in showing that the magnitude of dSA in hawkmoths is an order of magnitude lower than in asynchronous beetles, waterbugs and flies, because this is what ultimately gives physiological meaning to the parameter K_r in the dynamics analysis. I wonder whether it might be worth making this connection further down when the parameter K_r is introduced in Eq. 1?

We agree with the reviewer that it would help to draw a stronger connection between the measurements of *Manduca* dSA and the modeling parameter, K_r . We have added two sentences immediately following equation one that read as follows:

*“The value of K_r reflects the relative importance of synchronous versus asynchronous forcing in the system. Biologically, a high K_r means that the force change and crossbridge recruitment due to neural activation is large compared to the crossbridge recruitment due to stretch activation. The sensitivity of flight muscle to calcium compared to the stretch sensitivity of the myofilaments gives a plausible mechanism for K_r to vary across species and over evolutionary timescales. Because in-flight measurements of F_a and F_s are unavailable, we approximate K_r as $K_r = F_s/(F_s+F_a)$. In *M. sexta* for example K_r is relatively high (0.88) reflecting that the magnitude of the delayed stretch activation response is low compared to the forces generated via neural activation alone. Asynchronous species produce a delayed stretch activation force several times higher than isometric tetanus²⁶ and would have a very low K_r . By adjusting K_r from fully synchronous ($K_r = 1$) to fully asynchronous ($K_r=0$), we can explore the emergent interactions of synchronous and stretch-activated forcing in the same system.”*

12. Line 190. “transitions are possible, especially on evolutionary timescales that could span the bridge”. Again, it might be worth referring to variation in the dSA magnitude here, to explain what is meant by the currently somewhat vague “transitions ... on evolutionary timescales that could span the bridge”.

Thank you for prompting us to clarify here. Indeed we mean that dSA magnitude (and time scale) could change to enable these transitions.

“Matching muscular and mechanical timescales is evidently a critical requirement for both synchronous and asynchronous power production. However, variation in the strength of the delayed stretch activation response (changing K_r), its timescale (t_0), or the resonant mechanics of the thorax and wings (T_n) could enable smooth, gradual transitions across the bridge, especially over evolutionary timescales. Biologically these parameters will be closely tied to the molecular components of the delayed stretch activation, such as the crossbridging binding, calcium responsiveness, and troponin isoforms mentioned earlier.”

Referee #3 (Remarks to the Author):

Since the seminal work of Pringle, I have not seen a more compelling and exciting mechanistic study of insect muscle function. Having said this, it looks like the model switching between synchronous and asynchronous muscle function underpinning this work has been constructed without a biophysical representation in the context of actual insect functional morphology. Reading the manuscript and its mathematical basis in detail, I concluded that this is indeed the case and cannot be resolved as it is a central assumption in/simplification of the model. Based on the literature available and given the complexity of this system and the new mathematical framework the authors present, the best revision of the manuscript maybe to point this out explicitly. Further, the authors could replace the current speculative last paragraph(s) of the manuscript (that do not strengthen the manuscript) with new ones discussing how the actual embodiment of their model could be established by future insect muscle research. The new last paragraph should give some specific tangible pointers for this future work motivated and supported by specific findings of your research. That would address the biggest weakness of the current write-up of the otherwise visionary research and make it more robust as a Nature paper. As the authors probably realize themselves, their approach has several weaknesses that can be easily criticized because they do not present the necessity/logic for these simplifications clearly in the manuscript. By being open about this in the concluding paragraphs and more informative about why their assumptions and approach is the best way to make progress given the missing information in the literature, this otherwise visionary manuscript will more robustly influence the work of colleagues in the field and generate more citations accordingly the next decade.

We thank the reviewer for their insightful comments and suggestions. We have entirely rewritten the last subsection of the paper. As suggested we have provided a roadmap for some future work, especially as the model connects to biological parameters. We have made some changes throughout the rest of the text to connect these ideas and cite them below in our point-by-point response to the detailed comments. We also maintain the conclusions that can be drawn from the current phylogenetic, physiological, simulation, and robophysical modeling work.

A somewhat smaller, but pertinent, modelling limitation that is not outlined in the manuscript is that neither the mathematical model, nor the robophysical model, nor the micro flapping robot model includes a mechanistic model that explicitly translates muscle/actuator activation spikes into the muscle/actuator dynamics. There seems to be / is no asynchronous versus synchronous neural drive model based on the representative schematic insect activation pattern / pulse train shown in figure 1a. This is a large simplification that should be clear from the start in the manuscript and repeated in the last paragraph with the outline of a roadmap for future work that resolves this: including realistic activation patterns that results in the emergent dynamic behavior presented here.

This is a great point and we agree that ultimately constructing a detailed model, from spikes to wingbeats, is and has been a long term goal in insect flight studies. However, this is also an incredibly challenging if not impossible task given what is currently known about insect muscle

physiology. One advantage of our approach is that we can build on existing data (e.g. characterization of delayed stretch activation) to model the two major contributions of force to the flapping flight (calcium and stretch activation). There does not exist a current muscle modeling framework that can fully predict these forces from stimulation and strain parameters, in part because the molecular details of stretch activation are still being worked out.

Our dynamics model captures a minimal representation of the muscle output, and thus is suitable for studying how synchronous and asynchronous actuation can interact when present in the same muscle. This first of its kind modeling and experimental effort reveals bountiful dynamical insights into how asynchronous flight may have evolved. However, our model is not suitable for addressing questions of how these different actuation modes are controlled neurally.

Thus, we agree with the reviewer that we do not make a detailed model of muscle activation that maps from neural activity to force. We add the following text when we introduce the modeling approach in the Methods:

“To study how delayed stretch activation produces wing oscillations we needed to generate a feedback model for delayed stretch activation. No current detailed muscle model can predict both neural and stretch-activated force components under general dynamic conditions, in part because of limitations in our understanding of the multiscale interactions in muscle⁷². Thus, to model asynchronous force, we do not try to build a detailed molecular model that can predict force from arbitrary activation and strains. Instead, we seek to capture the basic functional input-output relations for delayed stretch activation between an imposed strain and the resulting force.”

Please find my detailed comments below, in the interest of reducing my review writeup time I am listing them trusting the authors will make the effort to understand their merit and resolving them productively. They may come across as quite critical, but both the authors and I know that any study trying to advance our mechanistic understanding of insect muscle will encounter most of them. Hence, I see this manuscript as a mechanistically substantiated roadmap that can guide the field for the next decade if the authors make the effort to point these issues out and write more balanced discussions and conclusions. The order of the comments is mostly chronological as I read the manuscript page by page.

Finally, please note none of the comments are for me, they are for helping the Nature readership understand your work. So please do not give me an explanation while not giving one for the reader in the manuscript. I am not debating the validity of your research; I am giving pointers for substantiating it fully and making the work clear for the entire multidisciplinary readership.

1) Throughout the manuscript no acronyms should be used. Even if wording is often repeated, it impedes the multidisciplinary readership of Nature because the reader is forced to memorize something that is at best field specific and generally unimportant to understanding the science when it is simply written out. Excellent writing makes acronyms obsolete and shortens the manuscript simultaneously. Acronyms should also not be used in figure captions and in the figures

themselves, variable symbols should be replaced with words representing what is plotted whenever possible: I noticed this was possible in virtually all figure panels, there is sufficient space along the axis. This makes the manuscript much more readable to all muscle physiologists, neuroscientists, biomechanists, evolutionary biologists, entomologists etc. who will all find this work of interest. It also helps physicists and engineers not familiar with this type of biophysical research and the associated nomenclature.

Thank you for prompting us to improve the readability of the manuscript. We have taken most of the reviewers suggestions below and noted a few exceptions (see specific responses). Moreover, we recognize that these were meant as examples not as a comprehensive list. We have taken a careful pass at the manuscript, and especially the figures, to make the choice of notation and labeling as clear as possible.

---Examples:

"dSA" I found it a drag to read and memorize, unhelpful.

We have reverted nearly every reference to dSA to the full term, delayed stretch activation, except where needed for labeling figures.

Fig 1b "X" "F" are unhelpful etc.

We have fixed these to "stretch" and "force" respectively. We use stretch because the muscle physiology literature typically defines force as shortening in the positive direction and so strain should also be positive in the shortening direction. We clarify that stretch is negative strain and the sign conventions when we get into the specific measurements and models in Fig. 2 and 3. "Stretch" makes this simpler in the first figure.

Fig 2, the labels 2c,d,e vertical are cryptic: but notice the horizontal labels are great (except for in 2e dSA and Tet., Twitch is good again).

Agreed, we now use words and leave the variables parenthetically. We also clarify more in the caption.

Fig. 3 all labels. Variables in a,d are helpful for introducing them but fonts are small and it would be great to also add a word explaining them.

Etc. I am not repeating these below in the figure comments, they are all centralized here.

Agreed, we have increased font size and simplified the notation a little. We have also clarified in the caption and make the notation fully consistent with the revisions in the methods and supplement.

2) "found that there has most likely been" caveats should be outlined in the methods and the reader should be alerted with "(but see methods)"

e.g. since entomologist typically estimate only 10% of insect species have been discovered and of those about a million species almost none have been studied there are caveats. Also, the reconstruction of the phylogeny tree is based on selected sequence snippets that do not code for muscle properties as typical for almost all phylogenetic studies. These are limitations for phylogenetic studies generally, and acceptable according to several fields, but they need to be listed.

Thank you for prompting us to be more clear in the phrasing. We used the term (most likely) in the maximum likelihood sense, but appreciate that this could be confusing and not fully capture what can and cannot be concluded from a phylogenetic basis. We have addressed the sampling limitation of this (and most comparative studies) in the discussion of the phylogenetic comparative results:

“Ancestral state reconstruction can change with more sampling and different phylogenetic reconstructions, but the current best evidence supports a single origin of asynchrony at the order level. Most importantly, our analysis raises the possibility that the physiological properties associated with asynchrony, such as delayed stretch activation, could be conserved in secondarily synchronous fliers.”

Prompted in part by Reviewer 1’s comments and some of the comments further below, we have significantly expanded our analysis of phylogenetic reconstruction assumptions and a discussion of that is now in the first section discussion and in the Supporting Information (also see response to Reviewer 1).

When reconstructing traits it is often advantageous to have the phylogeny based on the sequences that are not functionally linked to the trait, so that the reconstruction is independent. So we do not view that particular point as a limitation (rather it is probably a strength). Since we did not construct the phylogeny, we leave the specific details of that reconstruction to the referenced work on which it is based. However, we do more clearly indicate the basis of the phylogenetic reconstruction:

“We used a previously published molecular phylogeny grounded in fossil records spanning all insect orders¹⁴, which modifies the fossil calibration of the extensive insect phylogeny developed by Misof et al.¹⁷.”

3) " These patterns suggest that after the evolution of asynchrony, the physiological properties associated with asynchrony, such as dSA, could be conserved in secondarily synchronous fliers. " it just shows the most likely scenario supported by the particular assumptions underpinning the phylogenetic tree construction and the way evolution was propagated. This is fine, but not a certain statement, the manuscript is not clear on this. The biophysical framework in combination with the phylogenetic tree are an important advance regardless.

We thank the reviewer for their careful parsing of the uncertainty of (any) phylogenetic reconstruction with the role it plays in our conclusions. Ultimately we use the phylogeny to prompt the biophysical investigation and the evidence of secondary synchronous fliers really comes from a combination of the phylogenetic and physiological (Fig. 2) evidence, bolstered by the feasibility and biophysical framework in Fig. 3. Our expanded discussion of the basis for ancestral state reconstruction, alternative models of evolutionary rates, and what patterns are consistent across the phylogenetic reconstructions help provide this nuance. In addition, we now make it more explicit that phylogenetic reconstructions still provide only an inference of the past patterns of evolution in newly added analysis we provided in response to Reviewer 1's.

4) Figure 1:

1a) what parameters are plotted, show horizontal and vertical axes

1b) color parameters according to trace. Do we need parameters here, these axes should be labeled with informative names to increase accessibility. Add a horizontal scale for all these traces.

Caption:

- "The evolutionary history " should be "Current..."

- "actuation reveals " should be "suggests"

- It is great style that the acronym is introduced again in the caption of the first figure, but a readable nature paper in our small multidisciplinary field requires writing without acronyms and maintaining length constraints by cutting non-essential text of which there still is quite a bit (e.g. the last few paragraphs are highly speculative and not the most informative given the new research road map this new model dictates, which is more worthy to outline briefly in a single last paragraph).

We have implemented most of these changes. The traces are illustrative so there is not a distinct horizontal scale in 1b, but we have clarified the axes. We have fixed the first sentence to indicate that it is a phylogenetic analysis rather than "evolutionary history" per se, and add the modifier "likely" to indicate that there is still uncertainty. To use "suggests" means more indirect evidence, whereas here we provide a conclusion based on the based evidence available.

5) Figure 2:

2c) the legend is unclear and not all traces indicated, what is what?

We have fully revised the caption as well as the figure labels to clarify the traces and use more accessible language. In Fig. 2c we show a magnification of the experimental data from 2b. Namely this is the delayed stretch activation from hawkmoth muscle. The red line is a fit to the mathematical model of delayed stretch activation.

Caption:

-replace acronyms with text (also DLMs, it is unreadable)

We have enlarged the text in relevant areas. We have replaced the acronym DLM throughout the main text, except for one parenthetical reference so that specialists know precisely which muscle we mean when we say downstroke flight muscle. We use the abbreviation DLM in the methods

only to describe the experimental preparation and we have added a definition of the abbreviation at the first instance of usage in methods.

- r_3 is poor naming since it is the wing radius variable for aerodynamic power required to beat the insect wing - rename or don't use a parameter symbol and instead write out the name. This is a problem throughout that needs to be resolved systematically.

This is challenging to address for the following reasons. While the variable name r_3 may refer to a wing morphological parameter by flight biomechanists (See Ellington etc.), it also refers to the rate constant of delayed stretch activation in the muscle literature. The first reporting of r_3 for delayed stretch activation is as far back as the 1950's.

To keep with the well established norms in the study of asynchronous muscle we have opted to keep the usage of r_3 to refer to delayed stretch activation rate. This is consistent with other studies both recent and past, as well as studies published in *Nature* (see reference 17 and reproduced below).

17. Molloy, J. E., Kyrtatas, V., Sparrow, J. C., & White, D. C. S. (1987). Kinetics of flight muscles from insects with different wingbeat frequencies. *Nature*, 328(6129), 449–451.

However, we appreciate that this is potentially confusing and in the prior version r_2 was used in the aerodynamics equations. We have revised the aerodynamics equations in the methods to use different notation and to explicitly note where this is non-conventional to allay confusion. Ultimately the only way to deal with overlap notation convention in these interdisciplinary papers is to be explicit and we thank the reviewer for prompting this clarification.

- " previously-established linear relationship between " provide reference here.

While we had provided a literature reference (19), we now make it explicit that this is a prior empirical finding by referring to the paper and the author's in the caption text. The new sentence is provided below:

"Despite being synchronous, M. sexta's delayed stretch activation rising rate constant, r_3 , lies near the prior empirical finding of a linear relationship between r_3 and wingbeat frequency by Molloy¹⁹"

6) " The timing of the dSA response are typically characterized by fitting a sum of three exponents¹⁷ representing three phases: " show in a figure, fig 1b would probably be good fit.

Thanks for this suggestion. This fit is actually the fit we show in Fig. 2c, but we appreciate that this was not explicit in the sentence so we now cite the figure parenthetically here.

"The timing of the delayed stretch activation response is typically characterized by fitting

a sum of three exponential terms with rate constants r_2 , r_3 , and r_4 ¹⁹ (Red curve Fig. 2c, and see Methods)."

7) " where the spring represents parallel elasticity from the deformable exoskeleton " is this a linear model, clarify here. What is the justification for assuming a linear thorax spring system? The reader needs it.

The justification for using a linear thorax spring model arises from several studies this author group has previously performed on *M. sexta*. In these studies we performed physiologically relevant displacements of a thorax at the point of muscle insertion, and we measured the force on the opposite side of the thorax also at the point of muscle insertion. We found that the thorax response is well approximated by a linear elastic model, with large enough stiffness to aid in flight energetics. We very much agree that this motivation should be clear to the readers. We have included reference to the first of this work in the manuscript, reference 38 reproduced below. We have also added a justification after the introduction of the body mechanics equation as follows:

*"This equation captures the indirect actuation of synchronous and asynchronous insect flight muscle which act via the deformation of the thorax in parallel with the muscle to sweep the wings back and forth. Measurements of the *M. sexta* thorax are well approximated by a linear elastic spring in parallel with muscle³⁸."*

Reference:

38. Gau, J., Gravish, N., & Sponberg, S. (2019). Indirect actuation reduces flight power requirements in *Manduca sexta* via elastic energy exchange. *Journal of the Royal Society, Interface / the Royal Society*, 16(161), 20190543.

8) " we modeled dSA force as the convolution of the stretch-hold-release-hold response with velocity " please define for a general audience: e.g. wikipedia is a lot clearer: the integral of the product of the two functions after one is reversed and shifted. Why is a convolution model appropriate, what is the logic? Please include it concisely. In the discussion of results, are there limitations of this model for the flight muscle apparatus? In terms of solving the underlying coupled differential equation, please clarify. Equation 3 is a lot clearer than writing about a convolution since that is just the solution of this particular system with its assumptions.

This is a good point and we thank the reviewer for suggesting better clarification for an interdisciplinary audience. We simplify our description of a convolution using the concept of a filter, which gives the general reader an intuition about what stretch activation is (a force response that is a filtered version of the stretch). When we introduce the mathematical form we now indicate that this filter comes directly from the physiological experiments and can transform a strain trajectory into the result stretch-activated force response. The limitations here are that a convolution works in the linear regime, which has been shown to hold for small amplitude stretch activated conditions. We now make this assumption explicit when introducing this approach.

The last part about connecting to the underlying coupled differential equation we now address in the Supplementary Information where we express the convolution as a differential equation. The differential equation form of delayed stretch activation can be useful for analyzing when oscillations occur in parameter space. Critically, the convolution and differential equation representations are identical and produce identical results—they just provide different intuition for the system where convolutions are often used to represent input/output responses like that of delayed stretch activation. So we keep the convolution formulation in the main paper.

The following text in the main document has been added:

“We developed a model of delayed stretch activation that captures the strain-dependent force output of asynchronous muscles (see Methods). Delayed stretch activation can be modeled as a filter (convolution) that transforms strain rate into force through a velocity impulse response function. Furthermore, the force response to a stretch-hold-release-hold (step) strain input approximates this response function to an impulsive stretch. This filter can then be applied to continuously varying patterns of strain (such as during a wingstroke) provided it remains in the linear regime (See Supplementary Information IIB). Prior experiments have demonstrated that for low amplitudes the delayed stretch activation response was linear⁴⁰, justifying this approach. We fit the convolution filter to the stretch-hold-release-hold response as $F_{\text{async...}}$ ”

Reference added:

40. Jewell, B. R., & Ruegg, J. C. (1966). Oscillatory Contraction of Insect Fibrillar Muscle after Glycerol Extraction. *Proceedings of the Royal Society of London. Series B, Biological Sciences*, 164(996), 428–459.

9) " The muscle strain rate scales with wing angular velocity by a factor of LT , where L is the resting length of the muscle, and T is the transmission ratio, $LT \dot{\epsilon}(t) = \dot{\phi}(t)$ ". Where does this come from and how come the transmission ratio is a factor here. Given this is strain rate, non-dimensional length change rate, how do we get per second? Aren't there scaling parameters missing here. Especially considering the transmission ratio is wing stroke amplitude / muscle amplitude. This should be resolved for any reader without assuming knowledge about traditions. Also, in some fields this is considered wrong/erroneously and unphysical so keep in mind such idiosyncratic traditions do not cross multidisciplinary borders because they have no generalizable logic.

Thank you for the suggestion to make this clearer. The muscle physiological experiments are in respect to muscle strain whereas the equations of motion are with respect to the angular movement of the wings. The reviewer's intuition about needing scaling is spot on, we need to go from a strain rate to angular velocity. This is done by multiplying the strain by the length of the muscle and then by the transmission ratio which is the ratio of the angular displacement of the wings to the linear displacement of the muscle. L has units length, and T has units of

radians/length so the physical units of the resulting quantity remain the same (“per second”). We appreciate that the larger point here is that this was not clear to the reader and so we have revised our description of this scaling to reflect the point above.

In the main text we now say:

“The muscle strain rate scales with wing angular velocity by a factor of LT , where L is the resting length of the muscle, and T is the transmission ratio of angular wing displacement to linear muscle displacement, $LT \dot{\epsilon}(t) = \dot{\phi}(t)$.”

A little lower we provide a reference for this value for *Manduca sexta* (Gau, et al. 2022) and in the supplement we again have now clarified what T is in the description of the hawkmoth simulation.

10) Figure 3:

1a) symbols too small; is F_{aero} applied at r_2 (aka the radius of gyration in insect wing aerodynamics, the radius where the net force acts)? If not resolve everywhere and clarify where the force acts in your model.

The aerodynamic force is applied at the center of pressure of the wings. Because the wing is simplified in the figure, we replace F_{aero} with the aerodynamic torque and point to the methods in the caption. In the methods, in equations 11-13 we define the aerodynamic torque as the aerodynamic force acting at the center of pressure (l_{cp}), with l_{cp} taken from the literature. In the main text we have updated the sentence to describe this as follows:

“To incorporate these interactions, we first modeled aerodynamic damping using a quasi-steady approximation with aerodynamic torque equal to the wing angular velocity squared, multiplied by a coefficient (Γ) that accounts for wing shape, air density, and experimentally measured drag coefficients²⁴ (See Methods).”

1b,e) unusual power units that are unhelpful, why not Watts or non-dimensional?

Agreed, we have replaced these with real units.

1c,f) the short blue segment on the color bar is really unclear and unexpected/unusual and thus confusing to most readers. Why not plot a more traditional plot with different curves for different f/f_s values so we can see which combinations give 1? Also, how close do we need to be to one? Are the values deviating from 1 far away, it seems most of the space has the frequency match, but it is the power that is the issue.

This is a good question and one raised by Reviewer 2 as well. The blue portion of the simulation represents the region where the emergent frequency is equal to the synchronous frequency within the time-precision of our simulations. To help better visualize the emergent wingbeat frequency we have provided a new Extended Data Figure (EDF6) which shows the Fourier transform of the emergent wingbeats at different locations in parameter space. In addition this figure includes a

continuous colormap to better visualize the frequency jumps that occur as the boundary is crossed.

To answer your question of “how close do we need to be to one?”, this depends on the value of K_r . When the synchronous force is low (small K_r) then the frequency difference between the synchronous and asynchronous regimes is small along the bridge. This is also because the bridge precisely emerges at the location where the asynchronous frequency (when $K_r = 0$) exactly equals the synchronous frequency (See new Extended Data Figure EDF5) . However, as K_r increases, the frequency difference between synchronous and asynchronous modes becomes much larger and more importantly, these modes are separated by a region of very large and erratic amplitude fluctuations on the boundary. We have included a new Extended Data Figure (EDF7) which shows these amplitude fluctuations and how they peak at the boundary and then disappear in the synchronous mode.

Thus, the division between synchronous and asynchronous regimes not only represents a change in emergent frequency, but also a regime of highly erratic wingbeats. We have added the following text that describes this phenomena:

“As we move away from the bridge along the t_0/T_n axis, there is a bifurcation and the the asynchronous and synchronous frequencies diverge, ending the entrainment, and leading to emergent asynchronous oscillations (Extended Data Figure EDF6). Crossing between these two regimes leads to significant interference between these oscillatory modes, thus leading to lower power, less smooth flapping trajectories that are unsuitable for flight (see Extended Data Figure EDF7 & EDF8).”

1h,1) clarify color code, looks like this is blue synch and red asynch and that the transition is from one to the other via the gain. Again, how is the gain controlling mode switching implemented in the robot? Clarify, I only understood after reading the methods that basically the actuator is controlled with a voltage puls generated by the model in 1a: Meaning that this is not an emergent property and simply a playback of the model in a robot showing the model ignores friction in the robot and possibly in the insect. Because the robot does not embody the model in 1a)

In the figure panels h,i, the color represents the value of K_r which governs the ratio of synchronous and asynchronous forcing. To clarify, the wingbeats in this robot are emergent from the combination of synchronous and asynchronous dynamics—we are not just playing in a feedforward signal. Instead we have implemented a real-time simulation of delayed stretch activation, we measure the wing velocity in real-time and we simulate the dSA force-response in real time. Thus, the wingbeat frequency when $K_r = 0$ is truly emergent and arises from the delayed stretch activation feedback. We have clarified this point in the figure caption and in the manuscript.

In figure caption:

“The blue and red colors indicate the synchronous (blue) and asynchronous (red) regimes.”

In main text”

“To generate delayed stretch activation in the robobee we used a fiber-optic displacement sensor to estimate wing velocity. The instantaneous wing velocity was supplied to a real-time delayed stretch activation dynamics model (the same as for the robophysical system with parameters adapted for the robobee) and the output of this model was converted to a voltage that was amplified and supplied to the piezoelectric actuator. Thus, we were able to establish a real-time feedback loop between wing velocity and actuator voltage that could generate asynchronous wingbeats of the robobee. By combining the outputs of feedforward synchronous actuation, and feedback delayed stretch activation (Eq. 1) in real-time experiments we were able to demonstrate transitions between synchronous and asynchronous oscillations in this robot.”

It is also not an independent proof of its validity, it just shows the model can control the flapper feed-forward. Yet this is nowhere clear in the manuscript and even somewhat hidden in the methods. This should be communicated upfront in the caption and in the main text with a reference for details in the methods. The methods should be revised to be clearer on this as well. The final paragraph should outline how the model could be embodied in the robot, so the behavior is naturally emergent from the electromechanical system in closed loop.

As we noted above, both the larger and smaller robots are operated with delayed stretch activation through wing velocity feedback. We are emulating the delayed stretch activation force through sensors that instantaneously measure the wing velocity, and actuators that convert the real-time computed delayed stretch activation force into the appropriate motor torque (for larger system) or piezoelectric actuation force (for smaller system). Thus, both robot systems are demonstrating feedback generated oscillations from delayed stretch activation in real-time, and the output oscillations are emergent from this electromechanical feedback loop. The above text clarifies this point. Furthermore, we have revised the methods to better explain how this feedback is implemented.

Caption:

-“ stretch activated feedback via a dSA dynamics model (Eq. 8-11), and combined synchronous and asynchronous muscle force with “gain knob” K_r ” make specific what the parameter is and captures. The definition used in the methods should be written here since it is central to the scientific idea that this paper discusses: “ K_r describes the ratio of synchronous and stretch-activated forcing”

A critical aspect missing in this manuscript is a biophysical explanation for what K_r actually is and how it is represented in the flapping wing apparatus. If this is not known, it is just a model that shows that by linearly combining async and sync forcing both can be represented. Given the same knob is used in the experiments it would be key for proving the main idea to pinpoint this mechanism in the insect muscle physiology.

Thanks for encouraging us to make this more clear and to better connect the model parameters to the relevant biology. Our evolutionary analysis and muscle physiology indicate that synchronous insects can also possess a delayed stretch activation response. From a biophysical perspective, the variable K_r represents how much delayed stretch activation response is present (its magnitude), with respect to the synchronous forcing that may also occur. While an exact measurement of K_r requires measuring dSA under flight conditions, we can approximate it through our muscle physiology experiments and we have included this in the manuscript. As mentioned in our other response the exact molecular basis of dSA is still not agreed upon, but we do know some of the likely players. The number of myosin heads that are stretch activated compared to those that are calcium activated gives a natural connection to K_r . There is also prior work showing that the ratio of troponin isoforms (some of which are thought to be stretch activated) correlates with asynchronous force. We have clarified both how we make the approximation of K_r by bringing the equation into the main text and clarifying our derivation in the methods. We also added a discussion about how the asynchronous and synchronous forcing could vary in magnitude in the biological system. This would be the biological basis for changing K_r :

*“It is not precisely known what mechanism controls the magnitude and rate of the delayed stretch activation²⁷, but it is dependent on calcium levels and likely involves recruitment of additional myosin heads (crossbridges) through stretch-sensitive myofilament proteins^{22,28}. In asynchronous insects neural activation typically only recruits ~30% of crossbridges which explains why the stretch activation can far exceed tetanic activation²⁹. One possible way this is regulated is by the ratio of isoforms of the regulatory molecule troponin which promotes release of myosin-binding sites. This ratio is correlated with asynchronous force output^{25,30}. Surprisingly, the stretch-activated troponin isoforms found in asynchronous insects and implicated in delayed stretch activation are also found in *M. sexta*^{30,31}. This provides one possible mechanism for residual delayed stretch activation in moths. Our physiological results indicate that delayed stretch activation can be present in quite reduced magnitudes and it is already known that the rate constants can vary widely¹⁹. The flight muscles of different asynchronous (and synchronous) orders may have further specialized, especially in extreme cases of performance which may contribute to the molecular differences observed in some groups³²⁻³⁴. Our results show that conserved molecular components are potentially part of the same continuous dynamical parameter space that spans across synchronous and asynchronous flight modes.”*

And later in the discussion of K_r :

“Biologically, a high K_r means that the force change and crossbridge recruitment due to neural activation is large compared to the crossbridge recruitment due to stretch activation. The sensitivity of flight muscle to calcium compared to the stretch sensitivity of the myofilaments gives a plausible mechanism for K_r to vary across species and over evolutionary timescales.”

Also, it is unclear if the insect aeromechanics dynamical system is typically a periodic limit cycle system, there is evidence for it being less regular or even chaotic (strange attractor) in some flapping wing regimes. I am not concerned about this, but this context would be valuable to add in the road map for future work.

We agree, the coupled dynamics of synchronous forcing, delayed stretch activation, and spring-wing body mechanics may permit more exotic dynamics such as chaos. We address this point in the next comment.

11) " as a quasi-steady term proportional to velocity squared, " fine but do know the coefficients can vary during the cycle and that they don't always fall on a limit cycle. They can also fall on a strange attractor. I don't think this invalidates the idea in this paper, it is just that more research will be needed in the future as usual at the cutting edge.

This is a great point and one we are eager to study in future work. Aerodynamic coefficients may vary in time due to wing pitch/deviation, or through aerodynamic phenomena such as shedding leading edge vortices, or other phenomena. The interaction of these time varying coefficients and the synchronous + asynchronous dynamics is an extremely interesting next step. Furthermore, we think it is quite interesting that even without these complex models the interactions of synchrony and asynchrony exhibit erratic behavior. We have included a sentence to acknowledge these facts and to cite a new relevant reference that highlights possible strange attractor and limit-cycle behavior in insect flight:

"Thus, while complex aerodynamics phenomena⁴⁵ and sensory-motor feedback systems⁸ can exhibit unpredictable flapping wing behavior, our results indicate that even simplified fluid and body mechanics under combined synchronous and asynchronous actuation are sufficient to lead to erratic wingbeat dynamics."

Reference added:

45. Lentink, D., Van Heijst, G. F., Muijres, F. T., & Van Leeuwen, J. L. (2010). Vortex interactions with flapping wings and fins can be unpredictable. *Biology Letters*, 6(3), 394–397.

12) "(T) that transforms linear muscle displacement into rotational wing movement. " define in the main text since its an important factor.

Agreed, we have now specified that this is just a ratio of angular movement to linear displacement in the main paper and cited the relevant paper for this measurement.

"T is the transmission ratio of angular wing displacement to linear muscle displacement, $L T \dot{\epsilon}(t) = \dot{\phi}(t)$. Values of L and T are taken from the literature^{41,42}."

13) "rates, we reparameterized r3 as to, which" reparameterization is completely unnecessary and highly confusing and thus unacceptable in a research paper. Choose one distinct

parametrization and stick with it. Also, if there are difficulties with this, refer to methods and provide clarifications for specific choices there.

Thanks for catching this confusion. It is not really a reparameterization and we have revised to more clearly indicate that we do this to combine multiple time constants (that are empirically related in prior work) to simplify the parameter space. This also lets us use t_0 which is an easier value to read off of the plots (the rise to peak asynchronous force peak). We have revised these sentences too:

“To reduce the complexity of the delayed stretch activation model we combine (r_3 , r_4) to one time scale t_0 (see Methods), which is the rise time to peak force (Fig. 1b). To compare across systems we then nondimensionalize this time by normalizing to T_n , the resonance period of the wing-thorax system. T_n is determined by the mechanics alone ...”

14) "which is the time delay to peak dSA force (Fig. 1b)." visually indicate the time parameters in 1b and use normal wording not only variable names, so the graph can be read by all Nature readers.

We have fixed the notation in Fig. 1b and now say that t_0 is the rise time to peak force in the caption.

15) "scales. T_n is determined by the mechanics alone (l , kl , and Γ ; see Methods)." give eqn here, since it the characteristic frequency / representative period of the system (depending on the most useful perspective to explain the main point of the paper to all readers).

We have added the equation to the main text.

16) "a narrow ratio of muscular and mechanical time constants" clarify: what is the critical constraint mathematically and how does it relate to muscle mechanics and aeromechanics.

Thank you for encouraging more clarity here. The new analysis and experiments we have included help address this. From the examination of the $K_r = 0$ dynamics we observe that the bridge appears at a value of t_0/T_n where the emergent frequency exactly equals the synchronous frequency. Extended Data Figure EDF5 shows this phenomenon. This makes sense because the bridge should occur when the asynchronous and asynchronous drive produce frequencies that do not greatly interfere with one another.

Figure EDF5: **Emergent frequency (f) normalized by the synchronous frequency (f_s) in the asynchronous regime ($K_r = 0$).** As the time to peak force from delayed stretch activation (t_0) is increased the system undergoes a bifurcation in which steady wingbeats emerge. Dashed line is the prediction of this critical t_0/T_n from analysis of a linearized system. The emergent wingbeat frequency in simulation (circles) decreases as t_0 is increased. A linearized analysis of Equations 1 & 2 (See SI for details) is able to capture the emergent wingbeat frequency. The colormap of simulation points matches the heatmap of Fig. 3c,f. The bridge emerges when the emergent frequency from delayed stretch activation matches the synchronous frequency.

As the synchronous drive magnitude is increased (K_r is increased) the bridge widens. The regime where the wingbeat frequency matches the synchronous frequency is surrounded by states with very erratic and fluctuating wingbeat motions, this can be seen in the new Extended Data Figure EDF7. This figure plots the peak-to-peak amplitude fluctuations on the bridge, and slightly off the bridge, from simulation results. The wingbeat motion is very erratic in steady-state when the parameters are not on the bridge.

Figure EDF7: **Wingbeat amplitude variation in Manduca simulations.** a) The three plots show the wingbeat angle versus time at three different values of t_0/T_n , for a constant $K_r = 0.2$ (shown by the white line in the heatmap). After transient oscillations die out we measure the amplitude of wingbeat peaks as a function of time (positive peaks are shown as red triangles). The top plot is an example in the asynchronous regime, displaying moderate amplitude fluctuations. The middle plot is the wing angle within the frequency locking synchronous regime (on the “bridge”) where the wing amplitude is steady. Lastly, the bottom plot shows the wing motion in the asynchronous regime below the frequency locking “bridge”. b) For all combinations of K_r and t_0/T_n we calculated the standard deviation of the wingbeat amplitudes which we show as a heatmap. Brighter regions of the plot correspond to where large stroke to stroke amplitude variation occurs (i.e. top and bottom plots in panel a). When the wingbeat is steady the amplitude variation is small these appear as the black regions. The boundaries between the synchronous and asynchronous regimes exhibit large amplitude fluctuations, while the bridge connects the synchronous and asynchronous regimes with smooth sinusoidal emergent wingbeats.

The location of this bridge is, as we noted above, determined by the emergent asynchronous dynamics (i.e. the time and magnitude constants of delayed stretch activation, and the body stiffness and wing inertia). Since our new analysis of the linearized system (see added Supplementary Information) matches well with the fully nonlinear system (Fig. EDF5), this shows that the aerodynamics forces don't determine the location of the bridge at $K_r = 0$ since there are no aerodynamic forces in the linear system. As for the narrowness of the bridge this is likely associated with the dissipation from aerodynamic force, balanced by the energy input from both synchronous and asynchronous contributions. Some insight into this can be gained from new simulations at varied delayed stretch activation and body mechanics parameters (see Supplementary Information) where these parameters can affect the location and shape of the bridge (although a bridge always exists). An exact mathematical treatment of this is challenging even to this day in the field of nonlinear dynamics; however this will be something we examine in future work.

17) "Although changing K_r is the most obvious transition, moving along the t_0/T_n axis can also drive the system from synchrony and asynchrony, but the power output quickly diminishes in either direction." draw both paths in the model figure: 3a-c

We have removed this sentence for clarity and thus do not include these paths in Fig. 3. Our new analysis and descriptions in the text now clearly illustrate that varying t_0/T_n at a constant K_r (i.e. moving vertically in Fig 3a-c) results in crossing regimes with highly erratic wingbeats. Thus this is likely not a viable path for transitioning between synchronous and asynchronous modes. Moreover we want to avoid too much clutter on the already busy plots.

18) "Around $t_0/T_n \approx 0.2$, we find" is this just math? Can this be predicted? What does this tell us about the interaction between muscle mechanics and aeromechanics? Clarify for the reader.

Indeed there is a mathematical prediction of where the bridge will emerge at. Our new linearized analysis provides this prediction, however we do not report the exact equation as it must be determined by the eigenvalue of a 4x4 state matrix and the equation takes up an entire page. Hence, not much insight is gained from the formula. However, the calculated wingbeat frequency from the linearized analysis matches the simulation extremely well, and shows that the bridge emerges exactly where the emergent frequency arising from asynchronous driving equals the synchronous wingbeat frequency. See figure in comment 16 above.

If we change other parameters in the model the bridge can move around but it is always where the emergent frequency of the (self-excited) asynchronous system matches the frequency of the synchronous system (the driving frequency). For example below shows simulations in supporting information where we varied the body elasticity and synchronous force, to illustrate that a bridge still exists.

FIG. 3. **Comparison of simulation results with and without active muscle elasticity.** Top row shows emergent frequency and bottom row shows the standard deviation of peak-to-peak amplitude. White lines designate boundary between asynchronous and synchronous modes. Both simulations display the same fundamental phenomena: 1) asynchronous oscillations emerge as the normalized time-to-peak of the delayed stretch activation response (t_0/T_n) increases, and 2) a bridge of smooth sinusoidal wingbeats exists between the asynchronous and synchronous modes of actuation.

19) "especially on evolutionary timescales that could span the bridge, while" for this we would like to know the biophysical mechanism that represents the knob as well as the presence of both synchronous and asynchronous muscle control. It helps the reader to remind them of the in vivo evidence you found so this doesn't become a purely mathematical construct. If it is just math then write this and don't speculate, if it is unknown, it is future research for your roadmap paragraph.

Thanks for this suggestion. We incorporated changes here along with those to a similar comment from reviewer 2. We now refer directly to the parameters that could move across the bridge and connect them to possible molecular and cellular properties of the muscle that could change over evolutionary timescales. While multiple mechanisms have been implicated in dSA, the necessary and sufficient molecular components at play are still debated, but we can still illustrate the point.

20) "subsequent shifts between asynchrony and synchrony (1c)" are you thus suggesting there exist insects today (given there are ~10 million species) that are on the crossbridge in the middle or near the middle? How could this be tested in future research? Evolution is ongoing, there is no evidence insect muscle evolution converged into a final state.

Yes! In fact we think this is what is possibly happening in Hemiptera, which is woefully understudied and possibly in other orders that have multiple transitions after the origin of asynchronous. We mentioned this briefly in the closing paragraphs, but we have now expanded this statement

"It is likely that highly-specialized fliers, like many dipterans and hymenopterans, have further specializations to enhance asynchronous flight^{27,32,33,54,55}, but these do not preclude a common underlying physiological mechanism for delayed stretch activation which can vary in magnitude and timing. Supporting this, we see multiple asynchronous-synchronous transitions in the earlier diverging orders like Hemiptera (Fig. 1c). This also suggests that hemipterans and other orders with multiple transitions may have muscle physiological parameters closer to the bridge in parameter space, thereby enabling more frequent transitions."

Also our hypothesis for future work is better motivated with the increased discussion earlier in the paper about how the physiological parameters could change. Please refer the response to other comments in this review about clarifying how evolution could move across the bridge and the molecular basis for K_r changing.

21) "To test this hypothesis that insects can realize both synchronous and asynchronous oscillations simply by changing a ratio of timescales and a gain coefficient,"

" ratio of timescales " this can be understood

" a gain coefficient " again how is this represented in vivo? The manuscript needs to be clear on this.

This gain coefficient, K_r , represents the ratio of synchronous force to synchronous + delayed stretch activation force. We now provide this equation directly in the methods of the text and we have added clarification into the paper in the following text.

"Biologically, a high K_r means that the force change and crossbridge recruitment due to neural activation is large compared to the crossbridge recruitment due to stretch activation. The sensitivity of flight muscle to calcium compared to the stretch sensitivity of

the myofilaments gives a plausible mechanism for K_r across species and over evolutionary timescales."

22) "This arrangement mimics the indirect actuation of both kinds of insect flight muscle which act via the deformation of the thorax to sweep the wings back and forth." shouldn't this information come early when describing the differential equation model, the reader is completely surprised when it comes this late.

We thank the reviewer for pointing this out. We have moved the sentence to immediately follow the equation and modified the sentence to:

"This equation captures the indirect actuation of synchronous and asynchronous insect flight muscle which, act via the deformation of the thorax in parallel with the muscle to sweep the wings back and forth."

which now follows the introduction of the spring-wing dynamics equation (Eq. 2) where it makes more sense as the reviewer suggests. We have now modified the section describing robophysical flapper section as follows:

"Unlike previous robophysical investigations of flapping wing flight^{38,39}, we did not directly prescribe wing angle versus time in our roboflapper. Instead, we provided torque commands to a motor that were either feedforward periodic (e.g. synchronous sinusoidal forcing), or velocity feedback generated (e.g. real-time simulated delayed stretch activation) and the wing angle versus time was an emergent property (see Methods). To mimic aerodynamic damping and the body elasticity of indirect actuation the motor was in parallel with a Silicone molded torsional spring driving a dynamically scaled wing³⁵.

23) " when t_0/T_n exceeds ≈ 0.3 , possibly due to the dSA force failing to overcome friction " so how do we know that the high aerodynamic drag doesn't impede this in insects? And doesn't the thorax also have frictional losses? Does the model predict this and may this constrain the discussion? Also, insect flight is energetically so cheap, simply due to the square cube law, that insects do not need to be as efficient. So the assumption that friction plays no role cannot easily be supported. Given the problems the Wood lab uncovered with micro scale mechanism friction and given there is a scaling paper on this issue and how it impacts micro scale flight by Hawkes, this requires clearer information for the reader.

This is a very good point. First let us consider the question of aerodynamic drag. We now understand that wing aerodynamics don't significantly contribute to the loss of amplitude as t_0/T_n because when we linearize the system we disregard aerodynamic forces (because linearization

assumes small amplitude, and thus velocity squared forces are negligibly small). The agreement between the linearized system and the full nonlinear simulations gives some indication that aerodynamics don't limit the generation of oscillations. Effectively the linearized system shows that for larger and larger t_0/T_n wings will always vibrate but the amplitude will just get smaller and smaller. This can be seen from the SI figure below

Figure EDF5: **Emergent frequency (f) normalized by the synchronous frequency (f_s) in the asynchronous regime ($K_r = 0$).** As the time to peak force from delayed stretch activation (t_0) is increased the system undergoes a bifurcation in which steady wingbeats emerge. Dashed line is the prediction of this critical t_0/T_n from analysis of a linearized system. The emergent wingbeat frequency in simulation (circles) decreases as t_0 is increased. A linearized analysis of Equations 1 & 2 (See SI for details) is able to capture the emergent wingbeat frequency. The colormap of simulation points matches the heatmap of Fig. 3c,f. The bridge emerges when the emergent frequency from delayed stretch activation matches the synchronous frequency.

For the second part of the question, energy loss from other sources such as viscous damping in the body, or other frictional type mechanisms can result in an upper limit on the oscillation amplitude even in the insects. This motivated us to run our simulations with modest body damping in the hawkmoth. As predicted we observe an upper limit on t_0/T_n where asynchronous oscillations are no longer present due to this dissipation. We include this new figure in the supporting information and we provide it below. Lastly, we acknowledge that energy dissipation in the thorax may be present yet that at expected levels it does not influence the overall phenomena observed:

“Incorporating estimates of active muscle elasticity and body dissipation that may be present in the thorax does not affect the overall conclusions or features of the simulation (see Supplementary Information IID).”

FIG. 5. **Comparison of simulation results with and without body viscous damping.** Top row shows emergent frequency, middle row shows power, and bottom row shows the standard deviation of peak-to-peak amplitude. White lines designate boundary between asynchronous and synchronous modes. Both simulations display the same fundamental phenomena: 1) asynchronous oscillations emerge as the normalized time-to-peak of the delayed stretch activation response (t_0/T_n) increases, and 2) a bridge of smooth sinusoidal wingbeats exists between the asynchronous and synchronous modes of actuation. However, viscous body damping results in the suppression of emergent asynchronous wingbeats for larger t_0/T_n as indicated by the arrow in the upper right plot.

24) Line 224-228: This observation would be highly informative in the introduction for motivating the study instead of presenting it at the end.

These lines referred to some of our motivation for testing these transitions in an at-scale robotic platform. We do not think this is the main motivation for the study as a whole as our goal was not simply to build an asynchronous robotic platform. However, we do agree that this makes a compelling argument for the motivation for this subsection and we have moved it up to be the first argument stated there. In addition, we did have a short motivation for this section in the abstract where we highlighted the development of a new self-excited wingstroke strategy for robotics.

25) " and asynchronous modes as K_r varies, as we predict insects " how was this achieved in the robot? What is the design that makes this possible? Is it off-board or on-board in the sense is it tiny enough to fly? I only found out when reading the methods: all of this information is critical for the reader to understand this work and needs to be communicated in the main manuscript.

The generation of asynchronous forcing (i.e. delayed stretch activation) happens in real-time using a position sensor that measures the motion of the robot actuator, and in Simulink this position versus time is numerically differentiated and the delayed stretch activation response is calculated and output force updated at every time-step (running at 10kHz in real-time mode). Thus, the asynchronous wingbeats in the robot are emergent and result purely from the feedback system. In the current form of the robot the sensing is provided from an external sensor and so the robot cannot directly fly. This is a goal of future work on this robot. We have updated the following text in the discussion to clarify these questions:

“To generate delayed stretch activation in the robobee we used a fiber-optic displacement sensor to estimate wing velocity. The instantaneous wing velocity was supplied to a real-time delayed stretch activation dynamics model (the same as for the robophysical system with parameters adapted for the robobee) and the output of this model was converted to a voltage that was amplified and supplied to the piezoelectric actuator. Thus, we were able to establish a real-time feedback loop between wing velocity and actuator voltage that could generate asynchronous wingbeats of the robobee. By combining the outputs of feedforward synchronous actuation, and feedback delayed stretch activation (Eq. 1) in real-time experiments we were able to demonstrate transitions between synchronous and asynchronous oscillations in this robot.”

26) " Then, we changed K_r linearly from 1 to 0 over two seconds. " how? Clarify this here for the reader.

This is performed by changing the relative output magnitude of the synchronous and asynchronous force (through modulation of K_r) in Simulink real-time which produces an output voltage which is amplified and supplied to the robot's actuator. We have added the following change to the text:

“Then, we changed K_r linearly from 1 to 0 over two seconds in the Simulink real-time control system”

27) " range of (K_r Fig. 3h). " typo, parenthesis should start before Fig.

Fixed.

28) 247-259: too speculative, unclear what benefit a robot would have to switch. Isn't the single one-line advantage that the robot wing flapping control system does not need to pulse each stroke as in insects. This is also what instigated this research so this is a nice closer. Otherwise, it is speculation without significant merit or proof for merit. Less is more. And the space can be used to clarify the rest of the highly informative and valuable work.

Thank you for suggesting a revision of this section. There are two definite advantages that asynchronous actuation can provide to a robot: 1) the separation of wingstroke generation from control (i.e. the control system does not need to pulse each stroke as you suggest), and 2) adaptation of wingbeat frequency to the body resonant frequency through delayed stretch activation (which can change under wing damage). We have shortened this section and highlight these two advantages.

“By capturing one of the key evolutionary innovations that enabled high frequency insect flight, this framework may unlock the potential for robotic systems to benefit from both asynchronous and synchronous actuation modes. For insects, asynchronous muscle enabled the decoupling between muscle contractions and neural input that enables wingbeat frequencies to exceed the limits of neural firing frequency^{2,3,5,50}. An asynchronous flapping-wing robot may benefit from this decoupling of power and control. Moreover, the ability to transition between synchronous and asynchronous modes suggests opportunities for even more versatile control.”

29) " , and we might find these features to lie on a continuum." what is the functional morphological or developmental evidence for this? Show it or remove.

This phrasing was confusing and relates to several of the comments below as well. We have extensively revised the last subsection of the paper to incorporate these. We have revised an earlier section to better indicate how this transition could occur biologically, while acknowledging as a field we still do not have consensus on the necessary and sufficient molecular components necessary for delayed stretch activation. We do know that additional stretch-activated recruitment of myosin heads is involved (likely through troponin isoforms that are mechanically sensitive). We have added this. Moreover the Molloy results, currently expanded to *Manduca*, show the time scales of asynchrony can vary broadly. The reduced, but still present, delayed stretch activation in *Manduca* gives evidence that magnitude relative to synchronous force can too. We make these points earlier in the revised manuscript and refer back to them here to keep this section concise.

30) 274-277: too speculative, I would prefer a discussion of how this could be observed by studying current species diversity, with 1mln counted species and an estimated 10mln total it seems like there should be evidence for these claims in extant species. How can this be found, what should entomologist test / study? Construct and present a research roadmap in the last paragraph.

Our intent was to lay out a hypothesis. We have revised to make this clear and indicate what data would be needed to support it further in the first paragraph of the last section.

31) " Still, the physiological properties and dynamics of indirect spring-wing systems are not necessarily dichotomous." unhelpful summary statement of a final paragraph.

In this revision the final paragraphs we have removed this sentence.

32) " (LaMSA)" please leave acronyms out - also not used again in the main text. The small subfield using LaMSA can also connect the acronym to " latch-mediated spring actuation " and this cryptic wording could be replaced with something clear to a biologist not knowledgeable about biomechanics. Make specific how the latch is mediating. Is this not simply a spring triggered to expand by a latch? Why not keep it simple?

We have removed the references to latch mediated spring actuation. The contrast with what was occurring in insects wing beats was not necessary to make our summary points. To be clear, delayed stretch activation and asynchronous or synchronous flight are not latch mediated.

33) " with a blended actuation strategy enables " the ms is unclear on the in vivo mechanism representing this. Resolve this for the reader.

The revisions to the last three paragraphs (and to Kr earlier) have removed this phrasing and made the possible biological mechanisms more precise.

34) " both time-periodic and stretch-activated forcing " please more discussion on this mechanism instead of all the speculation on less important implications that are not well substantiated.

Thanks for promoting this. As in the prior comment this is already more clear given the earlier changes when we bring up K_r , but we have revised these three paragraphs to clarify this discussion. While the exact mechanism is not known, we have taken the reviewers suggestion to frame this partially as a roadmap to what could be considered in future work.

35) " Via a set of simulation, robophysical, and robotic experiments, we have developed a unifying spring-wing framework that describes synchronous-asynchronous evolutionary transitions in insects and generalizes to robotics. " obsolete at the end, this is a summary sentence that better serves as an opening of a discussion or conclusion section.

We think summarizing the convergent lines of evidence is important but agree that this sentence should not come at the end. We have rephrased to emphasize the convergence and incorporated this in the revision of the last three paragraphs of the main manuscript.

Specific methods section comments:

36) " Iterative force tuning procedure K_r describes the ratio of synchronous and stretch-activated forcing, but does not capture the absolute magnitude of each. " this paragraph seems to explain some of the issues that may also pertain to why the gain K_r is not clearly defined based on in vivo parameters. And why the mechanism isn't mentioned. Please clarify these issues for the reader.

Thank you for prompting a clarification here. We have rewritten the entire force tuning section for clarity and we provide it below. As mentioned earlier we have also clarified our discussion around what K_r is, how it is taken from the data, and how it connects to biological components.

"The parameter K_r describes the relative amounts of synchronous and delayed stretch-activated forcing. To study how an insect that is actuated purely through delayed stretch activation ($K_r = 0$) can transition to being purely actuated through synchronous forcing ($K_r = 1$) we need to establish values of F_a and F_s that produce feasible wingbeat motions in both of these regimes. In the hawkmoth simulation we used a sinusoidal forcing amplitude of $F_s = 2721$ mN to generate wingbeat kinematics that match in-vivo observation of 117 degrees peak-to-peak. This value was previously used to synchronously drive an identical simulation to physiological wingbeat amplitudes³⁸.

However, the wingbeat kinematics in the purely asynchronous regime ($K_r = 0$) are emergent and thus we need to determine an appropriate F_a that can drive our insect model to appropriate wingbeat kinematics. We used a simple iterative force tuning procedure to determine the value of F_{async} such that asynchronous actuation ($K_r = 0$) can produce wingbeats with peak-to-peak amplitude of $\phi_0 = 117^\circ$. We slowly increment the value of μ until the output steady-state wingbeat amplitude is within 1% of the desired amplitude of ϕ_0 . In this way we ensure that the both synchronous ($K_r = 1$) and asynchronous ($K_r = 0$) actuation can produce the same wingbeat amplitudes."

37) "Calculation of K_r for *M. sexta* To calculate K_r for *M. sexta*, we use the following formula, with $F_s = 2721$ mN and $F_a = 320$ mN. " This is key, did I miss this in the main text? where do the input values come from?

We have included the equations, clarified the calculation of K_r and added the source of these measurements in the revised manuscript in the table.

Statistics statement:

replication-please provide sample sizes:
N individuals

n repetitions per N

I assume these have then been conducted once. But this is not clear

randomization-report statistics on this for all the trials, were all stimulations effective, what was the definition used to assess, those kind of insights.

Yes, the reviewer's assumption is correct. This has been clarified in the caption to figure 2. We have also clarified that the robophysical experiments are on one platform but averaged over 15 seconds of flapping for each parameter value. This is added to the methods. Only a single robobee platform was developed because it was meant as a demonstration of feasibility and principle. This is clarified in the caption for Fig. 3.

Reviewer Reports on the First Revision:

Referees' comments:

Referee #1 (Remarks to the Author):

I have studied the revised manuscript and the authors have adequately addressed the analytical concerns raised in the review. I do not have any further suggested revisions, and believe that the paper is now in order for acceptance.

Referee #2 (Remarks to the Author):

Please see attachment.

Referee #3 (Remarks to the Author):

Dear Authors,

Thank you for resolving all my questions, it really helped make this manuscript widely accessible and I enjoyed reading your compelling and helpful clarifications. This is exciting research that Nature's wide readership will find of general interest. The paper will probably end up well-cited long-term. In my mind it is a classic paper that I will return to whenever I have to explain how insect muscles work.

Best regards, David

All three reviewers were enthusiastic about this manuscript, and most of its key conclusions appear well evidenced. The conclusion that asynchronous flight musculature evolved once, with multiple reversions to synchronous flight musculature, is well substantiated. So too is the consequent observation that delayed stretch activation is present (if functionally insignificant) in a lepidopteran with synchronous musculature. It is also clear that the authors have demonstrated transitions between the different kinds of dynamics that they describe, both in simulation and in robotic flappers. Nevertheless, the authors clarifications leave one significant concern that needs to be addressed before I can recommend their manuscript for publication, plus several minor comments.

Major Comments:

My main concern, also reflected in the comments of the other two reviewers, relates to the meaning and representation of the all-important “gain coefficient” or “gain knob” term K_r . There is at best a lack of clarity, and at worst a possible inconsistency, here that the authors’ response to reviewers does not fully address. There are several coupled issues with the description, so I would ask the authors to address these points as a whole:

1. The block diagrams of Fig. 3 are highly misleading. There are several distinct issues here:
 - a. Fig. 3a,d show the “gain knob” (not a helpful term in my view) multiplying the summation of F_{sync} and F_{async} . The only natural reading of these schematics is that $F_m = K_r(F_{sync} + F_{async})$. This conflicts with Eq. 1 which the diagram is supposed to represent, where $F_m = K_r F_{sync} + (1 - K_r)F_{async}$. Separating out the multiplications and summation is an essential step to showing explicitly what is really being modelled in the block diagram. This will inevitably, but appropriately, complicate the figure, eliminating the current obfuscation.
 - b. The block diagrams are missing any differentiation step in going from wing angle to strain rate, which makes the meaning of the block containing $-g * \dot{\epsilon}$ unclear. Nevertheless, the text at line 160 and Eq. 9 clearly define F_{async} as $F_{async} = \mu F_a(-g * \dot{\epsilon})$. Hence, it appears that there is one fixed parameter (F_a) and one free parameter (μ) missing from the block diagram. It is important that these are shown, because the clarifications that the authors have added to the text at lines 833-840 state “We slowly increment the value of μ until the output steady-state wingbeat amplitude is within 1% of the desired amplitude of ϕ_0 .” The implication is that this step is done in the context of the model represented by the block diagram.
2. The preceding points are important to enabling the reader to understand the model, but they perhaps run deeper than that. Splitting the multiplications and summation in the block diagram clarifies that K_r is being used to modify two separable inputs, F_{sync} and F_{async} , in a coupled fashion (see also Eq. 1). This in turn begs the question of what K_r really means in the model, which is a point that was raised by the other reviewers too. The authors now write at lines 186-188 that “Biologically, a high K_r means that the force

change and crossbridge recruitment due to neural activation is large compared to the crossbridge recruitment due to stretch activation.” This is a reasonable characterisation of K_r as it is defined in Eq. 1, but would it not be much clearer to conceive of these physiological changes as modifying F_s and F_a (or μF_a , see next point) rather than combining these in the composite variable K_r ?

3. At line 190, the authors state, without further justification, that “Because in-flight measurements of F_a and F_s are unavailable, we approximate K_r as $K_r = F_s/(F_s + F_a)$.” Whilst this form looks reasonable at first glance, I have found myself unable to justify it from first principles. In particular, the fact that the authors define $F_{sync} = F_s \sin 2\pi f_s t$ and F_{async} as $F_{async} = \mu F_a (-g * \dot{\epsilon})$ leaves me wondering why it is F_s and F_a which appear here, rather than F_s and μF_a (see preceding point). Without the numerical value of the scaling factor μ included, how is it possible to interpret K_r as a measure of the relative importance of the two terms, except at the limits $F_s = 0$ and $F_a = 0$?
4. Treating K_r as an abstract coefficient that can vary on the interval $[0,1]$, we should presumably be able to substitute the “approximation” $K_r = F_s/(F_s + F_a)$ and the definitions $F_{sync} = F_s \sin 2\pi f_s t$ and $F_{async} = \mu F_a (-g * \dot{\epsilon})$ into Eq. 1:

$$F_m = K_r F_{sync} + (1 - K_r) F_{async}$$

and make meaningful sense of the answer. Doing so leads to the following result:

$$F_m = \frac{1}{F_s + F_a} \left(F_s^2 \sin 2\pi f_s t + \mu F_a^2 (-g * \dot{\epsilon}) \right)$$

which is baffling. Please could the authors offer a positive justification for defining $K_r = F_s/(F_s + F_a)$. At present, it feels like there is either some circular logic or recursion in here...

5. Given the preceding point, it is not even clear to me that it is comparing like-with-like to plot the Manduca icon at $K_r = 0.88$ on Fig. 3d. How does the explicit definition $K_r = F_s/(F_s + F_a)$ relate to the abstract definition of $K_r \in [0,1]$ in Eq. 1?
6. Unless the authors believe that F_s and F_a are inevitably physiologically coupled, surely it would have been clearer to treat K_r as a description of underlying changes in F_s and F_a (or μF_a) and to tune these parameters directly in the model?

Minor comments:

1. There are quite a few typos that will need to be addressed, but there is a substantive typographic error at line 839, which reads: “both synchronous ($K_r = 1$) and

asynchronous ($K_r = 1$) actuation.” Whatever K_r really means, its value should be $K_r = 0$ in the second case.

2. Fig. 1 legend. “By iteratively constraining ancestral nodes (see Methods), we find that the synchronous flight muscles of Lepidopterans (including *M. sexta*) are most likely evolved from a asynchronous ancestor (“secondarily synchronous”) as opposed to having only synchronous ancestors (plesiomorphic).” The authors need to take greater care in their use of phylogenetic terminology. Fig. 1 shows that the last common ancestor of Trichoptera and Lepidoptera was (secondarily) synchronous, and that synchronous musculature is a synapomorphy of the clade Trichoptera+Lepidoptera. It follows that synchronous musculature is in fact a symplesiomorphy of the clade Lepidoptera, being present in the last common ancestor of Lepidoptera and its sister clade Trichoptera. Please correct this.
3. Along similar lines, the use of the words “likelihood” and “likely” is a bit sloppy in the section of phylogeny. Likelihood refers to the probability of observing the data under the model, which is not the same thing as the model being the “most likely”. Likewise, the legend for Fig. 1d needs to state that the percentage figures shown are posterior probabilities (assuming that this is what they are).
4. The continuous colormap in Figure EDF6 is perfectly clear, and I fell quite strongly (with one of the other reviewers) that it is not appropriate to add an arbitrary discrete color switch in the colorbars in Fig. 3. Either the continuous data objectively do show distinct domains, or they do not: you should not need to resort to subjective thresholding to demonstrate this.
5. Figure EDF3 shows the posterior probabilities of muscle types at each node, but the legend does not state under what model the posterior probabilities apply. Please add this.
6. Throughout the manuscript, the authors talk about transitions between synchronous and asynchronous musculature via the “bridge”. Their phylogenetic analysis shows that there was one transition from synchronous to asynchronous, and multiple transitions in the opposite direction. Depending on which of these they are most interested in explaining, it is therefore unfortunate that their dynamical tests (Fig. 3) only appear to show transitions from synchronous to asynchronous dynamics. This is understandable, given that they presumably went into this study expecting to have to explain multiple transitions in this direction, as per the earlier literature. However, it would be helpful to have some confirmation that the transition is reversible. There is no reason to think that it shouldn't be, but I would suggest acknowledging this, or ideally demonstrating the point with an example (if available).

Dear Editors of Nature,

Thank you for the second opportunity to revise our manuscript. We are pleased that the initial revisions were received favorably and appreciate the additional points raised. The major change in this revision was to clarify the parameter K_r , especially with regards to how we estimate its value and how this is consistent with thinking of it as an interpolation factor (new wording that replaces the less clear term of “gain”) between synchronous and asynchronous forcing. We also added experiments to show the reversibility of the transition between synchronous and asynchronous flapping in the robophysical model. Overall these changes and the other comments have significantly improved the manuscript and we especially thank Reviewer 2 for their detailed and well-considered comments. These changes have not significantly impacted the overall conclusions of the manuscript, but were important improvements. Below we include a point-by-point response for Reviewer 2, with updated text and figures included in-line where possible and as callouts in some places.

Sincerely,

Simon Sponberg, Georgia Tech
Nick Gravish, UCSD

Reviewer 2

All three reviewers were enthusiastic about this manuscript, and most of its key conclusions appear well evidenced. The conclusion that asynchronous flight musculature evolved once, with multiple reversions to synchronous flight musculature, is well substantiated. So too is the consequent observation that delayed stretch activation is present (if functionally insignificant) in a lepidopteran with synchronous musculature. It is also clear that the authors have demonstrated transitions between the different kinds of dynamics that they describe, both in simulation and in robotic flappers. Nevertheless, the authors clarifications leave one significant concern that needs to be addressed before I can recommend their manuscript for publication, plus several minor comments.

We sincerely thank the reviewer for their second careful reading of our manuscript. It is good to hear that all reviewers agree the main points of the paper are well substantiated. Furthermore, your careful reading has brought to light several technical details that needed to be better resolved/explained in the manuscript which we have done below. Overall, the manuscript has been significantly improved from your comments in both rounds of review.

Major Comments:

My main concern, also reflected in the comments of the other two reviewers, relates to the meaning and representation of the all-important “gain coefficient” or “gain knob” term K_r . There is at best a lack of clarity, and at worst a possible inconsistency, here that the authors’ response to reviewers does not fully address. There are several coupled issues with the description, so I would ask the authors to address these points as a whole:

1. The block diagrams of Fig. 3 are highly misleading. There are several distinct issues here:
 - a. Fig. 3a,d show the “gain knob” (not a helpful term in my view) multiplying the summation of F_{sync} and F_{async} . The only natural reading of these schematics is that $F_m = K_r(F_{sync} + F_{async})$. This conflicts with Eq. 1 which the diagram is supposed to represent, where $F_m = K_r F_{sync} + (1 - K_r)F_{async}$. Separating out the multiplications and summation is an essential step to showing explicitly what is really being modelled in the block diagram. This will inevitably, but appropriately, complicate the figure, eliminating the current obfuscation.
 - b. The block diagrams are missing any differentiation step in going from wing angle to strain rate, which makes the meaning of the block containing $-g * \dot{\epsilon}$ unclear. Nevertheless, the text at line 160 and Eq. 9 clearly define F_{async} as $F_{async} = \mu F_a(-g * \dot{\epsilon})$. Hence, it appears that there is one fixed parameter (F_a) and one

free parameter (μ) missing from the block diagram. It is important that these are shown, because the clarifications that the authors have added to the text at lines 833-840 state “We slowly increment the value of μ until the output steady-state wingbeat amplitude is within 1% of the desired amplitude of ϕ_0 .” The implication is that this step is done in the context of the model represented by the block diagram.

We thank the reviewer for the opportunity to improve the visual representation of the asynchronous feedback loop and consistency. We agree that “gain knob” is potentially confusing and implied the wrong mathematical form of the block diagram. We have replaced it throughout the manuscript and in Fig. 3 with the more appropriate term, “interpolation factor,” since the effect of changing K_r is to linearly interpolate between the two ends of the spectrum. See also below where we resolve the inconsistency in K_r that you point out in your other comments. To reflect this in the block diagram we have changed the K_r knob to a “slider” reminiscent of a linear potentiometer. This representation more accurately visualizes the relationship between F_{async} and F_{sync} as K_r moves from 0 (fully async) to 1 (fully sync). This avoids any graphical or wording suggestion that the relationship should be a gain.

We have also added the derivative block and a scaling block. The block diagram is now consistent with the equations presented.

2. The preceding points are important to enabling the reader to understand the model, but they perhaps run deeper than that. Splitting the multiplications and summation in the block diagram clarifies that K_r is being used to modify two separable inputs, F_{sync} and

F_{async} , in a coupled fashion (see also Eq. 1). This in turn begs the question of what K_r really means in the model, which is a point that was raised by the other reviewers too. The authors now write at lines 186-188 that “Biologically, a high K_r means that the force change and crossbridge recruitment due to neural activation is large compared to the crossbridge recruitment due to stretch activation.” This is a reasonable characterisation of K_r as it is defined in Eq. 1, but would it not be much clearer to conceive of these physiological changes as modifying F_s and F_a (or μF_a , see next point) rather than combining these in the composite variable K_r ?

Yes, we mean for K_r to be characterized in the context of Eq. 1,

$$F_m = K_r F_{sync} + (1 - K_r) F_{async}$$

See the response to comments below where we resolve the confusion on the definition of K_r .

We use this to modify F_{async} rather than F_a because we want to keep F_a as a fixed value that could be taken directly from the stretch-hold physiological experiments. This is not necessarily equal to the force produced during the periodic strain of wingstrokes because it is a static release-and-hold (hence the need for μ). Modifying F_{sync} is equivalent to modifying F_s we just use F_s as a constant amplitude parameter to parallel the time independent F_a . Please also see our responses to the related comments below for more about how we estimate K_r . With those changes, the interpretation is now clearer.

-
3. At line 190, the authors state, without further justification, that “Because in-flight measurements of F_a and F_s are unavailable, we approximate K_r as $K_r = F_s / (F_s + F_a)$.” Whilst this form looks reasonable at first glance, I have found myself unable to justify it from first principles. In particular, the fact that the authors define $F_{sync} = F_s \sin 2\pi f_s t$ and F_{async} as $F_{async} = \mu F_a (-g * \dot{\epsilon})$ leaves me wondering why it is F_s and F_a which appear here, rather than F_s and μF_a (see preceding point). Without the numerical value of the scaling factor μ included, how is it possible to interpret K_r as a measure of the relative importance of the two terms, except at the limits $F_s = 0$ and $F_a = 0$?
 4. Treating K_r as an abstract coefficient that can vary on the interval $[0,1]$, we should presumably be able to substitute the “approximation” $K_r = F_s / (F_s + F_a)$ and the definitions $F_{sync} = F_s \sin 2\pi f_s t$ and $F_{async} = \mu F_a (-g * \dot{\epsilon})$ into Eq. 1:

$$F_m = K_r F_{sync} + (1 - K_r) F_{async}$$

and make meaningful sense of the answer. Doing so leads to the following result:

$$F_m = \frac{1}{F_s + F_a} (F_s^2 \sin 2\pi f_s t + \mu F_a^2 (-g * \dot{\epsilon}))$$

which is baffling. Please could the authors offer a positive justification for defining $K_r = F_s/(F_s + F_a)$. At present, it feels like there is either some circular logic or recursion in here...

We respond to comment 3 and 4 together here. The reviewer is correct to point out an error in the equation used in line 190. We have updated the line in question, as well as the more detailed discussion in the supplementary information to reflect our modified approach. The approximation of $K_r = F_s/(F_s + F_a)$ was a carry-over for earlier ways in which we had tried to formulate the problem and suffers from the challenges identified by the reviewer. Fortunately, the correct use of K_r in the context of the interpolation factor, $F_m = K_r F_{sync} + (1 - K_r) F_{async}$, is how it was used in the rest of the manuscript, simulations and discussion. The role of the paragraph where we had estimated $K_r = F_s/(F_s + F_a)$ was to approximate a value specifically for *Manduca*, but as the reviewer implied, it should use values specifically from our experiments and not F_s , which would lead to confusion with Eq. 1. We have revised this paragraph to provide this estimate in a clearer way based on the description in the next several paragraphs.

The identification of K_r is tricky. Fundamentally, K_r represents the contribution of synchronous force relative to total muscle force (sync + async) from a flight muscle *during flight*. A K_r approaching 0 implies asynchronous (stretch-activated) force comprises an increasing fraction of total muscle force. The primary challenge of determining K_r experimentally is that it is nontrivial to separate the synchronous and asynchronous components from dynamic force measurements because these have to be done under flight conditions. It is already known that the muscle forces taken from work loops underestimate the total force required for flight (Gau, et al. 2022, *Biology Letters*). We can calculate the needed F_m when given wing kinematics through our mechanics equation, but cannot directly measure how much of this comes from F_{sync} and F_{async} given current experiments. We do think there are potentially ways to do this, for example by using genetic knockouts and possibly a combination of real-time x-ray diffraction, physiology and modeling, but no one has yet done this and the interpretation would be confounded by not knowing for sure yet the underlying biological mechanism for delayed stretch activation.

However, we still want to have a way to at least *estimate* K_r for a real system. This is what we were trying to do in the paragraph that introduced the expression for K_r . What we had intended here and what we have now revised is to try to estimate K_r from the quasi-static muscle experiments that are present in the literature. The appropriate way to do this is to use F_{tet} , the maximum neurogenic (calcium activated) force and F_a , the peak stretch activated force when the

muscle is pulled with an amplitude and rate equivalent to *in vivo* stretch, to approximate the relative strength of stretch activation vs. calcium activation. This requires two revisions. First, we should use a new variable \widetilde{K}_r that denotes this is an estimate of K_r and not identical to the K_r that would be measured under flight conditions. Second the expression should include F_{tet} and not F_s . So our quasi-static estimate of \widetilde{K}_r is:

$$\widetilde{K}_r = F_{tet}/(F_a + F_{tet})$$

The other clarification is that this should be interpreted as approximately the proportion of force that comes from synchronous sources relative to total force rather than a ratio of synchronous to asynchronous force. This also resolves the misleading statement that “in-flight measurements of F_a and F_s are unavailable”, which should instead state that we cannot currently estimate from experiment the relative scaling of F_{sync} and F_{async} under flight conditions (and hence we would need to know μ for the actual insect).

Using this modified approach K_r remains interpretable. If the mechanism of stretch activation is indeed stretch activated troponin isoforms then \widetilde{K}_r represents the proportion of calcium activated isoforms to the total number of isoforms (calcium + mechanically activated). This also remains consistent with our interpretation of K_r as an interpolation factor between synchronous and asynchronous regimes. However, we need to take great care in interpretation because finding this proportion from quasi-static measurements is only an approximation of what K_r would be in flight and the troponin isoforms are only one possible mechanism for stretch activation. Still, this allows us to provide an estimate for \widetilde{K}_r for *Manduca* based on our measurements and we emphasize that none of the conclusions are particularly sensitive to estimating \widetilde{K}_r , except for exactly where *Manduca* would appear on the plot. This also provides a way forward for other researchers to estimate \widetilde{K}_r in other insects, if the appropriate quasi-static measurements are taken.

We thank the reviewer for pointing out this inconsistency. We have corrected the equation in question and the surrounding paragraph. We also revised our interpretation section and discuss the difference between K_r and \widetilde{K}_r . We have also added a more detailed discussion in the supplementary information to reflect our updated approach.

-
5. Given the preceding point, it is not even clear to me that it is comparing like-with-like to plot the *Manduca* icon at $K_r = 0.88$ on Fig. 3d. How does the explicit definition $K_r =$

$F_s/(F_s + F_a)$ relate to the abstract definition of $K_r \in [0,1]$ in Eq. 1?

Using the method explained above and in the supplement, we corrected our estimate of $\widetilde{K}_r \approx 0.86$ for *Manduca* and modified figure 3b,c to reflect the new value. As stated above the inconsistent use of K_r was restricted to the placement of *Manduca* on that figure and has now been corrected and clarified to be \widetilde{K}_r .

6. Unless the authors believe that F_s and F_a are inevitably physiologically coupled, surely it would have been clearer to treat K_r as a description of underlying changes in F_s and F_a (or μF_a) and to tune these parameters directly in the model?

The F_{sync} and F_{async} forces (and thus F_s and μF_a) are constructed in simulation to independently generate the two extremes of flight in a mechanical model of the hawkmoth: purely synchronous and purely asynchronous. Thus the interpolation factor, K_r , represents a traversal between these two extremes, and is a parameter to vary between the asynchronous and synchronous states. If instead we were to vary F_s and μF_a to transition between these states there is no guarantee that the extremes would produce flight, and we would need to modify two values to transition between these states. The linear interpolation factor handles exactly this case: it guarantees that as we sweep across transitional states of synchronous + asynchronous flight we can use a single parameter to locate the transitional state, and the extremes ($K_r = 0, 1$) are guaranteed to produce physiologically relevant wingbeats. We could change these values separately but the extremes set the maximum values of F_{sync} and F_{async} . In this context, then we do think that the synchronous and asynchronous forcing are coupled in the sense that there is a total number of crossbridges that can be formed and some are activated by calcium and some recruited by mechanical stretch.

The challenge of associating K_r , μF_a , and F_s is that K_r is interpolating stretch-activated and neural components of forcing during a wingstroke and this cannot be separated in experiment. This was our motivation for finding an approximate K_r to relate the quasi-static force measurements where one can separate the asynchronous and neural components of force to the periodic case. In this case F_s can be set as the force amplitude required by the mechanics to drive wingstrokes of appropriate amplitude in the purely synchronous case, but μ is a required free parameter because we can't directly measure the stretch activation contribution in flight conditions.

Minor comments:

1. There are quite a few typos that will need to be addressed, but there is a substantive typographic error at line 839, which reads: “both synchronous ($K_r = 1$) and asynchronous ($K_r = 1$) actuation.” Whatever K_r really means, its value should be $K_r = 0$ in the second case.

Thanks, this has been resolved. We have also done a few more passes through the paper and will make sure there is a copyediting round too.

2. Fig. 1 legend. “By iteratively constraining ancestral nodes (see Methods), we find that the synchronous flight muscles of Lepidopterans (including *M. sexta*) are most likely evolved from a asynchronous ancestor (“secondarily synchronous”) as opposed to having only synchronous ancestors (plesiomorphic).” The authors need to take greater care in their use of phylogenetic terminology. Fig. 1 shows that the last common ancestor of Trichoptera and Lepidoptera was (secondarily) synchronous, and that synchronous musculature is a synapomorphy of the clade Trichoptera+Lepidoptera. It follows that synchronous musculature is in fact a symplesiomorphy of the clade Lepidoptera, being present in the last common ancestor of Lepidoptera and its sister clade Trichoptera. Please correct this.

We thank the reviewer for their feedback on the Figure 1 legend. We very much agree that the transition back to synchrony was likely at the Trichoptera + Lepidopteran clade. In revising this we decided to remove as much phylogenetic terminology (e.g. plesiomorphy) as possible to make the paper as accessible as possible. We instead describe the relationships directly. We have edited the figure legend to state:

“By iteratively constraining ancestral nodes (see Methods), we find an 87% posterior probability that some node ancestral to Lepidoptera + Trichoptera (including *M. sexta*) was asynchronous (making this clade secondarily synchronous) as opposed all nodes ancestral to Lepidoptera being synchronous (ancestral synchronous).”

3. Along similar lines, the use of the words “likelihood” and “likely” is a bit sloppy in the section of phylogeny. Likelihood refers to the probability of observing the data under the model, which is not the same thing as the model being the “most likely”. Likewise, the legend for Fig. 1d needs to state that the percentage figures shown are posterior probabilities (assuming that this is what they are).

Thanks for suggesting more precise language. We have carefully gone through our revision to clarify this language. The pie charts present at various ancestral nodes do indeed represent the posterior probability of muscle type at that node. We have clarified in the caption of Fig. 1. The unabridged posterior probabilities are present in Figure EDF3 and Supplementary Data Table S2.

4. The continuous colormap in Figure EDF6 is perfectly clear, and I fell quite strongly (with one of the other reviewers) that it is not appropriate to add an arbitrary discrete color switch in the colorbars in Fig. 3. Either the continuous data objectively do show distinct domains, or they do not: you should not need to resort to subjective thresholding to demonstrate this.

We agree and Fig 3 now contains a continuous color plot that shows smooth divergence away from the synchronous frequency. We did keep it as a color spectrum that diverges away from the synchronous frequency (blue) in each direction so that the synchronous frequency matching is clear, but the system is not discrete any more. The added supplemental material motivated by this reviewer clearly shows the phase change behavior.

5. Figure EDF3 shows the posterior probabilities of muscle types at each node, but the legend does not state under what model the posterior probabilities apply. Please add this.

We have added that these are the posterior probabilities for the equal rates model and are in agreement with Figure 1.

6. Throughout the manuscript, the authors talk about transitions between synchronous and asynchronous musculature via the “bridge”. Their phylogenetic analysis shows that there was one transition from synchronous to asynchronous, and multiple transitions in the opposite direction. Depending on which of these they are most interested in explaining, it is therefore unfortunate that their dynamical tests (Fig. 3) only appear to show transitions from synchronous to asynchronous dynamics. This is understandable, given that they presumably went into this study expecting to have to explain multiple transitions in this direction, as per the earlier literature. However, it would be helpful to have some confirmation that the transition is reversible. There is no reason to think that it shouldn't be, but I would suggest acknowledging this, or ideally demonstrating the point with an example (if available).

We thank the reviewer for giving us the opportunity to strengthen our experimental evidence. We ran an additional set of experiments, shown in EDF11, on the dynamically-scaled roboflapper. We chose asynchronous parameters r_3 and μ that produced flapping at 3.2 Hz and ~100 deg peak-to-peak amplitudes. We set the synchronous force amplitude to the asynchronous forcing amplitude and drove the system at four frequencies relative to the emergent asynchronous frequency. We induced 3-second transitions from asynchronous to synchronous and back, 3 times each.

The results show that transitions are possible from asynchronous to synchronous and vice versa, and that only when the synchronous and asynchronous dynamics are frequency-matched do we see smooth transitions (minimal amplitude variations). Note, however, that going from asynchronous to synchronous does appear to induce an amplitude perturbation as the asynchronous phase entrains with the synchronous signal (see EDF8 for more evidence of entrainment). Additionally, since force was matched across frequencies, we see changes in amplitude when $f_s/f_a \neq 1$ due to the resonance properties of the system.

Reviewer Reports on the Second Revision:

Referees' comments:

Referee #2 (Remarks to the Author):

I am glad that the authors found my earlier comments helpful, and appreciate the effort they have gone to in addressing all of the points raised.

My only remaining suggestion would be to consider whether "partition" might be a clearer description of K_r for the reader than "interpolation", but I leave this entirely up to the authors to decide. Either way, the definitions are clear.

This is an outstanding manuscript, and an important contribution to the literature. I congratulate them on it!